# Improving Diffusion-based Inverse Algorithms under Few-Step Constraint via Linear Extrapolation

**Jiawei Zhang**
Tsinghua University
jiawei-z23@mails.tsinghua.edu.cn

**Ziyuan Liu**
Tsinghua University
liuziyua22@mails.tsinghua.edu.cn

**Leon Yan**
Tsinghua University
yansc23@mails.tsinghua.edu.cn

**Gen Li**
CUHK
genli@cuhk.edu.hk

**Yuantao Gu**[*]
Tsinghua University
gyt@tsinghua.edu.cn

## Abstract

Diffusion-based inverse algorithms have shown remarkable performance across various inverse problems, yet their reliance on numerous denoising steps incurs high computational costs. While recent developments of fast diffusion ODE solvers offer effective acceleration for diffusion sampling without observations, their application in inverse problems remains limited due to the heterogeneous formulations of inverse algorithms and their prevalent use of approximations and heuristics, which often introduce significant errors that undermine the reliability of analytical solvers. In this work, we begin with an analysis of ODE solvers for inverse problems that reveals a linear combination structure of approximations for the inverse trajectory. Building on this insight, we propose a canonical form that unifies a broad class of diffusion-based inverse algorithms and facilitates the design of more generalizable solvers. Inspired by the linear subspace search strategy, we propose Learnable Linear Extrapolation (LLE), a lightweight approach that universally enhances the performance of any diffusion-based inverse algorithm conforming to our canonical form. LLE optimizes the combination coefficients to refine current predictions using previous estimates, alleviating the sensitivity of analytical solvers for inverse algorithms. Extensive experiments demonstrate consistent improvements of the proposed LLE method across multiple algorithms and tasks, indicating its potential for more efficient solutions and boosted performance of diffusion-based inverse algorithms with limited steps. Codes for reproducing our experiments are available at https://github.com/weigerzan/LLE_inverse_problem.

## 1 Introduction

Diffusion models have demonstrated remarkable capability in modeling complex data priors [1, 2, 3, 4], which has led to their widespread application in solving inverse problems. Extensive efforts have been devoted to the development of diffusion-based inverse algorithms [5, 6, 7, 8, 9, 10, 11, 12, 13, 14], achieving impressive performance in numerous tasks including inpainting [15], super-resolution [16], deblurring [17], and compressed sensing [18].

One major drawback of diffusion-based inverse algorithms is the need for multiple neural network inferences, resulting in high computational complexity. Directly reducing the number of inference steps often degrades the performance of these inverse algorithms. Therefore, enhancing the performance of diffusion-based inverse algorithms under limited steps has emerged as a critical research

---

[*]The corresponding author is Yuantao Gu.

39th Conference on Neural Information Processing Systems (NeurIPS 2025).

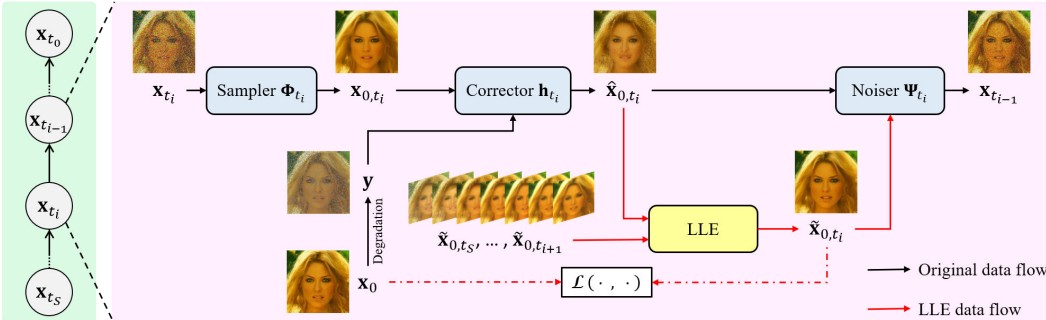

Figure 1: The proposed canonical form of diffusion-based inverse algorithms and the workflow of our LLE method. Our canonical form decomposes an inverse algorithm into three key modules: Sampler, Corrector, and Noiser, providing a unified framework that encompasses a wide range of existing approaches. The LLE method learns a linear combination of the current corrected estimate and previous results to obtain a better estimation of the original image, therefore universally enhancing diffusion-based inverse algorithms' performance under few steps.

direction, leading to several recent attempts such as shortcut sampling [19, 20], learned correction processes [21], and the introduction of conjugate integrators [22]. Recently, high-order diffusion ODE solvers [23, 24, 25, 26, 27] have shown strong performance in the few-step sampling of diffusion models without observations. However, although conceptually these solvers can also accelerate inverse algorithms, two challenges remain in practice. First, not all inverse algorithms admit an explicit ODE or SDE formulation, necessitating a more general framework to unify their analysis and the design of generalizable solvers. Second, the prevalence of approximations and heuristics in existing methods often leads to substantial estimation errors, undermining the reliability of analytical high-order solvers.

To tackle these challenges, we begin with an analysis of ODE solvers in inverse problems from the perspective of posterior sampling, revealing an equivalent interpretation in the form of linear combinations. Based on the insights, we propose a canonical form for diffusion-based inverse algorithms, which serves as a unifying framework for a broad class of existing methods, facilitating a more systematic understanding and more generalizable development of inverse problem solvers. Building on this canonical form, we design a universal enhancement strategy for diffusion-based inverse algorithms inspired by optimal linear subspace search [27]. Our proposed method, termed Learnable Linear Extrapolation (LLE), learns a set of linear combination coefficients to boost current corrected estimate with previous results, thereby mitigating the unreliability of analytical high-order solvers under large estimation errors or heuristics. For linear inverse problems, we further leverage the structure of the observation function by decoupling the coefficients associated with the range space and null space, leading to enhanced performance for LLE.

LLE is highly lightweight and can be trained with 50 diffusion-generated samples within minutes on a single NVIDIA RTX 3090 GPU, which makes it practical and easy to integrate into existing inverse algorithms. We reformulate nine mainstream diffusion-based inverse algorithms into the proposed canonical form and conduct extensive experiments to validate the effectiveness of LLE in both linear and nonlinear tasks. The results demonstrate that LLE effectively improves the performance of these algorithms under few steps, providing a novel and more general perspective to enhance diffusion-based inverse algorithms in the future.

## 2 Backgrounds

### 2.1 Diffusion models

Diffusion models [1, 2, 28] consist of a forward diffusion process that transforms a clean data distribution into a pure Gaussian distribution, and a reverse denoising process that generates clean data from the noise distribution. In this paper, we consider the setting of Variance Preserving (VP) diffusion. Given a target data distribution $q(\mathbf{x}_0)$, the forward diffusion process is modeled by the

following stochastic differential equation (SDE):

$$\mathrm{d}\mathbf{x}_t = f(t)\mathbf{x}_t\mathrm{d}t + g(t)\mathrm{d}\mathbf{w}_t, \quad t \in [0, T], \tag{1}$$

where $\mathbf{x}_0 \sim q(\mathbf{x}_0)$ and $\mathbf{w}_t$ is a standard Wiener process. VP diffusion specifics the drift term $f(t)$ and the diffusion term $g(t)$ as

$$f(t) = \frac{\mathrm{d}\log\sqrt{\overline{\alpha}_t}}{\mathrm{d}t}, \quad g^2(t) = \frac{\mathrm{d}(1-\overline{\alpha}_t)}{\mathrm{d}t} - 2\frac{\mathrm{d}\log\sqrt{\overline{\alpha}_t}}{\mathrm{d}t}(1-\overline{\alpha}_t),$$

respectively, where $\overline{\alpha}_t$ is a predefined monotonic parameter controlling the noise level at each timestep, with $\overline{\alpha}_0 = 1$ and $\overline{\alpha}_T = 0$.

The forward SDE (1) can be exactly reversed as [29]

$$\mathrm{d}\mathbf{x}_t = \left(f(t)\mathbf{x}_t - g^2(t)\nabla_{\mathbf{x}_t}\log q_t(\mathbf{x}_t)\right)\mathrm{d}t + g(t)\mathrm{d}\overline{\mathbf{w}}_t,$$

where $\overline{\mathbf{w}}_t$ is the reverse-time standard Wiener process, $\mathbf{x}_T \sim \mathcal{N}(\mathbf{0}, \mathbf{I})$, and $\nabla_{\mathbf{x}_t}\log q_t(\mathbf{x}_t)$ is known as the (Stein) score function [30, 31]. Practical implementation aims to estimate the unknown score function $\nabla_{\mathbf{x}_t}\log q_t(\mathbf{x}_t)$ via a parameterized neural network $\boldsymbol{\epsilon}_{\boldsymbol{\theta}}(\mathbf{x}_t, t)$ as

$$\boldsymbol{\epsilon}_{\boldsymbol{\theta}}(\mathbf{x}_t, t) = -\sqrt{1-\overline{\alpha}_t}\nabla_{\mathbf{x}_t}\log q_t(\mathbf{x}_t), \tag{2}$$

which can be learned via score matching [32, 33].

Song *et al.*[34] demonstrate that the above reverse SDE shares the same marginal distributions as the following reverse-time ordinary differential equation (ODE):

$$\frac{\mathrm{d}\mathbf{x}_t}{\mathrm{d}t} = f(t)\mathbf{x}_t - \frac{1}{2}g^2(t)\nabla_{\mathbf{x}_t}\log q_t(\mathbf{x}_t), \tag{3}$$

which facilitates the introduction of efficient ODE solvers to enhance sampling efficiency. Using a first-order solver to solve the diffusion ODE (3) leads to the iterative formula of the well-known DDIM sampler [3], which is

$$\mathbf{x}_{t-1} = \sqrt{\overline{\alpha}_{t-1}}\frac{\mathbf{x}_t - \sqrt{1-\overline{\alpha}_t}\boldsymbol{\epsilon}_{\boldsymbol{\theta}}(\mathbf{x}_t, t)}{\sqrt{\overline{\alpha}_t}} + \sqrt{1-\overline{\alpha}_{t-1}}\boldsymbol{\epsilon}_{\boldsymbol{\theta}}(\mathbf{x}_t, t).$$

## 2.2 High-order solvers for diffusion ODE

Numerous works have developed a variety of fast ODE solvers for diffusion models [35, 23, 24, 25, 26, 36, 37, 38, 39, 40]. Recent studies have adopted exponential integrators [41] to solve the diffusion ODE as follows

$$\mathbf{x}_t = \frac{\sqrt{\overline{\alpha}_t}}{\sqrt{\overline{\alpha}_s}}\mathbf{x}_s - \sqrt{\overline{\alpha}_t}\int_{\lambda(s)}^{\lambda(t)} \exp(-\lambda)\boldsymbol{\epsilon}_{\boldsymbol{\theta}}(\mathbf{x}_{t(\lambda)}, t(\lambda))\mathrm{d}\lambda,$$

where $\lambda(t) = \log\sqrt{\overline{\alpha}_t/(1-\overline{\alpha}_t)}$. High-order diffusion ODE-solvers [23, 24, 26, 25] leverage the Taylor expansions of $\boldsymbol{\epsilon}_{\boldsymbol{\theta}}(\cdot, \cdot)$ estimated from previous score functions to solve the integral more accurately, which reduces the dependency on small step sizes and enables efficient sampling with fewer steps. Despite differences in derivation and specific formulation, these solvers share a common structure in that $\mathbf{x}_t$ can be expressed as a linear combination of the initial point and the scores from all previous steps [27], i.e.,

$$\mathbf{x}_t \in \mathrm{span}\left\{\mathbf{x}_T, \boldsymbol{\epsilon}_{\boldsymbol{\theta}}(\mathbf{x}_T, T), \ldots, \boldsymbol{\epsilon}_{\boldsymbol{\theta}}(\mathbf{x}_{t+1}, t+1)\right\}. \tag{4}$$

Thus [27] proposes the optimal subspace search method to optimize the coefficients of $\mathbf{x}_t$ in the subspace shown in (4), aiming to align the trajectory of a few-step solver with a target solver using more steps (e.g., a 999-step DDIM). The learnable solver achieves remarkable performance with few steps while remaining lightweight as it only requires learning a set of linear combination coefficients per timestep. In this paper, we design a lightweight learnable method tailored for inverse algorithms to generally enhance their performance under few steps leveraging similar insights from the optimal subspace search approach.

### 2.3 Diffusion-based inverse algorithms

Consider the degradation equation

$$\mathbf{y} = \mathcal{A}(\mathbf{x}_0) + \sigma_{\mathbf{y}}\mathbf{n}, \quad \mathbf{n} \sim \mathcal{N}(\mathbf{0}, \mathbf{I}), \tag{5}$$

where $\mathcal{A}$ is the degradation function, $\mathbf{y}$ is the observation, and $\sigma_{\mathbf{y}}$ is the standard deviation of the observation noise. The goal of inverse problems is to recover the original data $\mathbf{x}_0$ from the degraded observation $\mathbf{y}$. The ability of diffusion models to capture complex data priors has inspired a wide range of studies exploring their applications in inverse problems from diverse perspectives.

DDRM [5] defines a new set of variational distributions to approximate the posterior $q(\mathbf{x}_0|\mathbf{y})$ and derives an inverse algorithm for linear problems based on projection operations. DDNM [6] directly employs the null-range decomposition and heuristically projects the intermediate estimation of $\mathbf{x}_0$ during sampling onto the hyperplane corresponding to the linear observation. Posterior sampling methods [8, 7, 10] attempt to estimate the conditional score $\nabla_{\mathbf{x}_t} \log q_t(\mathbf{x}_t|\mathbf{y})$ to sample from the posterior distribution $q(\mathbf{x}_0|\mathbf{y})$. DiffPIR [9] proposes a plug-and-play approach by guiding the intermediate estimation of $\mathbf{x}_0$ with a proximal point problem and further refining the reverse sampling trajectory. ReSample [12], originally designed for latent diffusion, can be interpreted as optimizing the intermediate estimation via gradient descent to enforce hard data consistency and guiding the reverse sampling trajectory with a resample algorithm. More recently, maximum a posteriori (MAP)-based methods such as RED-diff [11] have gained attention, which explicitly express the data prior $q(\mathbf{x}_0)$ using diffusion models and solve inverse problems by optimizing $\max_{\mathbf{x}_0} q(\mathbf{x}_0|\mathbf{y})$. DAPS [14] adopts Langevin dynamics to enforce consistency with the observations and progressively anneals the timestep to achieve noise reduction.

Despite the effectiveness of diffusion-based inverse algorithms, they typically require 100 to 1000 steps during inference. In this paper, we first establish a canonical form for these algorithms and then propose a lightweight and general approach to enhance their performance under few steps, achieving a better trade-off between performance and computational efficiency.

## 3 Analysis of ODE solvers in inverse problems

It can be shown that the associated probability flow ODE for sampling from the posterior distribution $p(\mathbf{x}_0|\mathbf{y})$ is given by

$$\frac{\mathrm{d}\mathbf{x}_t}{\mathrm{d}t} = f(t)\mathbf{x}_t - \frac{1}{2}g^2(t)\nabla_{\mathbf{x}_t} \log p(\mathbf{x}_t|\mathbf{y}). \tag{6}$$

In the context of inverse problems, the primary focus is to restore the clean signal. Thus we define

$$\mathbf{x}_0(t, \mathbf{y}) = \frac{\mathbf{x}_t + (1 - \overline{\alpha}_t)\nabla_{\mathbf{x}_t} \log p(\mathbf{x}_t|\mathbf{y})}{\sqrt{\overline{\alpha}_t}}. \tag{7}$$

To assist our analysis, we assume the temporal discretization for the numerical solver as $\{t_S, t_{S-1}, \ldots, t_0\} \subseteq [0, T]$ with $t_S > t_{S-1} > \cdots > t_0 = 0$, and $S$ denoting the total number of steps. Then the evolution of $\mathbf{x}_0(t, \mathbf{y})$ from $t_{i+1}$ to $t_i$ can be described as the following proposition.

**Proposition 1.** *The evolution of $\mathbf{x}_0(t, \mathbf{y})$ follows*

$$\mathbf{x}_0(t_i, \mathbf{y}) = \frac{\sqrt{1 - \overline{\alpha}_{t_i}}}{\sqrt{1 - \overline{\alpha}_{t_{i+1}}}} \frac{\sqrt{\overline{\alpha}_{t_{i+1}}}}{\sqrt{\overline{\alpha}_{t_i}}} \mathbf{x}_0(t_{i+1}, \mathbf{y}) - \frac{\sqrt{1 - \overline{\alpha}_{t_i}}}{\sqrt{\overline{\alpha}_{t_i}}} \underbrace{\int_{t_{i+1}}^{t_i} \phi(\tau)\mathbf{x}_0(\tau, \mathbf{y})\mathrm{d}\tau}_{(*)}$$

$$+ \frac{\sqrt{1 - \overline{\alpha}_i}}{\sqrt{\overline{\alpha}_{t_i}}} \underbrace{\left( \sqrt{1 - \overline{\alpha}_{t_i}} \nabla_{\mathbf{x}_{t_i}} \log p(\mathbf{x}_{t_i}|\mathbf{y}) - \sqrt{1 - \overline{\alpha}_{t_{i+1}}} \nabla_{\mathbf{x}_{t_{i+1}}} \log p(\mathbf{x}_{t_{i+1}}|\mathbf{y}) \right)}_{(**)}, \tag{8}$$

*where $\phi(\tau) = \frac{g^2(\tau)\sqrt{\overline{\alpha}_\tau}}{2(1 - \overline{\alpha}_\tau)^{3/2}}$.*

The detailed derivations are deferred to Appendix B. Note that term $(*)$ contains an integral involving $\mathbf{x}_0(\tau, \mathbf{y})$, which can be approximated using a standard $k$-th order Taylor expansion, i.e.,

$$\int_{t_{i+1}}^{t_i} \phi(\tau)\mathbf{x}_0(\tau, \mathbf{y})\mathrm{d}\tau = \sum_{l=0}^{k} \frac{\mathrm{d}^l \mathbf{x}_0(t_{i+1}, \mathbf{y})}{\mathrm{d}t_{i+1}^l} \int_{t_{i+1}}^{t_i} \frac{(\tau - t_{i+1})^l}{l!} \phi(\tau)\mathrm{d}\tau + \mathcal{O}((t_{i+1} - t_i)^{k+1}), \tag{9}$$

The high-order derivatives $\frac{\mathrm{d}^l \mathbf{x}_0(t_{i+1},\mathbf{y})}{\mathrm{d}t_{i+1}^l}$ in (9) can be estimated using multi-step methods. In particular, under uniform time discretization $t_{j+1} - t_j = h$, $\forall j$, the forward differences approximation [42, 43] of the $l$-th derivative yields

$$\frac{\mathrm{d}^l \mathbf{x}_0(t_{i+1},\mathbf{y})}{\mathrm{d}t_{i+1}^l} \approx \frac{1}{h^l}\sum_{j=0}^{l}(-1)^{l-j}\binom{l}{j}\mathbf{x}_0(t_{i+1+j},\mathbf{y}), \tag{10}$$

which implies that the Taylor expansion approximation for $(*)$ is a linear combination of previous steps $\mathbf{x}_0(t_j,\mathbf{y})$, for $j \geq i+1$.

Term $(**)$ involves the score at both time steps $t_{i+1}$ and $t_i$, which can be reformulated as

$$\frac{\sqrt{\overline{\alpha}_{t_i}}}{\sqrt{1-\overline{\alpha}_{t_i}}}\left(\underbrace{\frac{(1-\overline{\alpha}_{t_i})\nabla_{\mathbf{x}_{t_i}}\log p(\mathbf{x}_{t_i}|\mathbf{y}) + \hat{\mathbf{x}}_{t_i}}{\sqrt{\overline{\alpha}_{t_i}}}}_{\text{An estimate of } \mathbf{x}_0(t_i,\mathbf{y})} - \mathbf{x}_0(t_{i+1},\mathbf{y})\right), \tag{11}$$

where

$$\hat{\mathbf{x}}_{t_i} = \sqrt{\overline{\alpha}_{t_i}}\mathbf{x}_0(t_{i+1},\mathbf{y}) - \sqrt{(1-\overline{\alpha}_{t_i})(1-\overline{\alpha}_{t_{i+1}})}\nabla_{\mathbf{x}_{t_{i+1}}}\log p(\mathbf{x}_{t_{i+1}}|\mathbf{y}). \tag{12}$$

Note that $\hat{\mathbf{x}}_{t_i}$ serves as the sample calculated by a deterministic DDIM sampler. Consequently, the reformulation (11) can be interpreted as a linear combination of a coarse estimation at the current time step $t_i$ and the previous result at time $t_{i+1}$. Based on the above analysis on terms $(*)$ and $(**)$, we arrive at the following conclusion.

**Corollary 2.** *Approximating the evolution of $\mathbf{x}_0(t,\mathbf{y})$ from $t_{i+1}$ to $t_i$ can be interpreted as computing a linear combination of an estimate of the current step $t_i$ and all previous results at $t_j$ for $\forall j \geq i+1$.*

We note that many inverse algorithms introduce random noise, which makes them more naturally aligned with the SDE perspective. We provide a detailed analysis of the SDE case in Appendix B.3 and obtain conclusions similar to those in Corollary 2.

This insight provides a foundation for generalizing diffusion ODE and SDE solvers to a broad class of inverse algorithms. Conceptually, for most algorithms that are difficult to characterize via posterior sampling, we reinterpret $\mathbf{x}_0(t,\mathbf{y})$ as an estimate of the clean sample at time $t$ corrected with the observation $\mathbf{y}$. Meanwhile, the linear combination structure enables learning the combination coefficients in a data-driven manner, thus avoiding the failure of analytical coefficients when the posterior score estimation is highly inaccurate or heuristic.

## 4   Designing unified solvers for inverse algorithms

In this section, we first establish a canonical form that unifies a wide range of diffusion-based inverse algorithms and clarify its connection to the analysis in Section 3. We then propose the LLE method, which is compatible with any algorithm adhering to the canonical form and effectively improves their performance in the few-step regime by learning a set of linear combination coefficients.

### 4.1   A canonical form for diffusion-based inverse algorithms

As illustrated in Figure 1, our proposed canonical form decomposes each iteration of an inverse algorithm into three modules. Below we detail these modules at step $t_i$ for any $1 \leq i \leq S$.

**(1) Sampler** generates an estimate of the clean data $\mathbf{x}_{0,t_i}$ from the noisy input $\mathbf{x}_{t_i}$ using a pre-trained diffusion model:

$$\mathbf{x}_{0,t_i} = \mathbf{\Phi}_{t_i}(\mathbf{x}_{t_i}). \tag{13}$$

**(2) Corrector** incorporates the observation $\mathbf{y}$ to refine the sampled result, ensuring consistency with the measurement:

$$\hat{\mathbf{x}}_{0,t_i} = \mathbf{h}_{t_i}(\mathbf{x}_{0,t_i}, \mathcal{A}, \mathbf{y}). \tag{14}$$

**Algorithm 1** Training of LLE.

---

**Require:** Pretrained diffusion model $\epsilon_{\boldsymbol{\theta}}$, discrete timesteps $t_S, t_{S-1}, \ldots, t_0$, inverse algorithm in the canonical form $\boldsymbol{\Phi}_{t_i}$, $\mathbf{h}_{t_i}$, $\boldsymbol{\Psi}_{t_i}$, reference samples $\mathbf{x}_0^{(1)}, \ldots, \mathbf{x}_0^{(N)}$, observation function $\mathcal{A}$, observation noise deviation $\sigma_{\mathbf{y}}$.

1: **for** $n = 1$ to $n = N$ **do**
2:      Get observation $\mathbf{y}^{(n)}$ as (5);
3: **end for**
4: Initialize $\left\{\mathbf{x}_{t_S}^{(n)}\right\}_{n=1}^N$ as $\mathbf{x}_{t_S}^{(n)} \sim \mathcal{N}(\mathbf{0}, \mathbf{I})$;
5: **for** $i = S$ to $i = 1$ **do**
6:      **for** $n = 1$ to $n = N$ **do**
7:          Calculate $\hat{\mathbf{x}}_{0,t_i}^{(n)} = \mathbf{h}_{t_i}\left(\boldsymbol{\Phi}_{t_i}\left(\mathbf{x}_{t_i}^{(n)}\right), \mathcal{A}, \mathbf{y}^{(n)}\right)$;
8:      **end for**
9:      Optimizing $\gamma_{t_i,0}, \ldots, \gamma_{t_i,S-i}$ as (19);
10:      **for** $n = 1$ to $n = N$ **do**
11:          Calculate $\tilde{\mathbf{x}}_{0,t_i}^{(n)}$ as (18);
12:          Update $\mathbf{x}_{t_{i-1}}^{(n)} = \boldsymbol{\Psi}_{t_i}(\tilde{\mathbf{x}}_{0,t_i}^{(n)})$;
13:      **end for**
14: **end for**
15: **return** Coefficients $\gamma_{t_i,0}, \ldots, \gamma_{t_i,S-i}$ for $i = S, \ldots, 1$

---

**Algorithm 2** Inference with LLE.

---

**Require:** Pretrained diffusion model $\epsilon_{\boldsymbol{\theta}}$, discrete timesteps $t_S, t_{S-1}, \ldots, t_0$, observation $\mathbf{y}$, observation function $\mathcal{A}$, observation noise deviation $\sigma_{\mathbf{y}}$, inverse algorithm in the canonical form $\boldsymbol{\Phi}_{t_i}$, $\mathbf{h}_{t_i}$, $\boldsymbol{\Psi}_{t_i}$, optimized coefficients $\gamma_{t_i,0}, \ldots, \gamma_{t_i,S-i}$ for $i = S, \ldots, 1$.

1: Initialize $\mathbf{x}_{t_S} \sim \mathcal{N}(\mathbf{0}, \mathbf{I})$;
2: **for** $i = S$ to $i = 1$ **do**
3:      Calculate $\hat{\mathbf{x}}_{0,t_i} = \mathbf{h}_{t_i}\left(\boldsymbol{\Phi}_{t_i}\left(\mathbf{x}_{t_i}\right), \mathcal{A}, \mathbf{y}\right)$;
4:      Calculate $\tilde{\mathbf{x}}_{0,t_i}$ as (18);
5:      Update $\mathbf{x}_{t_{i-1}} = \boldsymbol{\Psi}_{t_i}(\tilde{\mathbf{x}}_{0,t_i})$;
6: **end for**
7: **return** $\mathbf{x}_{t_0}$

---

**(3) Noiser** maps the corrected clean sample back to the next noise level for the subsequent iteration:

$$\mathbf{x}_{t_{i-1}} = \boldsymbol{\Psi}_{t_i}(\hat{\mathbf{x}}_{0,t_i}). \tag{15}$$

One immediate advantage of this decomposition is that (14) broadens the definition of $\mathbf{x}_0(t_i, \mathbf{y})$ in (7), which is applicable to a wider range of inverse algorithms beyond posterior sampling, including but not limited to projection-based and plug-and-play approaches. Another benefit lies in that, by decoupling an inverse algorithm into discrete modules, we can reorganize these modules and directly obtain an estimate at $t_i$ from the results at $t_{i+1}$ as

$$\hat{\mathbf{x}}_{0,t_i} = \mathbf{h}_{t_i} \circ \boldsymbol{\Phi}_{t_i} \circ \boldsymbol{\Psi}_{t_{i+1}}(\hat{\mathbf{x}}_{0,t_{i+1}}), \tag{16}$$

which provides an effective alternative to the approximation term in (11). Consequently, we can construct unified solvers for any inverse algorithms conforming to the proposed canonical form, requiring only the specification of a set of linear combination coefficients, which we propose to optimize in a data-driven manner.

This modular decomposition is applicable to a broad class of inverse algorithms. In this work, we reformulate nine mainstream inverse algorithms into the proposed canonical form, with detailed decompositions presented in Appendix E.

### 4.2 Iterative learnable linear extrapolation

Intuitively, the proposed LLE method attempts to enhance the performance of inverse algorithms under few steps by iteratively learning a linear combination of the corrected sample (14) at $t_i$ and

all previous results at $t_j, j > i$, to obtain a better estimate $\tilde{\mathbf{x}}_{0,t_i}$. Below, we introduce the training objective of LLE at timestep $t_i$ inductively.

Consider a reference training set generated by the diffusion model $\mathbb{X} = \left\{ \mathbf{x}_0^{(1)}, \mathbf{x}_0^{(2)}, \ldots, \mathbf{x}_0^{(N)} \right\}$, where $N$ is the number of reference samples. Suppose we have obtained the estimates from the previous $S - i$ steps, i.e., $\tilde{\mathbf{x}}_{0,t_k}^{(n)}, k = i + 1, \ldots, S, n = 1, \ldots, N$. Then we replace each $\hat{\mathbf{x}}_{0,t_{i+1}}^{(n)}$ with $\tilde{\mathbf{x}}_{0,t_{i+1}}^{(n)}$ in (16) and compute an estimate at step $t_i$ as

$$\hat{\mathbf{x}}_{0,t_i}^{(n)} = \mathbf{h}_{t_i} \left( \mathbf{\Phi}_{t_i} \left( \mathbf{\Psi}_{t_{i+1}} \left( \tilde{\mathbf{x}}_{0,t_{i+1}}^{(n)} \right) \right), \mathcal{A}, \mathbf{y}^{(n)} \right), \tag{17}$$

where $\mathbf{y}^{(n)}$ represents the observation of $\mathbf{x}_0^{(n)}$. Now we optimize a set of linear combination coefficients, denoted as $\gamma_{t_i,0}, \ldots, \gamma_{t_i,S-i}$, and update the estimates at step $t_i$ as

$$\tilde{\mathbf{x}}_{0,t_i}^{(n)} = \gamma_{t_i,S-i} \hat{\mathbf{x}}_{0,t_i}^{(n)} + \sum_{j=0}^{S-i-1} \gamma_{t_i,j} \tilde{\mathbf{x}}_{0,t_{S-j}}^{(n)}. \tag{18}$$

Unlike unconditional generation tasks that often require sampling long trajectories as the training targets for learning fast solvers, inverse problems provide explicit supervision in the form of known ground-truth reconstructions. This eliminates the need to sample or store complete trajectories, significantly reducing training cost. As a result, the optimization objective of LLE is formulated as:

$$\min_{\gamma_{t_i,0}, \ldots, \gamma_{t_i,S-i}} \mathbb{E}_{n \sim \mathcal{U}\{1,\ldots,N\}} \mathcal{L} \left( \tilde{\mathbf{x}}_{0,t_i}^{(n)}, \mathbf{x}_0^{(n)} \right), \tag{19}$$

where the loss function is chosen as

$$\mathcal{L} \left( \tilde{\mathbf{x}}_{0,t_i}^{(n)}, \mathbf{x}_0^{(n)} \right) = \mathcal{L}_{\text{MSE}} \left( \tilde{\mathbf{x}}_{0,t_i}^{(n)}, \mathbf{x}_0^{(n)} \right) + \omega \mathcal{L}_{\text{LPIPS}} \left( \tilde{\mathbf{x}}_{0,t_i}^{(n)}, \mathbf{x}_0^{(n)} \right). \tag{20}$$

$\mathcal{L}_{\text{MSE}}$ is the Mean-Square Error loss that measures the pixel-level discrepancy between samples, while $\mathcal{L}_{\text{LPIPS}}$ represents the Learned Perceptual Image Patch Similarity [44] loss that quantifies the perceptual discrepancy between two samples. $\omega$ is introduced in our loss function (20) as a weight to balance the MSE loss and LPIPS loss. Smaller $\omega$ encourages the algorithm to minimize MSE, resulting in higher PSNR, whereas a larger $\omega$ biases the algorithm toward minimizing LPIPS, leading to higher perceptual similarity.

Once all coefficients are optimized, we fix them and perform inference in exactly the same manner as the data flow in the training process. The pseudo-codes for the training and inference of LLE are presented in Algorithm 1 and Algorithm 2, respectively.

### 4.3 Decoupled coefficients for linear problems

For linear inverse problems, we propose using decoupled linear combination coefficients to estimate the null-space and range-space components of a sample with respect to the observation matrix separately. The rationale is that observations impose strong constraints on the range-space component, while the null-space component remains more flexible. Consequently, decoupled sets of linear combination coefficients would perform better to capture the differences between these components.

Consider the observation equation $\mathbf{y} = \mathbf{A}\mathbf{x}_0 + \sigma_{\mathbf{y}} \mathbf{n}$, where $\mathbf{A}$ is the observation matrix. The range-space and null-space components of a sample $\mathbf{x}$ with respect to $\mathbf{A}$ are denoted as $\mathbf{x}^{\parallel} = \mathbf{A}^{\dagger} \mathbf{A} \mathbf{x}$ and $\mathbf{x}^{\perp} = \left( \mathbf{I} - \mathbf{A}^{\dagger} \mathbf{A} \right) \mathbf{x}$, respectively, where $\mathbf{A}^{\dagger}$ represents the Moore-Penrose pseudoinverse of matrix $\mathbf{A}$. At timestep $t_i$, we optimize a set of coefficients $\gamma_{t_i,j}^{\parallel}, \gamma_{t_i,j}^{\perp}, j = 0, \ldots, S - i$ to estimate $\tilde{\mathbf{x}}_{0,t_i}^{(n)}$ as

$$\tilde{\mathbf{x}}_{0,t_i}^{(n)} = \tilde{\mathbf{x}}_{0,t_i}^{(n),\parallel} + \tilde{\mathbf{x}}_{0,t_i}^{(n),\perp}, \tag{21}$$

where

$$\tilde{\mathbf{x}}_{0,t_i}^{(n),\parallel} = \gamma_{t_i,S-i}^{\parallel} \hat{\mathbf{x}}_{0,t_i}^{(n),\parallel} + \sum_{j=0}^{S-i-1} \gamma_{t_i,j}^{\parallel} \tilde{\mathbf{x}}_{0,t_{S-j}}^{(n),\parallel}, \quad \tilde{\mathbf{x}}_{0,t_i}^{(n),\perp} = \gamma_{t_i,S-i}^{\perp} \hat{\mathbf{x}}_{0,t_i}^{(n),\perp} + \sum_{j=0}^{S-i-1} \gamma_{t_i,j}^{\perp} \tilde{\mathbf{x}}_{0,t_{S-j}}^{(n),\perp}. \tag{22}$$

The optimization method and objective remain consistent with those described in Section 4.2.

Table 1: Results of DDNM, DDRM, ΠGDM, DMPS, and DPS on noisy linear tasks ($\sigma_{\mathbf{y}} = 0.05$) on the CelebA-HQ dataset using 3 and 5 steps. The better results are bolded.

| Steps | Algorithm | Strategy | PSNR↑ / SSIM↑ / LPIPS↓ | | | |
|---|---|---|---|---|---|---|
| | | | Deblur (aniso) | Inpainting | 4× SR | CS 50% |
| 3 | DDNM | - | 27.80 / 0.758 / 0.319 | 16.64 / 0.442 / 0.492 | 27.09 / **0.773** / **0.296** | 16.55 / 0.441 / 0.539 |
| | | LLE | **28.08** / **0.784** / **0.291** | **24.38** / **0.552** / **0.433** | **27.84** / 0.770 / 0.299 | **17.29** / **0.473** / **0.520** |
| | DDRM | - | 27.68 / **0.795** / 0.277 | 16.68 / 0.489 / 0.440 | 26.71 / **0.764** / **0.277** | 16.58 / 0.495 / 0.478 |
| | | LLE | **27.69** / **0.795** / **0.271** | **24.53** / **0.625** / **0.406** | **27.49** / 0.761 / 0.295 | **17.07** / **0.527** / **0.470** |
| | ΠGDM | - | 20.39 / 0.536 / 0.520 | 19.57 / 0.533 / 0.495 | 20.73 / 0.552 / 0.500 | 16.79 / 0.472 / 0.555 |
| | | LLE | **21.73** / **0.588** / **0.489** | **20.72** / **0.571** / **0.483** | **21.54** / **0.583** / **0.490** | **17.30** / **0.487** / **0.540** |
| | DMPS | - | 14.02 / 0.174 / 0.747 | 11.91 / 0.100 / 0.802 | 15.53 / 0.235 / **0.763** | 12.45 / 0.102 / 0.791 |
| | | LLE | **14.10** / **0.177** / **0.744** | **11.97** / **0.104** / **0.800** | **18.76** / **0.253** / 0.776 | **12.52** / **0.105** / **0.789** |
| | DPS | - | 23.59 / 0.650 / 0.415 | 23.57 / 0.558 / 0.497 | **25.49** / 0.647 / 0.528 | 14.30 / 0.359 / 0.636 |
| | | LLE | **24.59** / **0.675** / **0.405** | **27.51** / **0.748** / **0.366** | 24.57 / **0.666** / **0.465** | **15.83** / **0.468** / **0.591** |
| 5 | DDNM | - | 29.63 / 0.819 / 0.259 | 22.76 / 0.550 / 0.431 | 28.97 / **0.818** / 0.262 | 18.20 / 0.474 / 0.491 |
| | | LLE | **29.82** / **0.831** / **0.239** | **26.35** / **0.659** / **0.366** | **29.02** / 0.806 / **0.252** | **19.41** / **0.536** / **0.441** |
| | DDRM | - | 29.29 / 0.826 / 0.244 | 23.21 / 0.717 / 0.286 | 28.58 / **0.811** / **0.247** | 18.34 / 0.599 / 0.389 |
| | | LLE | **29.36** / **0.827** / **0.235** | **26.09** / **0.780** / **0.263** | **28.64** / 0.795 / 0.249 | **19.41** / **0.632** / **0.363** |
| | ΠGDM | - | 22.46 / 0.653 / 0.441 | 22.63 / 0.695 / 0.376 | 21.93 / 0.662 / 0.416 | 16.32 / 0.518 / 0.541 |
| | | LLE | **25.95** / **0.734** / **0.347** | **25.21** / **0.735** / **0.336** | **25.67** / **0.731** / **0.338** | **19.33** / **0.591** / **0.448** |
| | DMPS | - | 21.20 / 0.404 / 0.562 | 14.71 / 0.164 / 0.738 | 22.95 / 0.498 / 0.541 | 14.85 / 0.165 / 0.719 |
| | | LLE | **21.63** / **0.406** / **0.558** | **16.45** / **0.186** / **0.690** | **24.66** / **0.625** / **0.437** | **15.57** / **0.180** / **0.699** |
| | DPS | - | 23.94 / 0.680 / 0.369 | 25.14 / 0.617 / 0.434 | 26.07 / 0.675 / 0.470 | 17.12 / 0.515 / 0.481 |
| | | LLE | **25.56** / **0.722** / **0.326** | **27.42** / **0.796** / **0.303** | **26.63** / **0.758** / **0.315** | **17.73** / **0.536** / **0.468** |

## 5 Experiments

In this section, we validate the effectiveness of LLE across various inverse algorithms and tasks.

### 5.1 Experimental setup

**Inverse algorithms and hyperparameters.** We consider nine mainstream diffusion-based inverse algorithms with distinct formulations and design principles, including DDRM [5], DDNM [6], ΠGDM [7], DMPS [10], DPS [8], DiffPIR [9], ReSample [12], RED-diff [11], and DAPS [14]. All these algorithms are decomposed following the canonical form described in Section 4.1, with the details provided in Appendix E. We evaluate the performance of all algorithms under total steps of $S = 3, 4, 5, 7, 10, 15$. Due to space constraints, we report part of the results in the main text and defer the complete results to Appendix H. For the training of LLE, we fix $\omega = 0.1$ and utilize the schedule-free AdamW [45] to optimize the coefficients on 50 diffusion-generated samples. More training details are supplemented in Appendix G.

**Datasets and diffusion checkpoints.** Two datasets are used: CelebA-HQ [46] and FFHQ [47]. The checkpoint for CelebA-HQ is obtained from [48] and we follow [6] to use the 1k test set. The checkpoint for FFHQ is obtained from [8], and the evaluation follows [12, 14] with 100 randomly selected images from the validation set.

**Inverse problems and metrics.** We consider five inverse problems in this work: (1) gaussian deblurring (anisotropic), (2) inpainting (50% random), (3) $4\times$ super-resolution (average pooling), (4) compressed sensing (CS) with the Walsh-Hadamard transform (50% ratio), and (5) nonlinear deblurring. Except for nonlinear deblurring, all other four tasks are linear and we apply the decoupled coefficients method proposed in Section 4.3. The primary metrics include peak signal-to-noise ratio (PSNR), structural similarity index measure (SSIM) [49], and Learned Perceptual Image Patch Similarity (LPIPS) [44]. Results of Fréchet Inception Distance (FID) [50] are also included in the Appendix. All of the experiments are performed on a single NVIDIA RTX 3090 GPU.

### 5.2 Main results

Table 1 presents the performance of five algorithms on four noisy linear tasks from the CelebA-HQ dataset with 3 and 5 steps. Table 2 shows the results of nonlinear deblurring task on CelebA-HQ and FFHQ under both noiseless and noisy settings with 3 and 10 steps. The LLE method consistently improves the performance across different tasks and algorithms.

Figure 2 illustrates the PSNR and LPIPS metrics for four algorithms on linear tasks from FFHQ with steps varying from 3 to 15. LLE consistently achieves superior objective and perceptual metrics

Table 2: Nonlinear deblurring results of DPS, RED-diff, DiffPIR, ReSample, and DAPS using 3 and 10 steps. The better results are bolded.

| Steps | Algorithm | Strategy | PSNR↑ / SSIM↑ / LPIPS↓ | | | |
|---|---|---|---|---|---|---|
| | | | FFHQ | | CelebA-HQ | |
| | | | $\sigma_{\mathbf{y}} = 0$ | $\sigma_{\mathbf{y}} = 0.05$ | $\sigma_{\mathbf{y}} = 0$ | $\sigma_{\mathbf{y}} = 0.05$ |
| 3 | DPS | - | 15.75 / 0.403 / 0.646 | 15.75 / 0.402 / 0.647 | 16.09 / 0.408 / **0.575** | 16.09 / 0.407 / **0.575** |
| | | LLE | **16.94 / 0.440 / 0.590** | **16.93 / 0.440 / 0.590** | **19.06 / 0.479** / 0.579 | **19.06 / 0.475** / 0.580 |
| | RED-diff | - | 15.96 / 0.202 / 0.766 | 15.86 / 0.189 / 0.768 | 18.93 / 0.384 / 0.651 | 18.81 / 0.351 / 0.653 |
| | | LLE | **16.71 / 0.241 / 0.744** | **16.61 / 0.223 / 0.745** | **20.50 / 0.417 / 0.610** | **20.31 / 0.381 / 0.610** |
| | DiffPIR | - | 20.16 / 0.358 / 0.637 | **19.28 / 0.300** / 0.649 | 24.05 / 0.548 / 0.473 | 21.69 / 0.393 / 0.551 |
| | | LLE | **20.56 / 0.436 / 0.595** | 19.13 / **0.300** / 0.646 | **24.73 / 0.633 / 0.428** | **21.70 / 0.402 / 0.546** |
| | ReSample | - | 20.11 / 0.419 / 0.607 | 18.64 / 0.260 / 0.675 | 23.37 / 0.536 / 0.491 | 20.75 / 0.336 / 0.584 |
| | | LLE | **20.33 / 0.426 / 0.604** | **18.86 / 0.271 / 0.673** | **24.40 / 0.598 / 0.457** | **21.03 / 0.353 / 0.578** |
| | DAPS | - | 20.10 / 0.413 / 0.590 | 18.97 / 0.285 / 0.642 | 25.37 / 0.651 / 0.397 | **22.60** / 0.422 / **0.509** |
| | | LLE | **20.21 / 0.424 / 0.581** | **19.10 / 0.291 / 0.637** | **25.68 / 0.677 / 0.382** | 22.55 / **0.423** / 0.516 |
| 10 | DPS | - | 20.39 / 0.542 / 0.457 | 20.30 / 0.536 / 0.463 | 23.76 / 0.675 / 0.351 | 23.74 / 0.673 / 0.350 |
| | | LLE | **21.18 / 0.581 / 0.401** | **21.18 / 0.580 / 0.401** | **24.49 / 0.702 / 0.302** | **24.47 / 0.702 / 0.302** |
| | RED-diff | - | 19.41 / 0.367 / 0.658 | 19.26 / 0.330 / 0.657 | 22.13 / 0.522 / 0.535 | 21.85 / 0.462 / 0.542 |
| | | LLE | **20.85 / 0.484 / 0.602** | **20.89 / 0.491 / 0.578** | **23.22 / 0.627 / 0.479** | **23.77 / 0.567 / 0.479** |
| | DiffPIR | - | 22.97 / 0.481 / 0.548 | 23.53 / 0.552 / 0.421 | 27.53 / 0.672 / 0.386 | 27.14 / 0.663 / 0.345 |
| | | LLE | **24.27 / 0.649 / 0.442** | **24.58 / 0.571 / 0.410** | **29.98 / 0.835 / 0.282** | **27.44 / 0.695 / 0.327** |
| | ReSample | - | 25.35 / 0.699 / 0.381 | 21.61 / 0.348 / 0.567 | 29.89 / 0.833 / 0.237 | 23.06 / 0.407 / 0.507 |
| | | LLE | **25.59 / 0.719 / 0.378** | **22.30 / 0.392 / 0.550** | **30.73 / 0.861 / 0.221** | **24.06 / 0.457 / 0.471** |
| | DAPS | - | 25.56 / 0.699 / 0.345 | 23.35 / **0.453** / 0.447 | 29.54 / 0.810 / 0.240 | **25.71** / 0.544 / 0.364 |
| | | LLE | **25.80 / 0.731 / 0.343** | **23.37 / 0.453 / 0.446** | **30.55 / 0.849 / 0.223** | **25.71** / **0.547 / 0.357** |

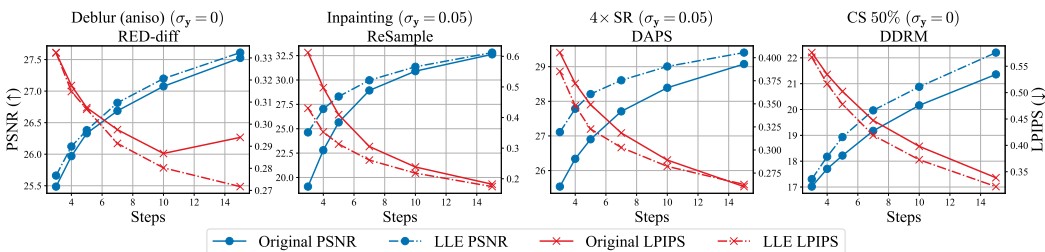

Figure 2: PSNR and LPIPS metrics under various steps for four algorithms on the FFHQ linear inverse problems.

under different steps. Some qualitative results are provided in Figure 3, and more visualization of the results is postponed to Appendix I.

A notable observation is that LLE tends to show more significant improvements when the original algorithm performs suboptimally, while achieving comparable or slightly better results when the original algorithm already performs satisfactorily. This aligns with our expectations since LLE searches for better solutions in a linear subspace spanned by previous steps. When an inverse algorithm's trajectory is close to optimal, LLE is expected to approximate this original trajectory.

### 5.3 Ablation studies

In this section, we present the ablation studies of the decoupled coefficients for linear problems and the effect of different $\omega$ in LLE. We defer more ablation studies and analysis of the results to Appendix D, including the visualizations of the learned coefficients, the training and inference time of LLE, as well as the cross-dataset and cross-task generalization results.

**Decoupled coefficients for linear observations.** Table 3 compares the performance of LLE on linear tasks using decoupled coefficients versus a single set of coefficients. All the algorithm uses LLE and 3 steps. Decoupled coefficients consistently achieve better results, validating the advantage of handling the null-space and range-space components separately.

Table 3: Ablation of the decoupled coefficients.

| Algorithm | Task | Decoupled | PSNR↑ / SSIM↑ / LPIPS↓ |
|---|---|---|---|
| DiffPIR | Deblur (aniso) ($\sigma_{\mathbf{y}} = 0$) | × | 32.81 / 0.887 / 0.194 |
| | | ✓ | **33.05 / 0.893 / 0.189** |
| DDNM | Inpainting ($\sigma_{\mathbf{y}} = 0.05$) | × | 20.74 / 0.367 / 0.542 |
| | | ✓ | **22.56 / 0.424 / 0.501** |
| DDRM | 4× SR ($\sigma_{\mathbf{y}} = 0.05$) | × | 25.64 / 0.703 / 0.342 |
| | | ✓ | **25.90 / 0.736 / 0.311** |
| ΠGDM | CS 50% ($\sigma_{\mathbf{y}} = 0$) | × | 18.39 / 0.553 / 0.536 |
| | | ✓ | **18.47 / 0.567 / 0.523** |

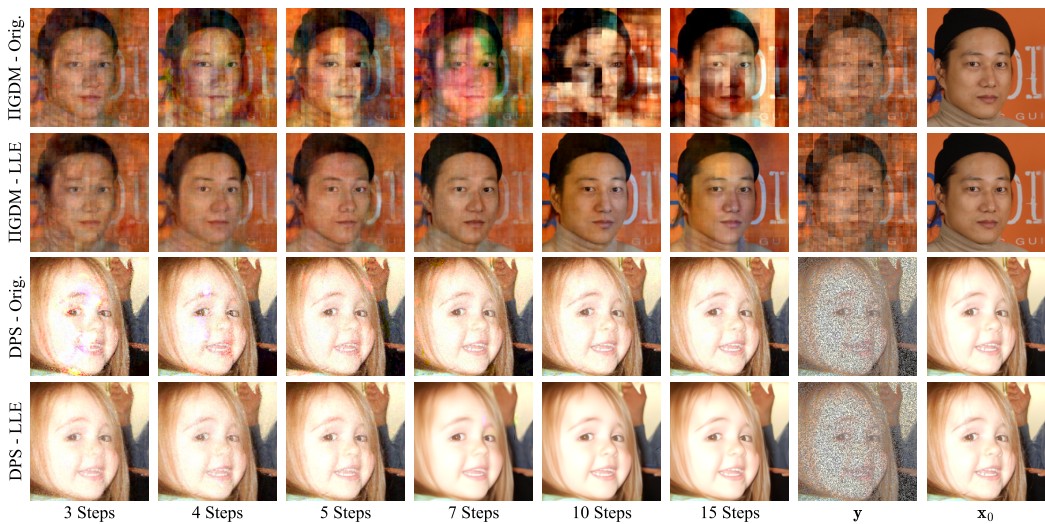

Figure 3: Qualitative comparison of restoration results with and without LLE on the FFHQ dataset. The top two rows present the results of ΠGDM on the noiseless compressed sensing task, while the bottom two rows present the results of DPS on the noiseless inpainting task.

**Effect of $\omega$.** Figure 4 presents the PSNR and LPIPS metrics of the LLE method when varying the hyperparameter $\omega$ in the range of 0 to 0.5. Both tasks are noiseless and the algorithms use 3 steps. Existing works [11, 13] have observed that objective metrics (PSNR) and perceptual quality (LPIPS) often trade off against each other in certain tasks. Figure 4 indicates that LLE provides a straightforward mechanism to control this balance by adjusting $\omega$.

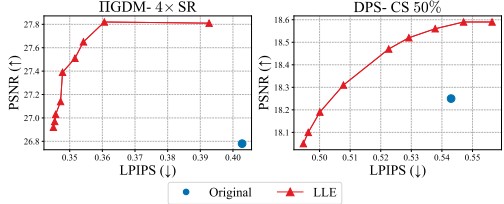

Figure 4: PSNR v.s. LPIPS for varying $\omega$ in LLE (from bottom-left to top-right: $\omega = 0.5, 0.4, 0.3, 0.2, 0.1, 0.075, 0.05, 0.025, 0.0$).

## 6 Conclusion

In this paper, we design a unified canonical form for existing diffusion-based inverse algorithms and propose the LLE method to enhance the performance of these algorithms under few steps. Extensive experiments validate the effectiveness of LLE across nine different inverse algorithms and five tasks on two datasets. LLE learns improved solvers for diffusion-based inverse algorithms in a lightweight approach, providing novel insights for the design and acceleration of diffusion-based inverse algorithms in the future.

**Limitations and Future Work.** One limitation of the LLE method is that its search space is confined to the linear subspace spanned by all previous steps. Future work could explore the introduction of nonlinear methods to further improve the algorithm's performance while maintaining lightweight and universal as LLE. Additionally, the canonical form proposed in this paper offers a flexible framework for combining and designing inverse algorithms. Research could focus on leveraging this framework to design more effective algorithms in the future. Moreover, we note that the CelebA-HQ and FFHQ datasets used in this paper have demographic imbalances, which may cause our results to fail to generalize to underrepresented groups. A more diverse and balanced dataset is needed in the field to ensure fairness across factors such as ethnicity, gender, and age. A more representative diffusion model trained on such data is also preferred to better support real-world applications.

## Acknowledgement

Jiawei Zhang, Ziyuan Liu, Leon Yan, and Yuantao Gu are supported by NSAF (Grant No. U2230201) and a Grant from the Guoqiang Institute, Tsinghua University. The authors would also like to express

sincere gratitude to Zhenwei Zhang, Cheng Jin, and Wenyu Wang for their invaluable feedback and suggestions, which significantly improved the quality of this paper. Jiawei Zhang would like to extend special thanks to Jiaxin Zhuang for his continuous efforts in maintaining the availability of our computational resources. His dedication was crucial in ensuring the successful completion of all experiments. This work would not have been published without their support and contributions.

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

# Appendix

In the appendix, we provide additional discussions and details that are omitted in the main paper due to space limitations. Appendix A reviews existing methods for solving inverse problems with diffusion models under few steps, highlighting how our work is complementary to these approaches and has the potential to be combined with them. We also discuss the relation of LLE to non-stationary solvers and its possible synergy with learnable timesteps. Appendix B presents the derivations of the probability flow ode for posterior sampling, the main results shown in the main text, and also the analysis of solvers for inverse algorithms in the SDE form. Appendix C provides additional experiments on the ImageNet dataset as well as additional nine baselines, further demonstrating the generality of LLE on more challenging tasks as well as its applicability to latent-space and consistency model-based methods. Appendix D supplements more analysis and ablation studies omitted in the main text due to space constraints. Appendix E elaborates on the detailed decomposition under the proposed canonical form of all inverse algorithms adopted in our experiments. Appendix F introduces our tailored design of LLE for DDRM and DDNM on noisy linear problems, which is specifically adapted to better align with both algorithms. Appendix G describes additional experimental details, including inverse problem settings, the hyperparameter configurations of the inverse algorithms, and the optimization methods for LLE training. Appendix H provides comprehensive results across CelebA-HQ and FFHQ datasets using 3, 4, 5, 7, 10, and 15 steps, covering four or five tasks (depending on the algorithm's ability to handle nonlinear problems) in both noiseless and noisy scenarios. Appendix I presents additional qualitative results.

## A More related works

### A.1 Other methods for few-step diffusion-based inverse algorithms

Recently, several works [19, 20, 21, 22] have attempted to design algorithms capable of solving inverse problems with diffusion models under few steps. We provide a detailed explanation of these methods and describe their differences from this work.

Chung *et al.*[19] point out that it is unnecessary to sample from pure Gaussian noise for solving inverse problems. Instead, starting from a lower noise level is more efficient by initializing with a noised version of the degraded image. Liu *et al.*[20] adopt a similar idea and refines it by using DDIM inversion to obtain the noisy image from the degraded observation as initialization. Both methods can essentially be seen as utilizing only a short segment of the diffusion sampling chain close to the clean image rather than the full sampling chain. In our work, we default to the use of the complete sampling chain for inverse algorithms. The proposed LLE method can also be applied to these methods, which only requires adjusting the sampling interval of timesteps $t_S, \ldots, t_0$ and using the corresponding initialization methods. Moreover, it can be verified that the algorithms in [19] and [20] can both be decomposed into our proposed canonical form.

Chen *et al.*[21] introduce deep data consistency (DDC), which employs an additional UNet to learn the data consistency process. We note that this is equivalent to using a UNet as the Corrector in our canonical form, making LLE directly compatible with DDC. Chen *et al.*[21] also observe that existing methods essentially are equivalent to imposing constraints on $\mathbf{x}_{0,t}$. While [21] primarily focuses on the data consistency step, our proposed canonical form systematically modularizes the diffusion model-based inverse algorithms into three components, providing a more comprehensive framework that fits well with our LLE method.

Pandey *et al.*[22] propose enhancing the efficiency of inverse problem solving by integrating the Conjugate Integrators method from diffusion sampling. They design a specialized Conjugate Integrator for $\Pi$GDM, named C-$\Pi$GDM, which transforms the solving process into a space that is more flexible and more friendly to inverse problems, thereby reducing the solver's dependency on small step sizes. The solver in C-$\Pi$GDM remains first-order, and the concept of LLE can still be applied to introduce higher-order information to further improve its performance under few steps. Additionally, the observation-based correction step in C-$\Pi$GDM can also been transformed to process in the original space by using projections between the original and transformed spaces, making it well-suited for our canonical form and LLE method.

In summary, existing few-step diffusion-based inverse algorithms are complementary to the proposed LLE method, and all fit within our canonical form. Combining LLE with these approaches has the

potential to further enhance the performance of inverse algorithms under few steps, and we leave the exploration of these combinations for future work.

## A.2 Relation to non-stationary solvers

Recently, [51] formulates non-stationary solvers as a general family of iterative schemes where the update at each step is expressed as a linear combination of all previous estimates and their corresponding velocity fields (or equivalently, score functions). Let $\mathbf{x}_{t_i}$ denote the current estimate at timestep $t_i$ and with the predicted velocity $\mathbf{v}_{t_i}$, then a non-stationary solver can be written as

$$\mathbf{x}_{t_{i-1}} = \sum_{j=i}^{S} a_{t_i,j}\mathbf{x}_{t_j} + \sum_{j=i}^{S} b_{t_i,j}\mathbf{v}_{t_j}, \tag{23}$$

where the coefficients $\{a_{t_i,j}, b_{t_i,j}\}$ are adaptively determined and can vary across timesteps. From this perspective, our proposed LLE can be regarded as a data-driven realization of non-stationary solvers tailored for diffusion-based inverse problems. By learning the combination weights directly from data, LLE adapts the solver dynamics to the empirical characteristics of the ideal trajectory.

## A.3 Combination with learnable timesteps

Recent works [52, 53, 54, 55] have proposed to improve the performance of diffusion solvers by learning the timestep discretization. Specifically, the timesteps can be parameterized and optimized under the supervision of a teacher solver's trajectory to obtain an optimal scheduler. We note that this approach and our proposed LLE represent two complementary perspectives for enhancing few-step solvers. Incorporating a learnable scheduler into the LLE framework could potentially lead to further performance gains for diffusion-based inverse algorithms in the few-step regime. We leave this as a promising direction for future work.

# B Derivations of the main results

## B.1 Probability flow ODE for posterior sampling

The derivation is similar to the probability flow ODE for diffusion sampling without observation in [56]. Recall the reverse time SDE for posterior sampling

$$d\mathbf{x}_t = \left(f(t)\mathbf{x}_t - g^2(t)\nabla_{\mathbf{x}_t}\log p(\mathbf{x}_t|\mathbf{y}, t)\right)dt + g(t)d\overline{\mathbf{w}}_t, \quad \mathbf{x}_T \sim p(\mathbf{x}_T|\mathbf{y}, T), \tag{24}$$

where $d\overline{\mathbf{w}}_t$ denotes the reverse-time Wiener process. Denote $\tau = T - t$, then we can rewrite (24) as

$$d\mathbf{x}_\tau = -\left(f(\tau) - g^2(\tau)\nabla_{\mathbf{x}_\tau}\log p(\mathbf{x}_\tau|\mathbf{y}, \tau)\right)d\tau + g(\tau)d\mathbf{w}_\tau. \tag{25}$$

Its corresponding Fokker-Planck equation follows

$$\frac{\partial p(\mathbf{x}|\mathbf{y}, \tau)}{\partial \tau} = \nabla_{\mathbf{x}} \cdot \left(\left(f(\tau)\mathbf{x} - g^2(\tau)\nabla_{\mathbf{x}}\log p(\mathbf{x}|\mathbf{y}, \tau)\right)p(\mathbf{x}|\mathbf{y}, \tau)\right) + \frac{1}{2}\nabla_{\mathbf{x}} \cdot \left(g^2(\tau)\nabla_{\mathbf{x}}p(\mathbf{x}|\mathbf{y}, \tau)\right), \tag{26}$$

with the fact that the marginal distribution of $\mathbf{x}$ at time $\tau$ is $p(\mathbf{x}|\mathbf{y}, \tau)$. Note that

$$\nabla_{\mathbf{x}}p(\mathbf{x}|\mathbf{y}, \tau) = p(\mathbf{x}|\mathbf{y}, t)\nabla_{\mathbf{x}}\log p(\mathbf{x}|\mathbf{y}, \tau). \tag{27}$$

Thus (26) becomes

$$\frac{\partial p(\mathbf{x}|\mathbf{y}, \tau)}{\partial \tau} = \nabla_{\mathbf{x}} \cdot \left(f(\tau)\mathbf{x} - \frac{1}{2}g^2(\tau)\nabla_{\mathbf{x}}\log p(\mathbf{x}|\mathbf{y}, \tau)\right), \tag{28}$$

which is exactly the Liouville equation for reverse-time probability flow ode

$$d\mathbf{x} = \left(f(t)\mathbf{x} - \frac{1}{2}g^2(t)\nabla_{\mathbf{x}}\log p(\mathbf{x}|\mathbf{y}, t)\right)dt. \tag{29}$$

## B.2 Derivations of Proposition 1

By definition, we have

$$\nabla_{\mathbf{x}_t} \log p(\mathbf{x}_t|\mathbf{y}) = \frac{\sqrt{\overline{\alpha}_t}\mathbf{x}_0(t,\mathbf{y}) - \mathbf{x}_t}{1 - \overline{\alpha}_t}. \tag{30}$$

Thus (6) can be rewritten as

$$\frac{d\mathbf{x}_t}{dt} = \left(f(t) + \frac{1}{2(1-\overline{\alpha}_t)}g^2(t)\right)\mathbf{x}_t - \frac{1}{2}g^2(t)\frac{\sqrt{\overline{\alpha}_t}}{1-\overline{\alpha}_t}\mathbf{x}_0(t,\mathbf{y}). \tag{31}$$

Given $\mathbf{x}_{t_{i+1}}$, the solution to the semilinear ODE (31) is known to be [57]

$$\mathbf{x}_{t_i} = \zeta(t_i, t_{i+1})\mathbf{x}_{t_{i+1}} - \int_{t_{i+1}}^{t_i} \zeta(t_i, \tau)\frac{1}{2}g^2(\tau)\frac{\sqrt{\overline{\alpha}_\tau}}{1-\overline{\alpha}_\tau}\mathbf{x}_0(\tau,\mathbf{y})d\tau. \tag{32}$$

where

$$\zeta(t,s) = \exp\left(\int_s^t \left(f(\tau) + \frac{1}{2(1-\overline{\alpha}_\tau)}g^2(\tau)\right)d\tau\right). \tag{33}$$

Note that

$$\int_s^t \left(f(\tau) + \frac{1}{2(1-\overline{\alpha}_\tau)}g^2(\tau)\right)d\tau = \int_s^t \frac{1}{2(1-\overline{\alpha}_\tau)}\frac{d(1-\overline{\alpha}_\tau)\tau}{d\tau}d\tau = \log\frac{\sqrt{1-\overline{\alpha}_t}}{\sqrt{1-\overline{\alpha}_s}}. \tag{34}$$

Thus (32) is

$$\sqrt{\overline{\alpha}_{t_i}}\mathbf{x}_0(t_i,\mathbf{y}) - (1-\overline{\alpha}_{t_i})\nabla_{\mathbf{x}_{t_i}}\log p(\mathbf{x}_{t_i}|\mathbf{y})$$

$$= \frac{\sqrt{1-\overline{\alpha}_{t_i}}}{\sqrt{1-\overline{\alpha}_{t_{i+1}}}}\left(\sqrt{\overline{\alpha}_{t_{i+1}}}\mathbf{x}_0(t_{i+1},\mathbf{y}) - (1-\overline{\alpha}_{t_{i+1}})\nabla_{\mathbf{x}_{t_{i+1}}}\log p(\mathbf{x}_{t_{i+1}}|\mathbf{y})\right)$$

$$- \sqrt{1-\overline{\alpha}_{t_i}}\int_{t_{i+1}}^{t_i}\frac{1}{2}g^2(\tau)\frac{\sqrt{\overline{\alpha}_\tau}}{(1-\overline{\alpha}_\tau)^{3/2}}\mathbf{x}_0(\tau,\mathbf{y})d\tau. \tag{35}$$

Rearrange (35) yields

$$\mathbf{x}_0(t_i,\mathbf{y}) = \frac{\sqrt{1-\overline{\alpha}_{t_i}}}{\sqrt{\overline{\alpha}_{t_i}}}\left(\sqrt{1-\overline{\alpha}_{t_i}}\nabla_{\mathbf{x}_{t_i}}\log p(\mathbf{x}_{t_i}|\mathbf{y}) - \sqrt{1-\overline{\alpha}_{t_{i+1}}}\nabla_{\mathbf{x}_{t_{i+1}}}\log p(\mathbf{x}_{t_{i+1}}|\mathbf{y})\right) \tag{36}$$

$$+ \frac{\sqrt{1-\overline{\alpha}_{t_i}}}{\sqrt{1-\overline{\alpha}_{t_{i+1}}}}\frac{\sqrt{\overline{\alpha}_{t_{i+1}}}}{\sqrt{\overline{\alpha}_t}}\mathbf{x}_0(t_{i+1},\mathbf{y}) - \frac{\sqrt{1-\overline{\alpha}_{t_i}}}{\sqrt{\overline{\alpha}_{t_i}}}\int_{t_{i+1}}^{t_i}\frac{1}{2}g^2(\tau)\frac{\sqrt{\overline{\alpha}_\tau}}{(1-\overline{\alpha}_\tau)^{3/2}}\mathbf{x}_0(\tau,\mathbf{y})d\tau.$$

## B.3 The case with SDE

Empirical evidence has shown that the injection of stochastic noise plays a critical role in the effectiveness of diffusion-based inverse algorithms. Accordingly, we extend our discussion to cover the SDE-based formulation in this section.

Considering the SDE for posterior sampling (24), we may follow [24] to solve the SDE as

$$\mathbf{x}_{t_i} = \xi(t_i, t_{i+1})\mathbf{x}_{t_{i+1}} - \int_{t_{i+1}}^{t_i}\xi(t_i,\tau)g^2(\tau)\frac{\sqrt{\overline{\alpha}_\tau}}{1-\overline{\alpha}_\tau}\mathbf{x}_0(\tau,\mathbf{y})d\tau + \int_{t_{i+1}}^{t_i}\xi(t_i,\tau)g(\tau)d\overline{\mathbf{w}}_\tau, \tag{37}$$

where

$$\xi(t_i, \tau) = \exp\left(\int_\tau^{t_i}\left(f(\tau) + \frac{1}{1-\overline{\alpha}_\tau}g^2(\tau)\right)d\tau\right). \tag{38}$$

We have

$$\int_\tau^{t_i}\left(f(r) + \frac{1}{1-\overline{\alpha}_r}g^2(r)\right)dr = \int_r^{t_i}\left(\frac{1}{1-\overline{\alpha}_r}\frac{d(1-\overline{\alpha}_r)}{dr} - \frac{d\log\sqrt{\overline{\alpha}_r}}{dr}\right)dr \tag{39}$$

$$= \log\left(\frac{1-\overline{\alpha}_{t_i}}{1-\overline{\alpha}_\tau}\frac{\sqrt{\overline{\alpha}_\tau}}{\sqrt{\overline{\alpha}_{t_i}}}\right). \tag{40}$$

Then the Itô integral is computed as

$$\int_{t_{i+1}}^{t_i} \xi(t_i, \tau) g(\tau) d\overline{\mathbf{w}}_\tau = \frac{1 - \overline{\alpha}_{t_i}}{\sqrt{\overline{\alpha}_{t_i}}} \kappa(t_i, t_{i+1}) \boldsymbol{\epsilon}, \quad \boldsymbol{\epsilon} \sim \mathcal{N}(\mathbf{0}, \mathbf{I}), \tag{41}$$

where

$$
\begin{aligned}
\kappa^2(t_i, t_{i+1}) &= \int_{t_{i+1}}^{t_i} \frac{\overline{\alpha}_\tau}{(1 - \overline{\alpha}_\tau)^2} \left( \mathrm{d}(1 - \overline{\alpha}_\tau) - 2(1 - \overline{\alpha}_\tau) \mathrm{d} \log \sqrt{\overline{\alpha}_\tau} \right) \\
&= \int_{t_{i+1}}^{t_i} \left( \frac{1}{(1 - \overline{\alpha}_\tau)^2} - \frac{1}{1 - \overline{\alpha}_\tau} \right) \mathrm{d}(1 - \overline{\alpha}_\tau) - \frac{1}{1 - \overline{\alpha}_\tau} \mathrm{d}\overline{\alpha}_\tau \\
&= \frac{1}{1 - \overline{\alpha}_{t_{i+1}}} - \frac{1}{1 - \overline{\alpha}_{t_i}}.
\end{aligned}
\tag{42}
$$

Thus with (30) one have

$$
\begin{aligned}
\mathbf{x}_0(t_i, \mathbf{y}) = &\frac{1 - \overline{\alpha}_{t_i}}{1 - \overline{\alpha}_{t_{i+1}}} \frac{\overline{\alpha}_{t_{i+1}}}{\overline{\alpha}_{t_i}} \mathbf{x}_0(t_{i+1}, \mathbf{y}) \\
&- \frac{1 - \overline{\alpha}_{t_i}}{\overline{\alpha}_{t_i}} \int_{t_{i+1}}^{t_i} \eta(\tau) \mathbf{x}_0(\tau, \mathbf{y}) \mathrm{d}\tau + \frac{1 - \overline{\alpha}_{t_i}}{\overline{\alpha}_{t_i}} \underbrace{\sqrt{\frac{1}{1 - \overline{\alpha}_{t_{i+1}}} - \frac{1}{1 - \overline{\alpha}_{t_i}}} \boldsymbol{\epsilon}}_{\text{①}} \\
&+ \frac{1 - \overline{\alpha}_{t_i}}{\overline{\alpha}_{t_i}} \underbrace{\left( \sqrt{\overline{\alpha}_{t_i}} \nabla_{\mathbf{x}_{t_i}} \log p(\mathbf{x}_{t_i}|\mathbf{y}) - \sqrt{\overline{\alpha}_{t_{i+1}}} \nabla_{\mathbf{x}_{t_{i+1}}} \log p(\mathbf{x}_{t_{i+1}}|\mathbf{y}) \right)}_{\text{②}},
\end{aligned}
\tag{43}
$$

where $\eta(\tau) = \frac{\overline{\alpha}_\tau}{(1 - \overline{\alpha}_\tau)^2} g^2(\tau)$.

Similarly, the integral of $\mathbf{x}_0(\tau, \mathbf{y})$ may be approximated by the (informal) Taylor expansion, with high-order derivatives estimated from previous-step results. An estimate $\hat{\mathbf{x}}_{t_i}$ can be sampled using a standard DDPM update, i.e.,

$$\hat{\mathbf{x}}_{t_i} \sim \mathcal{N} \left( \frac{\sqrt{\overline{\alpha}_{t_i}}}{1 - \overline{\alpha}_{t_{i+1}}} \left( 1 - \frac{\overline{\alpha}_{t_{i+1}}}{\overline{\alpha}_{t_i}} \right) \mathbf{x}_0 + \frac{1 - \overline{\alpha}_{t_i}}{1 - \overline{\alpha}_{t_{i+1}}} \frac{\sqrt{\overline{\alpha}_{t_{i+1}}}}{\sqrt{\overline{\alpha}_{t_i}}} \mathbf{x}_{t_{i+1}}, \frac{1 - \overline{\alpha}_{t_i}}{1 - \overline{\alpha}_{t_{i+1}}} \left( 1 - \frac{\overline{\alpha}_{t_{i+1}}}{\overline{\alpha}_{t_i}} \right) \mathbf{I} \right). \tag{44}$$

If the sampled noise is exactly the same $\boldsymbol{\epsilon}$ in (43), then ① + ② in (43) is

$$\frac{\overline{\alpha}_{t_i}}{1 - \overline{\alpha}_{t_i}} \frac{\hat{\mathbf{x}}_{t_i} + (1 - \overline{\alpha}_{t_i}) \nabla_{\mathbf{x}_{t_i}} \log p(\mathbf{x}_{t_i}|\mathbf{y})}{\sqrt{\overline{\alpha}_{t_i}}} - \frac{\overline{\alpha}_{t_i}}{1 - \overline{\alpha}_{t_{i+1}}} \mathbf{x}_0(t_{i+1}, \mathbf{y}), \tag{45}$$

which leads to a similar conclusion as Corollary 2.

## C   Additional results and comparisons

In this section, we present additional experimental results on the ImageNet dataset, as well as the performance of LLE over nine additional baseline algorithms.

### C.1   Results on ImageNet

Here we supplement the results of ΠGDM, RED-diff, DPS, and ReSample on the ImageNet dataset [58] with 4 steps and 10 steps. Following [12, 14], we sample 100 images from the ImageNet test set for evaluation. For noisy tasks, we add Gaussian noise with $\sigma = 0.1$ with respect to data range [-1, 1]. The ADM pre-trained checkpoint in [4] is adopted. The results are shown in Table 4, which demonstrates that LLE consistently improves performance across all methods in the few-step setting on ImageNet. This confirms that LLE is effective and broadly applicable on more complex datasets.

Table 4: More results of ΠGDM, RED-diff, DPS, and ReSample on ImageNet.

| Steps | Algorithm | Strategy | PSNR↑ / SSIM ↑ / LPIPS ↓ | | | |
|---|---|---|---|---|---|---|
| | | | Deblur (noiseless) | Inpainting (noisy) | $4\times$ SR (noiseless) | CS $50\%$ (noisy) |
| 4 | ΠGDM | - | 23.56 / 0.727 / 0.294 | 18.75 / 0.408 / 0.641 | 17.41 / 0.373 / 0.557 | 14.60 / 0.217 / 0.746 |
| | | LLE | **30.14 / 0.875 / 0.196** | **19.41 / 0.435 / 0.619** | **20.53 / 0.524 / 0.531** | **16.73 / 0.359 / 0.650** |
| | RED-diff | - | 22.57 / 0.589 / 0.490 | 18.00 / 0.357 / 0.592 | 24.81 / 0.719 / 0.419 | 17.10 / 0.409 / 0.574 |
| | | LLE | **22.7 / 0.602 / 0.486** | **21.69 / 0.524 / 0.492** | **25.19 / 0.726 / 0.359** | **17.77 / 0.468 / 0.531** |
| | DPS | - | 15.90 / 0.268 / 0.681 | 22.92 / 0.672 / 0.421 | 17.10 / 0.358 / 0.567 | 15.30 / 0.300 / 0.633 |
| | | LLE | **18.42 / 0.388 / 0.622** | **24.41 / 0.690 / 0.410** | **20.33 / 0.517 / 0.533** | **18.03 / 0.474 / 0.556** |
| | ReSample | - | 25.89 / 0.746 / 0.355 | 20.49 / 0.440 / 0.521 | 23.18 / 0.651 / 0.417 | 17.50 / 0.439 / 0.540 |
| | | LLE | **26.18 / 0.757 / 0.340** | **23.28 / 0.561 / 0.443** | **24.98 / 0.712 / 0.369** | **17.80 / 0.454 / 0.529** |
| 10 | ΠGDM | - | 26.22 / 0.789 / 0.289 | 21.99 / 0.609 / 0.487 | 21.49 / 0.532 / 0.503 | 16.21 / 0.360 / 0.681 |
| | | LLE | **29.92 / 0.856 / 0.216** | **23.69 / 0.611 / 0.471** | **22.76 / 0.596 / 0.450** | **18.61 / 0.509 / 0.540** |
| | RED-diff | - | 23.41 / 0.620 / **0.458** | 22.95 / 0.539 / 0.449 | 25.75 / 0.749 / 0.349 | 18.49 / 0.466 / 0.517 |
| | | LLE | **23.42 / 0.623** / 0.462 | **25.07 / 0.669 / 0.386** | **25.92 / 0.754 / 0.309** | **19.22 / 0.513 / 0.487** |
| | DPS | - | 19.55 / 0.426 / 0.608 | 24.65 / 0.711 / 0.364 | 21.28 / 0.520 / 0.512 | 16.61 / 0.413 / 0.612 |
| | | LLE | **20.33 / 0.463 / 0.555** | **27.17 / 0.780 / 0.317** | **22.42 / 0.585 / 0.460** | **19.52 / 0.590 / 0.463** |
| | ReSample | - | 26.49 / 0.765 / 0.331 | 25.26 / 0.625 / 0.367 | 25.27 / 0.721 / 0.353 | 19.60 / 0.519 / 0.449 |
| | | LLE | **26.72 / 0.772 / 0.317** | **25.78 / 0.651 / 0.349** | **25.61 / 0.736 / 0.329** | **20.02 / 0.541 / 0.434** |

## C.2 Additional baselines

To further validate the general applicability of LLE, we include nine additional inverse algorithms as baselines, including five pixel-space methods (LGD [59], DCDP [60], MPGD [61], FPS [62], and SGS-EDM [63]), three latent-space methods (Latent-DPS [12], PSLD [64], and STSL [65]), and one consistency model-based method (CMInversion [66]). For pixel-space methods, we follow the same validation settings on FFHQ as used in the main content. For latent-space methods, we perform evaluation on the FFHQ-256 dataset using the pretrained LDM checkpoint provided by [67]. For CMInversion, we adopt the pretrained consistency model from [68], which was trained for class-conditional generation on ImageNet-64 with $\ell_2$ loss. We randomly sample 100 images from class 0 ("tench") in the ImageNet validation set and resize them to $64 \times 64$ for evaluation. The corresponding results are presented in Table 5, Table 6, and Table 7. The results indicate that LLE consistently improves the performance of all nine baselines in the few-step regime, further confirming its generality and broad applicability.

We now elaborate on how latent diffusion-based and consistency model-based inverse algorithms can be incorporated into the LLE framework. There are no conceptual difficulties in extending the canonical form and LLE from pixel space to latent space. The observation model with LDMs can be formulated as

$$\mathbf{y} = \mathcal{A}(\mathcal{D}(\mathbf{z}_0)) + \sigma \mathbf{n}, \tag{46}$$

where $\mathcal{D}(\cdot)$ denotes the decoder of the VAE, $\mathcal{A}(\cdot)$ is the observation operator, and $\mathbf{z}_0$ is the latent. This corresponds to a nonlinear inverse problem with respect to $\mathbf{z}_0$, since the composed operator $\mathcal{A} \circ \mathcal{D}$ is highly nonlinear. Thus the theoretical foundation of linear extrapolation in LLE still holds, as it does not rely on specific assumptions about the observation function. The only modification required for training LLE in the latent space is a minor adjustment to the loss function (20), which becomes

$$\mathcal{L}\left(\mathcal{D}\left(\tilde{\mathbf{z}}_{0,t_i}^{(n)}\right), \mathcal{D}\left(\mathbf{z}_0^{(n)}\right)\right), \tag{47}$$

since LPIPS is computed in the pixel space.

Regarding consistency model-based methods, though consistency models do not provide an explicit sampling trajectory, they can always be viewed as one-step efficient samplers. Inverse algorithms built upon consistency models (e.g., CMInversion [68]) also follow a sampling-correction-noising paradigm, which naturally fits into our canonical form and can therefore benefit from the LLE framework as well.

The detailed decomposition of all nine algorithms above into our canonical form is supplemented in Appendix E.

Table 5: Additional results of DCDP, LGD, MPGD, SGS-EDM, and FPS-SMC on FFHQ with 4 steps.

| Steps | Algorithm | Strategy | PSNR↑ / SSIM ↑ / LPIPS ↓ | | | |
| --- | --- | --- | --- | --- | --- | --- |
| | | | Deblur (noisy) | Inpainting (noisy) | 4× SR (noisy) | CS 50% (noisy) |
| 4 | DCDP | - | 26.00 / 0.619 / 0.428 | 24.75 / 0.558 / 0.436 | 24.16 / 0.585 / 0.578 | 17.70 / 0.461 / 0.535 |
| | | LLE | **26.12 / 0.681 / 0.378** | **25.54 / 0.596 / 0.411** | **25.24 / 0.707 / 0.350** | **18.16 / 0.471 / 0.524** |
| | LGD | - | 22.81 / 0.645 / 0.433 | 21.77 / 0.548 / 0.513 | 24.95 / 0.680 / 0.439 | 15.84 / 0.451 / 0.513 |
| | | LLE | **24.59 / 0.695 / 0.376** | **28.25 / 0.768 / 0.339** | **25.55 / 0.705 / 0.402** | **16.52 / 0.503 / 0.499** |
| | MPGD | - | 25.29 / 0.705 / 0.380 | 27.33 / 0.753 / 0.309 | 24.93 / 0.562 / 0.579 | 17.36 / 0.437 / 0.565 |
| | | LLE | **25.43** / 0.704 / **0.366** | **27.76 / 0.768 / 0.298** | **25.42 / 0.703 / 0.360** | **18.02 / 0.446 / 0.543** |
| | SGS-EDM | - | **27.72** / 0.776 / 0.308 | 19.71 / 0.416 / 0.538 | 26.53 / 0.738 / 0.337 | 17.72 / 0.482 / 0.530 |
| | | LLE | 27.67 / **0.782 / 0.307** | **24.44 / 0.511 / 0.453** | **26.95 / 0.762 / 0.309** | **18.40 / 0.516 / 0.486** |
| | FPS | - | 25.11 / 0.517 / 0.475 | 17.52 / 0.188 / 0.686 | 24.62 / 0.524 / 0.474 | 16.58 / 0.224 / 0.666 |
| | | LLE | **26.26 / 0.617 / 0.428** | **18.63 / 0.229 / 0.643** | **26.14 / 0.672 / 0.400** | **17.66 / 0.384 / 0.569** |

Table 6: Additional results of Latent-DPS, PSLD, and STSL on FFHQ with 4 steps.

| Steps | Algorithm | Strategy | PSNR ↑ / SSIM ↑ / LPIPS ↓ | | | |
| --- | --- | --- | --- | --- | --- | --- |
| | | | Deblur (noisy) | Inpainting (noisy) | 4× SR (noisy) | CS 50% (noisy) |
| 4 | Latent-DPS | - | 16.42 / 0.428 / 0.636 | 15.43 / 0.426 / **0.628** | 15.30 / 0.369 / 0.682 | 14.51 / 0.364 / 0.665 |
| | | LLE | **17.08 / 0.441 / 0.609** | **17.83 / 0.492** / 0.652 | **15.49 / 0.399 / 0.648** | **14.70 / 0.366** / 0.646 |
| | PSLD | - | 17.55 / 0.468 / 0.630 | 17.71 / 0.474 / 0.627 | 17.72 / 0.471 / 0.630 | 15.06 / **0.438** / 0.647 |
| | | LLE | **18.10 / 0.477 / 0.607** | **18.55 / 0.490 / 0.599** | **18.57 / 0.489 / 0.597** | **15.24** / 0.433 / **0.640** |
| | STSL | - | 16.08 / 0.378 / 0.655 | 16.87 / 0.404 / 0.639 | 15.70 / **0.368** / 0.663 | 15.31 / 0.383 / 0.629 |
| | | LLE | **16.97 / 0.392 / 0.616** | **17.50 / 0.407 / 0.598** | **16.27** / 0.360 / **0.628** | **15.72 / 0.384 / 0.621** |

# D   More ablation studies and additional analysis

In this section, we supplement the ablation studies and discussion that are omitted in the main text due to space constraints.

### D.1   Visualization of the LLE coefficients

Here, we present and analyze the visualization of the learned coefficients from several algorithms trained on the CelebA-HQ dataset. Figure 5, Figure 6, Figure 7, and Figure 8 correspond to DDNM on noisy inpainting, ΠGDM on noisy compressive sensing, DPS on noiseless super-resolution, and DiffPIR on noiseless anisotropic deblurring, respectively. All visualizations are based on 7-step inference. The vertical axis indicates the current step in the inverse process, and each row represents the linear combination weights applied to all previous steps. The left subfigure shows the coefficients for range space, while the right subfigure shows the coefficients for null space.

Different algorithms yield distinct patterns of the learned linear combination coefficients, which is attributed to their underlying different architectures and designs. At the same time, several shared characteristics emerge across methods, aligning well with the design philosophy of our framework.

First, we observe a clear difference between the range space and null space coefficients. This supports our hypothesis that learning separate coefficients allows the algorithm to better capture the varying strengths of the observation constraints in the range and null spaces. Notably in DDNM, the range space coefficients are almost entirely concentrated along the diagonal with values close to one, while off-diagonal elements remain near zero. This is consistent with the design philosophy of DDNM: DDNM's correction step explicitly projects the sample onto the hyperplane defined by the observation, thereby enforcing a hard constraint in the range space while preserving greater flexibility in the null space. Some non-zero off-diagonal coefficients in the range space are attributed to the observation noise. The other three methods do not employ explicit projection operations. Consequently, these methods impose relatively less stringent constraints on the range space compared to DDNM, which may result in estimation discrepancies that LLE could compensate for by learning a more effective linear combination.

Table 7: Additional results of CMInversion on ImageNet $64 \times 64$ with 4 steps.

| Steps | Algorithm | Strategy | PSNR ↑ / SSIM ↑ / LPIPS ↓ | | | |
|---|---|---|---|---|---|---|
| | | | Deblur (noisy) | Inpainting (noisy) | $4\times$ SR (noisy) | CS $50\%$ (noisy) |
| 4 | CMInversion | - | 12.69 / 0.204 / 0.613 | 12.71 / 0.269 / 0.649 | 14.65 / 0.317 / 0.559 | 14.80 / 0.242 / 0.619 |
| | | LLE | **16.50 / 0.283 / 0.522** | **17.64 / 0.383 / 0.566** | **18.57 / 0.410 / 0.471** | **16.58 / 0.268 / 0.580** |

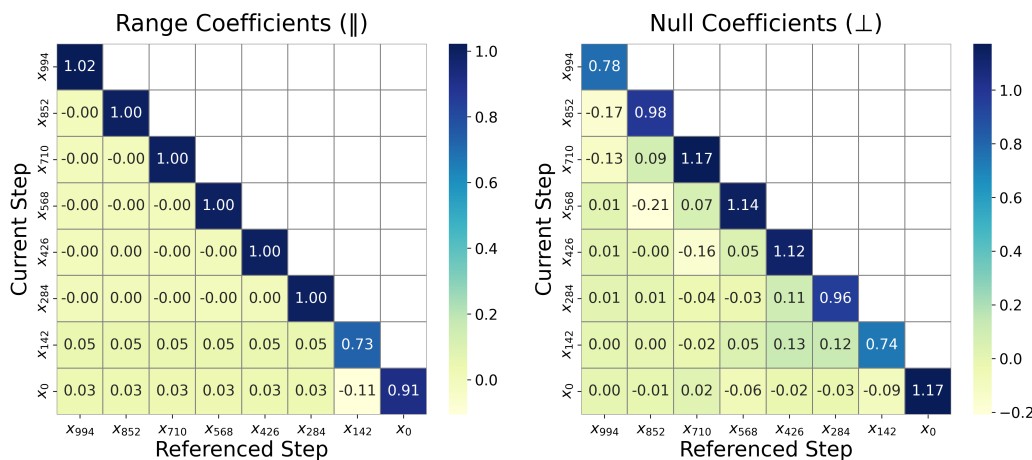

Figure 5: Visualization of learned coefficients of DDNM on the noisy inpainting task.

Second, we observe that the learned coefficients are not always dominated by diagonal elements. For instance, in $\Pi$GDM at step 770 and in DiffPIR at steps 142 and 0, the model assigns larger weights to earlier steps rather than the current one. This suggests that LLE may, in certain cases, favor earlier estimates over the most recent one. Such behavior reveals an implicit form of regularization, akin to "early stopping", within the dynamics of LLE, echoing a similar strategy reported in prior studies [69]. This demonstrates LLE's ability to dynamically assess the reliability of predictions and turn to past estimates when appropriate, thereby enhancing robustness.

Moreover, a consistent pattern across different inverse algorithms is that the LLE coefficients tend to concentrate around the diagonal and its immediate neighborhood. This suggests that more recent steps exert a stronger influence, while earlier steps contribute less as their estimates become increasingly outdated. According to Equation (10), elements farther from the diagonal correspond to higher-order derivatives of the sampling trajectory. Since most sampling trajectories are expected to be smooth, the high-order derivatives gradually decay to zero as the order grows, which results in the weight matrix being dominated by the diagonal and first off-diagonal elements. Such behavior is also intuitive, since earlier predictions are typically less accurate and thus less informative for refining later outputs. Building on this insight, one possible improvement is to introduce a limited look-back window for LLE, which could reduce both computational and memory overhead during training and inference. This modification may further improve the scalability of LLE, particularly in inverse problems involving longer sampling trajectories. We leave a deeper investigation to future work.

## D.2 Cross-task and cross-dataset generalization

Although conceptually LLE requires training with the same prior distribution and inverse task as inference, we also evaluate its generalizability across datasets and tasks. Table 8 reports LLE's performance with cross-dataset training between FFHQ and CelebA-HQ, Table 9 shows the cross-dataset results between FFHQ and ImageNet, and Table 10 presents performance with corss-task training.

We observe that LLE trained on one dataset generalizes well to another, even in the more challenging ImageNet-to-FFHQ cross-domain setting. Cross-dataset training achieves comparable performance to training on the matching dataset, and occasionally even surpasses it. This indicates that LLE possesses a certain degree of cross-dataset generalization capability.

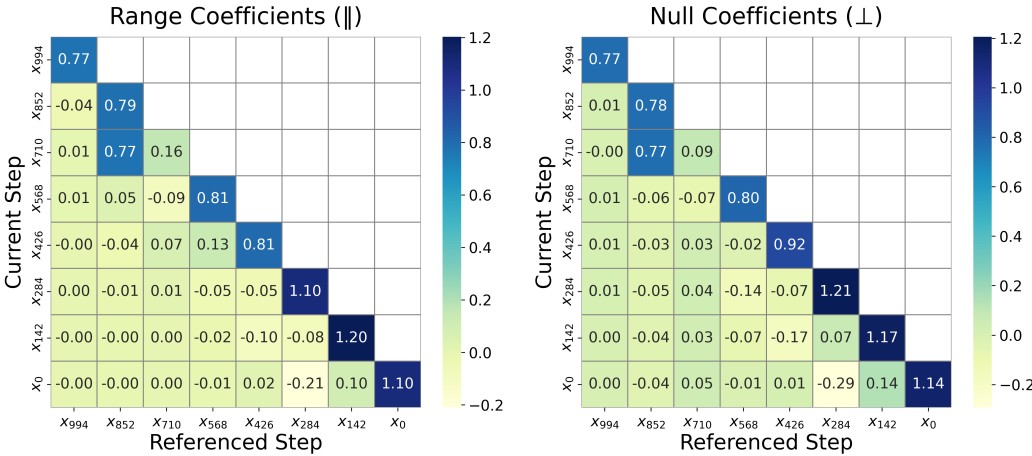

Figure 6: Visualization of learned coefficients of $\Pi$GDM on the noisy compressive sensing task.

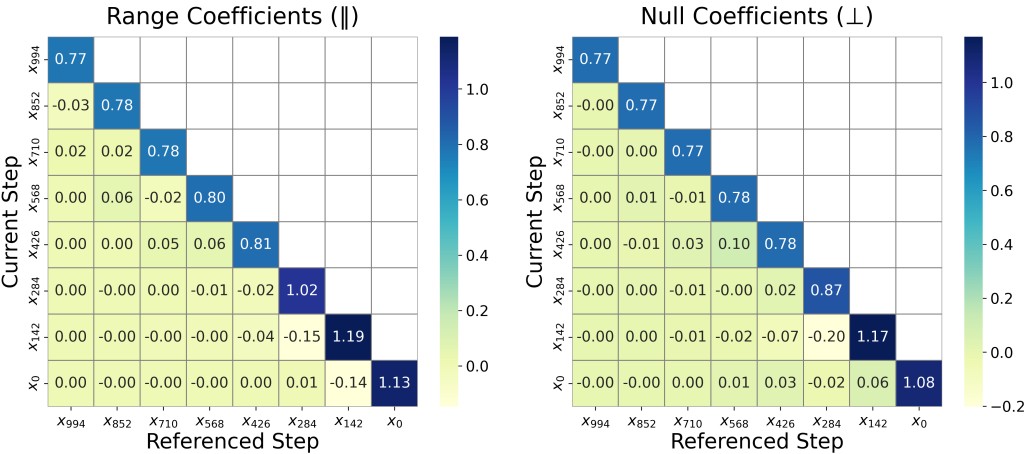

Figure 7: Visualization of learned coefficients of DPS on the noiseless super-resolution task.

When the training and testing tasks differ, LLE performance generally degrades. However, in some cases, the cross-task trained LLE coefficients still outperform the original baseline without LLE. For instance, LLE trained on super-resolution or compressive sensing can improve performance on inpainting. This suggests that certain inverse problems may share similar underlying structures, potentially leading to similar optimal coefficient patterns. Future work could further explore these shared structures to enhance LLE's generalization across tasks.

### D.3 Sensitivity to the hyperparameters of original algorithms

It is well known that some inverse algorithms require careful tuning of hyperparameters for each task and dataset, for example, the step size $\zeta$ in DPS and the regularization weight $\lambda$ in RED-diff. Small perturbations in these hyperparameters can often lead to significant performance degradation.

We note that the linear combination mechanism in LLE can be interpreted as equivalently involving a learnable scaling factor, which may reduce the sensitivity of the original algorithm to its hyperparameters. We conduct an experiment analyzing the performance of DPS and RED-diff with and without LLE under varying hyperparameter settings on the noiseless compressed sensing task from FFHQ, with hyperparameter $\zeta$ or $\lambda$ ranging from $0.2$ to $0.8$. The performance is shown in Figure 9. The results demonstrate that LLE mitigates the influence of hyperparameter variations and consistently achieves better performance, indicating that LLE can potentially reduce the reliance on precise hyperparameter tuning.

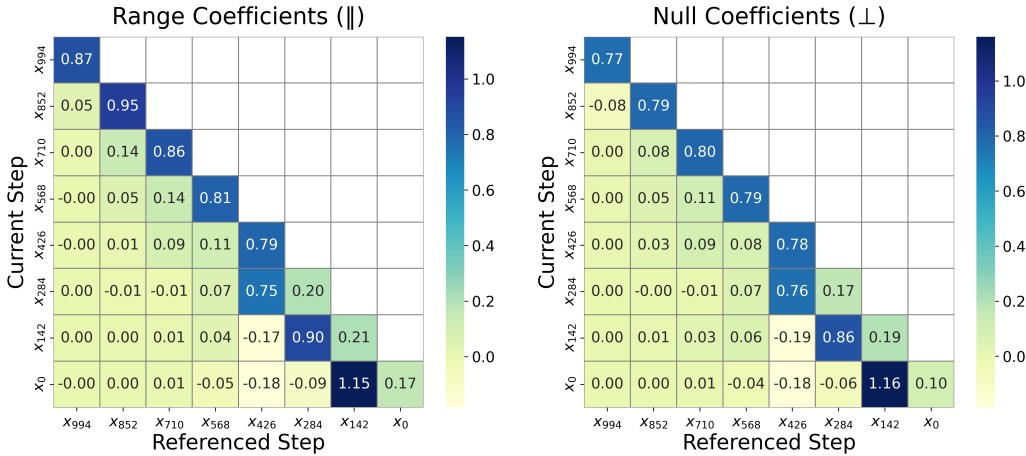

Figure 8: Visualization of learned coefficients of DiffPIR on the noiseless anisotropic deblurring task.

Table 8: Cross-dataset generalization results of LLE between CelebA-HQ and FFHQ on the noiseless tasks. *Testset* denotes the dataset used for inference, while *Trainset* indicates the dataset used to train the LLE coefficients. We also include results without LLE for comparison, denoted by "-" in the *Trainset* column.

| Testset | Trainset | PSNR↑ / SSIM ↑ / LPIPS ↓ | | | |
|---------|----------|------------------|-----------|--------|--------|
| | | Deblur (aniso) | Inpainting | 4× SR | CS 50% |
| CelebA-HQ | - | 39.99 / 0.962 / 0.090 | 27.83 / 0.758 / 0.332 | **30.97 / 0.877**/ 0.188 | 18.98 / 0.619 / 0.450 |
| | CelebA-HQ | 40.20 / 0.964 / 0.087 | **30.93 / 0.887 / 0.221** | 30.90 / 0.874 / **0.182** | **20.15** / 0.640 / 0.415 |
| | FFHQ | **40.28 / 0.965 / 0.086** | 30.70 / 0.883 / 0.226 | 30.89 / 0.874 / **0.182** | 20.13 / **0.643 / 0.413** |
| FFHQ | - | 39.82 / 0.964 / 0.087 | 26.34 / 0.719 / 0.380 | **29.99 / 0.865** / 0.210 | 18.29 / 0.591 / 0.488 |
| | FFHQ | **40.08 / 0.966 / 0.084** | 28.71 / **0.838 / 0.292** | 29.94 / 0.863 / **0.206** | **19.21 / 0.605 / 0.463** |
| | CelebA-HQ | 40.00 / 0.965 / 0.085 | **28.84 / 0.838 / 0.292** | 29.92 / 0.863 / **0.206** | 19.19 / 0.599 / 0.466 |

## D.4   Training cost, inference time, and momery footprint

Table 11 presents the training time of LLE for DDNM, DPS, DiffPIR, and DAPS on noiseless inpainting tasks from CelebA-HQ. LLE requires only 2 to 20 minutes for training, depending on the task, the complexity of the original algorithm, and the number of steps. This demonstrates the lightweight advantage of our LLE method.

We also compare the inference time of each algorithm with and without the LLE module. Table 12 presents the inference time on the CelebA-HQ 1k test set for the inpainting task. The results indicates that incorporating LLE introduces negligible additional computational overhead during inference. The efficiency of LLE stems from the simplicity of its operation, which only involves computing a linear combination of previous samples without introducing any complex transformations.

Regarding the memory footprint, here we further report the GPU memory usage during inference (in MB) of DDNM and DPS with and without LLE in Table 13. We measure the CUDA peak allocated memory, and we account only for the memory used by model parameters, data, and additional storage required by the algorithm. The results show that LLE introduces only a few to several dozen MB of additional memory overhead, even under the 15-step setting. This is consistent with our expectations: during inference, LLE only needs to store a few $256 \times 256$ RGB images from previous steps, typically around 0.7 MB per image, which is negligible compared to model weights or other algorithm-specific overhead (e.g., back-propagation in DPS).

## D.5   Analysis with default steps

Though our primary focus is on performance under limited steps, we also present the 100-step results of DDNM on CelebA-HQ in Table 14 as a reference. These results further indicate that LLE

Table 9: Cross-dataset generalization results of LLE between FFHQ and ImageNet on noiseless tasks. *Train task* denotes the task used to train the LLE coefficients. We also include results without LLE for comparison, indicated by "-" in the *Train task* column.

| Testset | Trainset | PSNR↑ / SSIM ↑ / LPIPS ↓ | | | |
| --- | --- | --- | --- | --- | --- |
| | | Deblur (aniso) | Inpainting | 4× SR | CS 50% |
| FFHQ | - | 39.82 / 0.964 / 0.087 | 26.34 / 0.719 / 0.380 | **29.99 / 0.865 / 0.210** | 18.29 / 0.591 / 0.488 |
| | FFHQ | **40.08 / 0.966 / 0.084** | **28.71 / 0.838 / 0.292** | 29.94 / 0.863 / 0.206 | **19.21 / 0.605 / 0.463** |
| | ImageNet | 39.94 / 0.965 / 0.086 | 28.30 / 0.818 / 0.317 | 29.94 / 0.863 / 0.208 | 19.01 / 0.608 / 0.470 |
| ImageNet | - | 36.48 / 0.949 / 0.099 | 23.35 / 0.628 / 0.416 | 25.74 / 0.754 / 0.302 | 17.86 / 0.535 / 0.486 |
| | ImageNet | 36.52 / 0.949 / 0.098 | 24.87 / 0.726 / 0.364 | **25.76 / 0.752 / 0.297** | **18.35 / 0.541 / 0.478** |
| | FFHQ | **36.53 / 0.949 / 0.099** | **25.14 / 0.759 / 0.341** | 25.70 / 0.750 / 0.296 | 18.20 / 0.527 / 0.485 |

Table 10: Cross-task generalization results of LLE on noiseless tasks from FFHQ. *Train task* denotes the task used to train the LLE coefficients. We also include results without LLE for comparison, indicated by "-" in the *Train task* column.

| Train task | PSNR↑ / SSIM ↑ / LPIPS ↓ | | | |
| --- | --- | --- | --- | --- |
| | Deblur (aniso) | Inpainting | 4× SR | CS 50% |
| Deblur | **40.08 / 0.966 / 0.084** | 24.53 / 0.655 / 0.432 | 29.90 / 0.862 / 0.231 | 18.06 / 0.578 / 0.509 |
| Inpainting | 39.04 / 0.960 / 0.094 | **28.71 / 0.838 / 0.292** | 29.56 / 0.854 / 0.217 | 19.07 / 0.602 / 0.467 |
| 4× SR | 39.32 / 0.959 / 0.095 | 27.70 / 0.774 / 0.339 | 29.94 / 0.863 / **0.206** | 18.73 / 0.602 / 0.475 |
| CS 50% | 38.89 / 0.957 / 0.098 | 27.90 / 0.799 / 0.324 | 29.46 / 0.854 / 0.213 | **19.21 / 0.605 / 0.463** |
| - | 39.82 / 0.964 / 0.087 | 26.34 / 0.719 / 0.380 | **29.99 / 0.865** / 0.210 | 18.29 / 0.591 / 0.488 |

is particularly effective when there is a large performance gap between the few-step setting and the optimal setting. For example, in the inpainting and compressive sensing tasks, the difference between the few-step performance and the default-step (100 steps) performance is substantial, thus LLE provides significant improvements, with PSNR gains ranging from 1dB to 8dB. In contrast, for deblurring and super-resolution tasks, the performance gap between few steps and 100 steps is relatively limited, and correspondingly, LLE yields smaller improvements. This is expected as LLE is constrained to searching within the linear combinations of previous predictions; thus, when the few-step trajectory already approximates the optimal performance, LLE tends to learn a trajectory similar to the original one.

### D.6 Performance of LLE with more steps

Here we supplement an additional experiment comparing DDNM with and without LLE under a larger number of sampling steps (25, 50, and 100) on the FFHQ dataset, as presented in Table 15. To avoid the need to store an excessive number of past results when the number of steps is large, we limit the lookback window to 10 steps. The results show that as the number of steps increases, the performance gap between methods with and without LLE becomes smaller. This is consistent with our expectation as LLE searches for an improved sampling trajectory within the linear subspace spanned by previous samples. When the original algorithm takes more steps, the discretization error is reduced, and its trajectory already closely approximates the optimal one. In such cases, LLE tends to follow the original trajectory more closely, resulting in smaller gaps. Therefore, LLE brings more significant improvements in few-step regimes, which aligns with our primary objective of enhancing the performance of inverse algorithms under a limited number of steps.

### D.7 Effect of Optimization Iterations on LLE Performance

To assess how sensitive LLE is to the optimization of Equations (19), we further investigate its convergence behavior. We optimize LLE for $\{25, 50, 100, 150, 200\}$ iterations and report the corresponding performance on the FFHQ noiseless task using the 5-step DDNM. The results are summarized in Table 16. We find that LLE achieves consistently strong and comparable results across a wide range of iterations, indicating that it converges quickly and shows no significant overfitting even with extended

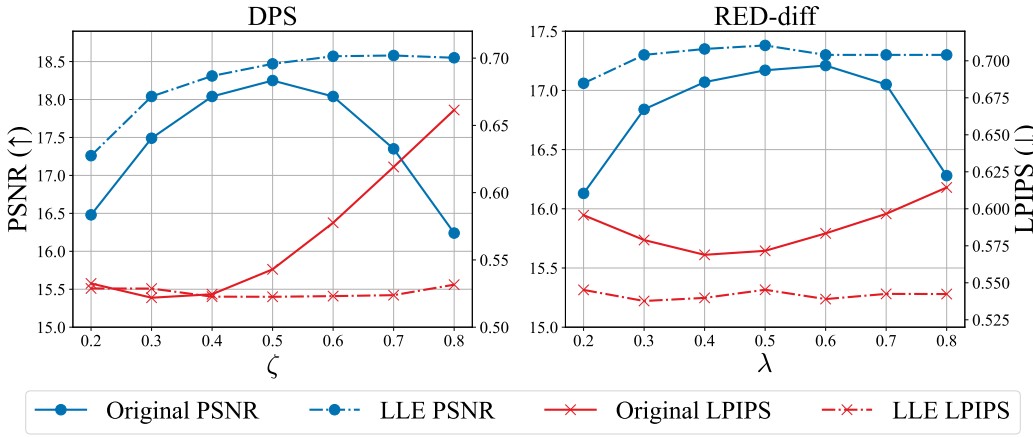

Figure 9: PSNR and LPIPS for varying $\zeta$ in DPS and $\lambda$ in RED-diff on noiseless compressed sensing task on FFHQ using 3 steps.

Table 11: Training time (minutes) of LLE for noiseless inpainting task on CelebA-HQ under different steps. Experiments are performed on a single NVIDIA RTX 3090 GPU.

| Algorithm | 3 Steps | 4 Steps | 5 Steps | 7 Steps | 10 Steps | 15 Steps |
|---|---|---|---|---|---|---|
| DDNM | 2.0 | 2.7 | 3.4 | 5.4 | 6.7 | 10.9 |
| DPS | 2.2 | 2.9 | 3.6 | 5.8 | 7.3 | 11.8 |
| DiffPIR | 2.2 | 2.9 | 3.7 | 5.9 | 7.3 | 11.9 |
| DAPS | 2.6 | 3.7 | 4.7 | 7.1 | 8.6 | 14.3 |

training. Theoretically, this behavior is expected because LLE learns interpolation coefficients by minimizing a weighted sum of the MSE loss and the LPIPS loss. When the LPIPS weight is set to zero, the problem reduces to a standard least squares formulation, which is convex and therefore free from local minima. Although the LPIPS loss introduces non-convexity, its weight is chosen to be sufficiently small such that the overall loss landscape remains smooth and easy to optimize. These findings confirm that LLE's optimization process is stable and computationally efficient in practice.

### D.8 Diversity of the solution

We conduct an additional visualization experiment comparing the diversity of ΠGDM and DPS with and without the proposed LLE. As shown in Figure 10, for typical image restoration tasks, the posterior distribution is generally narrow, meaning that the plausible solutions tend to concentrate around the ground truth. Consequently, the ideal output should remain close to the reference image rather than exhibiting large variations. LLE effectively stabilizes the output and maintains high reconstruction quality.

## E  Detailed decomposition of the inverse algorithms

Here, we present the specific forms of DDRM [5], DDNM [6], DPS [8], ΠGDM [7], RED-diff [11], DiffPIR [9], DMPS [10], ReSample [12], DAPS [14], DCDP [60], LGD [59], MPGD [61], FPS [62], SGS-EDM [63], Latent-DPS [12], PSLD [64], STSL [65], and CMInversion [68] within the canonical form proposed in this paper. For DDNM, DDRM, DMPS, ΠGDM, FPS, and PSLD, we consider linear observations,

$$\mathbf{y} = \mathbf{A}\mathbf{x}_0 + \sigma_{\mathbf{y}}\mathbf{n}, \quad \mathbf{n} \sim \mathcal{N}(\mathbf{0}, \mathbf{I}). \tag{48}$$

For the other algorithms, we consider general observations

$$\mathbf{y} = \mathcal{A}(\mathbf{x}_0) + \sigma_{\mathbf{y}}\mathbf{n}, \quad \mathbf{n} \sim \mathcal{N}(\mathbf{0}, \mathbf{I}). \tag{49}$$

Table 12: Inference time (minutes) of LLE for noiseless inpainting task on the 1k test set of CelebA-HQ. Experiments are performed on a single NVIDIA RTX 3090 GPU.

| Algorithm | Strategy | 3 Steps | 4 Steps | 5 Steps | 7 Steps | 10 Steps | 15 Steps |
|-----------|----------|---------|---------|---------|---------|----------|----------|
| DDNM | - | 2.0 | 2.66 | 3.3 | 5.24 | 6.57 | 10.47 |
| | LLE | 2.0 | 2.66 | 3.34 | 5.35 | 6.68 | 10.74 |
| DPS | - | 5.07 | 6.76 | 8.48 | 13.46 | 16.75 | 26.92 |
| | LLE | 5.13 | 6.84 | 8.49 | 13.61 | 16.96 | 27.21 |
| DiffPIR | - | 5.18 | 7.14 | 8.66 | 14.27 | 17.54 | 28.18 |
| | LLE | 5.45 | 7.14 | 8.76 | 14.35 | 17.79 | 28.31 |
| DAPS | - | 13.2 | 17.29 | 20.57 | 36.7 | 43.58 | 73.36 |
| | LLE | 13.3 | 17.53 | 21.14 | 36.88 | 43.66 | 76.02 |

Table 13: GPU peak memory (MB) of DDNM and DPS with and without LLE on the noiseless inpainting task on FFHQ.

| Algorithm | Strategy | 3 | 4 | 5 | 7 | 10 | 15 |
|-----------|----------|---|---|---|---|-----|-----|
| DDNM | - | 590.25 | 590.25 | 590.25 | 590.25 | 590.25 | 590.25 |
| | LLE | 597.50 | 599.37 | 601.50 | 608.25 | 612.75 | 626.25 |
| DPS | - | 2388.78 | 2388.78 | 2388.78 | 2388.78 | 2388.78 | 2388.78 |
| | LLE | 2396.41 | 2398.41 | 2400.41 | 2407.29 | 2411.79 | 2429.04 |

## E.1 DDRM

The Sampler of DDRM is a single-step DDIM, i.e., tweedie's formula, which is

$$\mathbf{\Phi}_{t_i,\text{DDRM}}(\mathbf{x}_{t_i}) = \frac{\mathbf{x}_{t_i} - \sqrt{1-\overline{\alpha}_{t_i}}\boldsymbol{\epsilon}_{\boldsymbol{\theta}}(\mathbf{x}_{t_i},t_i)}{\sqrt{\overline{\alpha}_{t_i}}}. \tag{50}$$

Consider $\mathbf{A}$ as a diagonal matrix and $s_k$ as the $k$-th singular value. A general matrix $\mathbf{A}$ can be equivalently transformed into a diagonal matrix using Singular Value Decomposition. The correction process is defined element-wise as

$$\mathbf{h}^k_{t_i,\text{DDRM}}(\mathbf{x}_{0,t_i},\mathbf{A},\mathbf{y}) = \begin{cases} \mathbf{x}^k_{0,t_i}, & s_k = 0, \\ \mathbf{x}^k_{0,t_i} + \sqrt{1-\eta^2}\frac{\sqrt{1-\overline{\alpha}_{t_{i-1}}}}{\sqrt{\overline{\alpha}_{t_{i-1}}}}\frac{\mathbf{y}^k-\mathbf{x}^k_{0,t_i}}{\sigma_{\mathbf{y}}/s_k}, & \sqrt{1-\overline{\alpha}_{t_{i-1}}} \leq \frac{\sqrt{\overline{\alpha}_{t_{i-1}}}\sigma_{\mathbf{y}}}{s_k}, \\ (1-\eta_b)\mathbf{x}^k_{0,t_i} + \eta_b\mathbf{y}^k, & \sqrt{1-\overline{\alpha}_{t_{i-1}}} \geq \frac{\sqrt{\overline{\alpha}_{t_{i-1}}}\sigma_{\mathbf{y}}}{s_k}. \end{cases} \tag{51}$$

The Noiser combines noise addition which is similar to DDIM sampling:

$$\mathbf{\Psi}^k_{t_i,\text{DDRM}}(\hat{\mathbf{x}}_{0,t_i}) = \begin{cases} \sqrt{\overline{\alpha}_{t_{i-1}}}\hat{\mathbf{x}}^k_{0,t_i} + \sqrt{1-\eta^2}\sqrt{1-\overline{\alpha}_{t_{i-1}}}\boldsymbol{\epsilon}^k_{\boldsymbol{\theta}}(\mathbf{x}_{t_i},t_i) + \eta\sqrt{1-\overline{\alpha}_{t_{i-1}}}\boldsymbol{\epsilon}^k, & s_k = 0, \\ \sqrt{\overline{\alpha}_{t_{i-1}}}\hat{\mathbf{x}}^k_{0,t_i} + \eta\sqrt{1-\overline{\alpha}_{t_{i-1}}}\boldsymbol{\epsilon}^k, & \sqrt{1-\overline{\alpha}_{t_{i-1}}} < \frac{\sqrt{\overline{\alpha}_{t_{i-1}}}\sigma_{\mathbf{y}}}{s_k}, \\ \sqrt{\overline{\alpha}_{t_{i-1}}}\hat{\mathbf{x}}^k_{0,t_i} + \sqrt{1-\overline{\alpha}_{t_{i-1}} - \frac{\overline{\alpha}_{t_{i-1}}\sigma^2_{\mathbf{y}}}{s^2_k}\eta^2_b}\boldsymbol{\epsilon}^k, & \sqrt{1-\overline{\alpha}_{t_{i-1}}} \geq \frac{\sqrt{\overline{\alpha}_{t_{i-1}}}\sigma_{\mathbf{y}}}{s_k}, \end{cases} \tag{52}$$

where $\boldsymbol{\epsilon} \sim \mathcal{N}(\mathbf{0},\mathbf{I})$, $\eta$ and $\eta_b$ are hyperparameters. $\mathbf{x}^k$ denotes the $k$-th element of the vector $\mathbf{x}$, and the same notation applies to other vectors.

## E.2 DDNM

DDNM is similar to DDRM, where the Sampler is the single-step DDIM as (50). Considering $\mathbf{A}$ as a diagonal matrix, the Corrector is a modified projection operator as

$$\mathbf{h}_{t_i,\text{DDNM}}(\mathbf{x}_{0,t_i},\mathbf{A},\mathbf{y}) = \mathbf{x}_{0,t_i} + \mathbf{\Sigma}_{t_i}\mathbf{A}^{\dagger}(\mathbf{y}-\mathbf{A}\mathbf{x}_{0,t_i}), \tag{53}$$

Table 14: Comparison of DDNM with few-steps and default steps on CelebA-HQ Dataset.

| Condition | Steps | Strategy | PSNR↑ / SSIM↑ / LPIPS↓ | | | |
|---|---|---|---|---|---|---|
| | | | Deblur (aniso) | Inpainting | 4× SR | CS 50% |
| $\sigma_{\mathbf{y}} = 0.05$ | 3 | - | 27.80 / 0.758 / 0.319 | 16.64 / 0.442 / 0.492 | 27.09 / **0.773** / **0.296** | 16.55 / 0.442 / 0.539 |
| | | LLE | **28.08** / **0.784** / **0.291** | **24.38** / **0.552** / **0.433** | **27.84** / 0.770 / 0.299 | **17.29** / **0.473** / **0.520** |
| | 4 | - | 28.98 / 0.795 / 0.285 | 20.27 / 0.510 / 0.457 | 28.35 / **0.803** / 0.276 | 17.39 / 0.446 / 0.518 |
| | | LLE | **29.27** / **0.817** / **0.256** | **25.29** / **0.592** / **0.407** | **28.65** / 0.792 / **0.270** | **18.43** / **0.499** / **0.474** |
| | 5 | - | 29.63 / 0.819 / 0.259 | 22.76 / 0.550 / 0.431 | 28.97 / **0.818** / 0.262 | 18.20 / 0.474 / 0.491 |
| | | LLE | **29.82** / **0.831** / **0.239** | **26.35** / **0.659** / **0.366** | **29.02** / 0.806 / **0.252** | **19.41** / **0.536** / **0.441** |
| | 7 | - | 30.19 / 0.838 / 0.238 | 25.93 / 0.627 / 0.388 | **29.52** / **0.833** / 0.242 | 20.28 / 0.554 / 0.434 |
| | | LLE | **30.29** / **0.841** / **0.225** | **27.69** / **0.719** / **0.328** | 29.33 / 0.818 / **0.233** | **21.32** / **0.608** / **0.385** |
| | 10 | - | 30.45 / 0.845 / 0.227 | 28.19 / 0.716 / 0.333 | **29.07** / **0.821** / 0.241 | 21.30 / 0.636 / 0.379 |
| | | LLE | **30.50** / 0.845 / **0.214** | **29.13** / **0.789** / **0.279** | 29.08 / 0.808 / **0.235** | **22.42** / **0.674** / **0.341** |
| | 15 | - | 30.50 / **0.846** / 0.217 | 30.57 / 0.821 / 0.259 | **29.39** / **0.829** / 0.227 | 23.39 / 0.748 / 0.296 |
| | | LLE | **30.52** / **0.846** / **0.206** | **30.74** / **0.854** / **0.228** | 29.17 / 0.812 / **0.227** | **24.04** / **0.769** / **0.271** |
| | 100 | - | 29.69 / 0.829 / 0.203 | 33.06 / 0.904 / 0.171 | 29.21 / 0.823 / 0.219 | 30.42 / 0.882 / 0.183 |

Table 15: Performance of DDNM with and without LLE under 25, 50, and 100 steps on FFHQ.

| Condition | Steps | Strategy | PSNR↑ / SSIM ↑ / LPIPS ↓ | | | |
|---|---|---|---|---|---|---|
| | | | Deblur (aniso) | Inpainting | 4× SR | CS 50% |
| $\sigma_{\mathbf{y}} = 0$ | 25 | - | 41.14 / 0.972 / 0.064 | **34.33** / **0.941** / 0.109 | 30.92 / **0.885** / 0.146 | 25.01 / 0.837 / 0.220 |
| | | LLE | **41.35** / **0.974** / **0.061** | 34.23 / **0.941** / 0.110 | **30.93** / **0.885** / 0.147 | **25.61** / **0.841** / **0.215** |
| | 50 | - | 41.57 / 0.974 / 0.056 | **35.48** / **0.953** / **0.077** | 31.14 / **0.889** / 0.142 | **28.94** / **0.897** / 0.158 |
| | | LLE | **41.71** / **0.976** / **0.055** | 35.33 / 0.952 / 0.079 | **31.15** / **0.889** / 0.142 | 28.81 / **0.897** / 0.158 |
| | 100 | - | 41.70 / 0.974 / 0.056 | **36.29** / **0.960** / **0.058** | 31.41 / 0.883 / 0.140 | **32.15** / **0.927** / 0.122 |
| | | LLE | **41.86** / **0.976** / **0.054** | 36.18 / 0.959 / 0.060 | 31.37 / **0.883** / **0.139** | 31.56 / **0.927** / 0.122 |

where $\Sigma_{t_i} = \mathrm{diag}\{\lambda_1, \ldots, \lambda_n\}$ is a diagonal matrix with elements defined as

$$
\lambda_k = \begin{cases} 1, & \sigma_{t_{i-1}} \geq \frac{\sqrt{\overline{\alpha}_{t_{i-1}}}\sigma_{\mathbf{y}}}{s_k}, \\ \frac{s_k \sigma_{t_{i-1}}\sqrt{1-\eta^2}}{\sqrt{\overline{\alpha}_{t_{i-1}}}\sigma_{\mathbf{y}}}, & \sigma_{t_{i-1}} < \frac{\sqrt{\overline{\alpha}_{t_{i-1}}}\sigma_{\mathbf{y}}}{s_k}, \\ 1, & s_k = 0. \end{cases} \tag{54}
$$

The Noiser is defined element-wise as

$$
\boldsymbol{\Psi}^k_{t_i,\mathrm{DDNM}}(\hat{\mathbf{x}}_{0,t_i}) = \begin{cases} \sqrt{\overline{\alpha}_{t_{i-1}}}\hat{\mathbf{x}}^k_{0,t_i} + \sqrt{1-\eta^2}\sigma_{t_{i-1}}\boldsymbol{\epsilon}^k_{\boldsymbol{\theta}}(\mathbf{x}_{t_i}, t_i) + \eta \sigma_{t_{i-1}}\boldsymbol{\epsilon}^k, & s_k = 0, \\ \sqrt{\overline{\alpha}_{t_{i-1}}}\hat{\mathbf{x}}^k_{0,t_i} + \eta \sigma_{t_{i-1}}\boldsymbol{\epsilon}^k, & \sigma_{t_{i-1}} < \frac{\sqrt{\overline{\alpha}_{t_{i-1}}}\sigma_{\mathbf{y}}}{s_k}, \\ \sqrt{\overline{\alpha}_{t_{i-1}}}\hat{\mathbf{x}}^k_{0,t_i} + \sqrt{\sigma^2_{t_{i-1}} - \frac{\sigma^2_{\mathbf{y}}\overline{\alpha}_{t_{i-1}}}{s_k^2}}\boldsymbol{\epsilon}^k, & \sigma_{t_{i-1}} \geq \frac{\sqrt{\overline{\alpha}_{t_{i-1}}}\sigma_{\mathbf{y}}}{s_k}, \end{cases}
$$

$$\tag{55}$$

where $\boldsymbol{\epsilon} \sim \mathcal{N}(\mathbf{0}, \mathbf{I})$ and $\sigma_{t_i} = \sqrt{1-\overline{\alpha}_{t_i}}$.

### E.3 DPS

We rewrite the update formula of DPS as

$$
\mathbf{x}_{t_{i-1}} = \sqrt{\overline{\alpha}_{t_{i-1}}}\left(\mathbf{x}_{0,t_i} - \zeta_{t_i}/\sqrt{\overline{\alpha}_{t_i}}\nabla_{\mathbf{x}_{t_i}}\|\mathbf{y} - \mathcal{A}(\mathbf{x}_{0,t_i})\|^2\right) + c_1 \boldsymbol{\epsilon} + c_2 \boldsymbol{\epsilon}_{\boldsymbol{\theta}}(\mathbf{x}_{t_i}, t_i), \tag{56}
$$

where $\mathbf{x}_{0,t_i}$ is the estimation using Tweedie's formula as (50). Thus the Sampler of DPS is a single-step DDIM. The Corrector is

$$
\mathbf{h}_{t_i,\mathrm{DPS}}(\mathbf{x}_{0,t_i}, \mathcal{A}, \mathbf{y}) = \mathbf{x}_{0,t_i} - \frac{1}{\sqrt{\overline{\alpha}_{t_{i-1}}}}\zeta_{t_i}\nabla_{\mathbf{x}_{t_i}}\|\mathbf{y} - \mathcal{A}(\mathbf{x}_{0,t_i})\|^2, \tag{57}
$$

where $\zeta_{t_i}$ is the learning rate hyperparameter. The Noiser is exactly the same as DDIM sampling

$$
\boldsymbol{\Psi}_{t_i,\mathrm{DPS}}(\hat{\mathbf{x}}_{0,t_i}) = \sqrt{\overline{\alpha}_{t_{i-1}}}\hat{\mathbf{x}}_{0,t_i} + c_1 \boldsymbol{\epsilon} + c_2 \boldsymbol{\epsilon}_{\boldsymbol{\theta}}(\mathbf{x}_{t_i}, t_i), \tag{58}
$$

Table 16: Effect of optimization iterations on LLE performance.

| Train steps | Deblur (noiseless) | Inp (noiseless) | SR (noiseless) | CS (noiseless) |
|---|---|---|---|---|
| - | 39.82 / 0.964 / 0.087 | 26.34 / 0.719 / 0.380 | **29.99 / 0.865 / 0.211** | 18.29 / 0.591 / 0.487 |
| 25 | 39.93 / 0.965 / 0.086 | 28.21 / 0.821 / 0.316 | 29.93 / 0.863 / 0.207 | 18.87 / 0.597 / 0.476 |
| 50 | 40.12 / 0.967 / 0.085 | 28.61 / 0.834 / 0.303 | 29.95 / 0.863 / 0.208 | 19.13 / 0.602 / 0.468 |
| **100** | 40.08 / 0.966 / 0.084 | 28.71 / 0.838 / 0.293 | 29.94 / 0.863 / 0.206 | 19.21 / 0.605 / 0.463 |
| 150 | 40.10 / 0.966 / 0.084 | 28.77 / 0.840 / 0.290 | 29.91 / 0.862 / 0.204 | 19.23 / 0.603 / 0.461 |
| 200 | **40.14 / 0.967 / 0.084** | **28.82 / 0.842 / 0.288** | 29.93 / 0.863 / 0.204 | **19.25 / 0.604 / 0.462** |

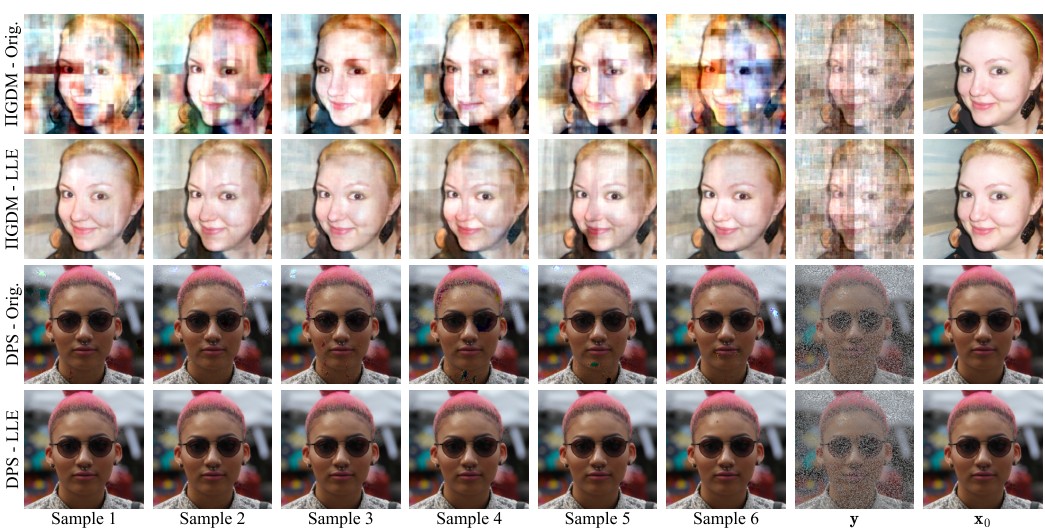

Figure 10: Visualization of reconstruction results under different random seeds.

where

$$c_1 = \eta \sqrt{1 - \frac{\overline{\alpha}_{t_i}}{\overline{\alpha}_{t_{i-1}}}} \sqrt{\frac{1 - \overline{\alpha}_{t_{i-1}}}{1 - \overline{\alpha}_{t_i}}},$$

$$c_2 = \sqrt{1 - \overline{\alpha}_{t_{i-1}} - c_1^2}, \tag{59}$$

with $\eta$ as a hyperparameter and $\epsilon \sim \mathcal{N}(\mathbf{0}, \mathbf{I})$. Similar transformation is applicable to all posterior sampling approaches, including $\Pi$GDM and DMPS. The derivation is omitted hereafter.

### E.4 $\Pi$GDM

The Sampler of $\Pi$GDM is a single-step DDIM as (50). The Corrector is

$$\mathbf{h}_{t_i, \Pi\text{GDM}}(\mathbf{x}_{0,t_i}, \mathbf{A}, \mathbf{y}) = \mathbf{x}_{0,t_i} + \sqrt{\overline{\alpha}_{t_i}/\overline{\alpha}_{t_{i-1}}} \left( (\mathbf{y} - \mathbf{A}\mathbf{x}_{0,t_i})^T \left( \mathbf{A}\mathbf{A}^T + \frac{\sigma_{\mathbf{y}}^2}{r_{t_i}^2}\mathbf{I} \right)^{-1} \mathbf{A} \frac{\partial \mathbf{x}_{0,t_i}}{\partial \mathbf{x}_{t_i}} \right)^T, \tag{60}$$

where $r_{t_i} = \sqrt{1 - \overline{\alpha}_{t_i}}$. The Noiser is the same as DDIM sampling as (58).

### E.5 RED-diff

The Sampler of RED-diff is a single-step DDIM as (50). RED-diff differs from other algorithms in that it uses an optimization process to update $\hat{\mathbf{x}}_{0,t_i}$. Here we consider only the gradient descent update, where the Corrector of RED-diff is equivalently given by

$$\mathbf{h}_{t_i, \text{RED-diff}}(\mathbf{x}_{0,t_i}, \mathcal{A}, \mathbf{y}) = \hat{\mathbf{x}}_{0,t_{i+1}} + \xi \left( \mathbf{x}_{0,t_i} - \hat{\mathbf{x}}_{0,t_{i+1}} - \lambda \nabla_{\hat{\mathbf{x}}_{0,t_{i+1}}} \left\| \mathbf{y} - \mathcal{A}\left(\hat{\mathbf{x}}_{0,t_{i+1}}\right) \right\|^2 \right), \tag{61}$$

where $\xi$ is the learning rate, and $\lambda$ controls the trade-off between the prior and likelihood terms. However, this formulation fails to reconstruct reasonable results under few steps in our experiments. We find the issue lies in the likelihood term, so we adjust it as follows to improve its performance unders few steps

$$\mathbf{h}_{t_i,\text{RED-diff}}(\mathbf{x}_{0,t_i}, \mathcal{A}, \mathbf{y}) = \hat{\mathbf{x}}_{0,t_{i+1}} + \xi\left(\mathbf{x}_{0,t_i} - \hat{\mathbf{x}}_{0,t_{i+1}} - \lambda\nabla_{\mathbf{x}_{0,t_i}} \|\mathbf{y} - \mathcal{A}(\mathbf{x}_{0,t_i})\|^2\right), \qquad (62)$$

where we use the $\mathbf{x}_{0,t_i}$ to calculate the loss for the likelihood term rather than $\hat{\mathbf{x}}_{0,t_{i+1}}$. The Noiser of RED-diff is directly noise addition as

$$\mathbf{\Psi}_{t_i,\text{RED-diff}}(\hat{\mathbf{x}}_{0,t_i}) = \sqrt{\overline{\alpha}_{t_{i-1}}}\hat{\mathbf{x}}_{0,t_i} + \sqrt{1 - \overline{\alpha}_{t_{i-1}}}\boldsymbol{\epsilon}, \quad \boldsymbol{\epsilon} \sim \mathcal{N}(\mathbf{0}, \mathbf{I}). \qquad (63)$$

### E.6 DiffPIR

The Sampler of DiffPIR is a single-step DDIM as (50). The Corrector solves a proximal point problem as

$$\mathbf{h}_{t_i,\text{DiffPIR}}(\mathbf{x}_{0,t_i}, \mathcal{A}, \mathbf{y}) = \arg\min_{\mathbf{x}} \|\mathbf{y} - \mathcal{A}(\mathbf{x})\|^2 + \rho_t \|\mathbf{x} - \mathbf{x}_{0,t_i}\|^2, \qquad (64)$$

where $\rho_t = \lambda\sigma_{\mathbf{y}}^2\overline{\alpha}_t / (1 - \overline{\alpha}_t)$ and $\lambda$ is a hyperparameter. The Noiser of DiffPIR is a modified version of (58) as

$$\mathbf{\Psi}_{t_i,\text{DiffPIR}}(\hat{\mathbf{x}}_{0,t_i}) = \sqrt{\overline{\alpha}_{t_{i-1}}}\hat{\mathbf{x}}_{0,t_i} + \eta\sqrt{1 - \overline{\alpha}_{t_{i-1}}}\boldsymbol{\epsilon} + \sqrt{1 - \eta^2}\frac{\sqrt{1 - \overline{\alpha}_{t_{i-1}}}}{\sqrt{1 - \overline{\alpha}_{t_i}}}\left(\mathbf{x}_{t_i} - \sqrt{\overline{\alpha}_{t_i}}\hat{\mathbf{x}}_{0,t_i}\right), \qquad (65)$$

where they calculate an effective $\hat{\boldsymbol{\epsilon}}_{\boldsymbol{\theta}}(\mathbf{x}_{t_i}, t_i)$ using the corrected sample $\hat{\mathbf{x}}_{0,t_i}$, and $\boldsymbol{\epsilon} \sim \mathcal{N}(\mathbf{0}, \mathbf{I})$.

### E.7 DMPS

The Sampler of DMPS is the single-step DDIM (50). Consider the SVD of the observation matrix $\mathbf{A} = \mathbf{U}\mathbf{\Sigma}\mathbf{V}^T$, then the Corrector is

$$\mathbf{h}_{t_i,\text{DMPS}}(\mathbf{x}_{0,t_i}, \mathbf{A}, \mathbf{y}) = \mathbf{x}_{0,t_i} + \frac{1}{\sqrt{\overline{\alpha}_{t_{i-1}}}}\lambda\frac{1 - \alpha_{t_i}}{\sqrt{\alpha_{t_i}}}\nabla_{\mathbf{x}_{t_i}} \log \tilde{p}(\mathbf{y}|\mathbf{x}_{t_i}), \qquad (66)$$

where $\alpha_{t_i} = \overline{\alpha}_{t_i}/\overline{\alpha}_{t_{i-1}}$ and

$$\nabla_{\mathbf{x}_{t_i}} \log \tilde{p}(\mathbf{y}|\mathbf{x}_{t_i}) = \frac{1}{\sqrt{\overline{\alpha}_{t_i}}}\mathbf{V}\mathbf{\Sigma}\left(\sigma_{\mathbf{y}}^2\mathbf{I} + \frac{1 - \overline{\alpha}_{t_i}}{\overline{\alpha}_{t_i}}\mathbf{\Sigma}^2\right)^{-1}\left(\mathbf{U}^T\mathbf{y} - \frac{1}{\sqrt{\overline{\alpha}_{t_i}}}\mathbf{\Sigma}\mathbf{V}^T\mathbf{x}_{t_i}\right). \qquad (67)$$

The Noiser is as (58).

### E.8 ReSample

ReSample [12] is originally designed for latent diffusion, thus we set the Encoder and Decoder in ReSample to identity mappings for pixel diffusion models. The Sampler of ReSample is the single-step DDIM as (50). The Corrector is an optimization process which is termed hard data consistency as

$$\mathbf{h}_{t_i,\text{ReSample}}(\mathbf{x}_{0,t_i}, \mathcal{A}, \mathbf{y}) = \arg\min_{\mathbf{x}} \|\mathbf{y} - \mathcal{A}(\mathbf{x})\|^2, \qquad (68)$$

with the initial point of the optimization algorithm set as $\mathbf{x}^{\text{init}} = \mathbf{x}_{0,t_i}$. The Noiser of ReSample is the ReSample method, which is

$$\mathbf{\Psi}_{t_i,\text{ReSample}}(\hat{\mathbf{x}}_{0,t_i}) = \frac{\sigma_{t_{i-1}}^2\sqrt{\overline{\alpha}_{t_{i-1}}}\hat{\mathbf{x}}_{0,t_i} + (1 - \overline{\alpha}_{t_{i-1}})\mathbf{x}'_{t_i}}{\sigma_{t_{i-1}}^2 + 1 - \overline{\alpha}_{t_{i-1}}} + \sqrt{\frac{\sigma_{t_{i-1}}^2(1 - \overline{\alpha}_{t_{i-1}})}{\sigma_{t_{i-1}}^2 + 1 - \overline{\alpha}_{t_{i-1}}}}\boldsymbol{\epsilon}, \quad \boldsymbol{\epsilon} \sim \mathcal{N}(\mathbf{0}, \mathbf{I}), \qquad (69)$$

where

$$\sigma_{t_i}^2 = \gamma\left(\frac{1 - \overline{\alpha}_{t_{i-1}}}{\overline{\alpha}_{t_i}}\right)\left(1 - \frac{\overline{\alpha}_{t_i}}{\overline{\alpha}_{t_{i-1}}}\right), \qquad (70)$$

and

$$\mathbf{x}'_{t_i} = \sqrt{\overline{\alpha}_{t_{i-1}}}\mathbf{x}_{0,t_i} + c_1\boldsymbol{\epsilon} + c_2\boldsymbol{\epsilon}_{\boldsymbol{\theta}}(\mathbf{x}_{t_i}, t_i), \qquad (71)$$

with $c_1 = \eta\sqrt{1 - \frac{\overline{\alpha}_{t_i}}{\overline{\alpha}_{t_{i-1}}}}\sqrt{\frac{1 - \overline{\alpha}_{t_{i-1}}}{1 - \overline{\alpha}_{t_i}}}$ and $c_2 = \sqrt{1 - \overline{\alpha}_{t_{i-1}} - c_1^2}$. $\gamma$ is a hyperparameter.

## E.9 DAPS

The Sampler of DAPS is a $k$-step DDIM sampler, i.e.,

$$\mathbf{\Phi}_{t_i,\text{DAPS}} = \text{DDIM}_k(\mathbf{x}_{t_i}, t_i). \tag{72}$$

The Corrector follows a Langevin dynamics process, with the iteration given by

$$\mathbf{x}_{0,t_i}^{j+1} = \mathbf{x}_{0,t_i}^j - \eta_{t_i} \nabla_{\mathbf{x}_{0,t_i}^j} \left( \left\| \mathbf{x}_{0,t_i}^j - \mathbf{x}_{0,t_i} \right\|^2 / 2r_{t_i}^2 + \left\| \mathcal{A}(\mathbf{x}_{0,t_i}^j) - \mathbf{y} \right\|^2 / 2\sigma_{\mathbf{y}}^2 \right) + \sqrt{2\eta_{t_i}} \boldsymbol{\epsilon}_j, \tag{73}$$

where $\boldsymbol{\epsilon}_j \sim \mathcal{N}(\mathbf{0}, \mathbf{I})$, and $r_{t_i}$ is a heuristics hyperparameter, which can be set as $r_{t_i} = \sqrt{1 - \overline{\alpha}_{t_i}}$ [7]. The Noiser is direct noise addition as (63).

## E.10 DCDP

The Sampler of DCDP is either a single-step DDIM as (50) or a k-step DDIM as (72). The Corrector starts from $\hat{\mathbf{x}}_{0,t_i}$ and performs fixed number of gradient descent steps to minimize $\|\mathcal{A}(\mathbf{x}_0) - \mathbf{y}\|^2$. The Noiser is direct noise addition as (63).

## E.11 LGD

The Sampler of LGD is a single-step DDIM as (50). The Corrector is

$$\mathbf{h}_{t_i,\text{LGD}}(\mathbf{x}_{0,t_i}, \mathcal{A}, \mathbf{y}) = \mathbf{x}_{0,t_i} - \frac{1}{\sqrt{\overline{\alpha}_{t_{i-1}}}} \zeta_{t_i} \nabla_{\mathbf{x}_{t_i}} \log \left( \frac{1}{n} \sum_{j=1}^n \exp \left( -\ell_{\mathbf{y}} \left( \mathbf{x}^{(j)} \right) \right) \right), \tag{74}$$

where $\zeta_{t_i}$ is the step-size, $\mathbf{x}^{(j)} \sim q(\mathbf{x}_0|\mathbf{x}_{t_i})$, and $\ell_{\mathbf{y}}(\mathbf{x}) = \|\mathbf{y} - \mathcal{A}(\mathbf{x})\|^2$. $q(\mathbf{x}_0|\mathbf{x}_{t_i})$ can be approximated as $\mathcal{N}(\mathbf{x}_{0,t_i}, r_{t_i}^2 \mathbf{I})$ where $r_{t_i} = \sigma_t/\sqrt{1 + \sigma_t^2}$ with

$$\sigma_t^2 = \frac{1 - \overline{\alpha}_{t_{i+1}}}{1 - \overline{\alpha}_{t_i}} \left( 1 - \frac{\overline{\alpha}_{t_i}}{\overline{\alpha}_{t_{i+1}}} \right). \tag{75}$$

The Noiser is as (58).

## E.12 MPGD

We only consider MPGD without manifold projection. The Sampler is a single-step DDIM as (50). The Corrector is

$$\mathbf{h}_{t_i,\text{MPGD}}(\mathbf{x}_{0,t_i}, \mathcal{A}, \mathbf{y}) = \mathbf{x}_{0,t_i} - \zeta_{t_i} \nabla_{\mathbf{x}_{0,t_i}} \|\mathbf{y} - \mathcal{A}(\mathbf{x}_{0,t_i})\|. \tag{76}$$

The Noiser is as (58).

## E.13 FPS

For simplicity, we only consider FPS with particle size 1. The Sampler is a single-step DDIM as (50). The Corrector is

$$\mathbf{h}_{t_i,\text{FPS}}(\mathbf{x}_{0,t_i}, \mathbf{A}, \mathbf{y}) = \frac{1}{\sqrt{\overline{\alpha}_{t_{i-1}}}} \Sigma_{t_{i-1}}^{\text{FPS}} \left( \frac{1}{c(1 - \overline{\alpha}_{t_{i-1}})} \sqrt{\overline{\alpha}_{t_{i-1}}} \mathbf{x}_{0,t_i} + \frac{1}{\sigma_{\mathbf{y}}^2 \sqrt{\overline{\alpha}_{t_{i-1}}}} \mathbf{A}^T \mathbf{y} \right), \tag{77}$$

where

$$\Sigma_{t_{i-1}}^{\text{FPS}} = \left( \frac{1}{c(1 - \overline{\alpha}_{t_{i-1}})} + \frac{1}{\sigma_{\mathbf{y}}^2 \overline{\alpha}_{t_{i-1}}} \mathbf{A}^T \mathbf{A} \right)^{-1}, \tag{78}$$

and $c$ is the DDIM hyperparameter. The Noiser is

$$\mathbf{\Psi}_{t_i,\text{FPS}}(\hat{\mathbf{x}}_{0,t_i}) = \sqrt{\overline{\alpha}_{t_{i-1}}} \hat{\mathbf{x}}_{0,t_i} + \Sigma_{t_i}^{\text{FPS}} \left( \sqrt{\frac{1-c}{c}} \boldsymbol{\epsilon}_{\boldsymbol{\theta}}(\mathbf{x}_{t_i}, t_i) + \frac{1}{\sigma_{\mathbf{y}}^2 \overline{\alpha}_{t_{i-1}}} \mathbf{A}^T \boldsymbol{\epsilon}_{\mathbf{y},t_i} \right) + \Sigma_{t_i}^{\text{FPS} \frac{1}{2}} \boldsymbol{\epsilon}, \tag{79}$$

where $\boldsymbol{\epsilon} \sim \mathcal{N}(\mathbf{0}, \mathbf{I})$, and $\boldsymbol{\epsilon}_{\mathbf{y}, t_i}$ is defined recursively as

$$\boldsymbol{\epsilon}_{\mathbf{y}, t_i} = \sqrt{(1-c)(1-\overline{\alpha}_{t_i})} \frac{\mathbf{y}_{t_{i+1}} - \sqrt{\overline{\alpha}_{t_{i+1}}} \mathbf{y}}{\sqrt{1 - \overline{\alpha}_{t_{i+1}}}} + \sqrt{c(1 - \overline{\alpha}_{t_i})} \mathbf{A} \boldsymbol{\epsilon}_{t_i}, \tag{80}$$

$$\mathbf{y}_{t_i} = \sqrt{\overline{\alpha}_{t_i}} \mathbf{y} + \boldsymbol{\epsilon}_{\mathbf{y}, t_i}, \quad \boldsymbol{\epsilon}_{t_i} \sim \mathcal{N}(\mathbf{0}, \mathbf{I}), \tag{81}$$

with $\mathbf{y}_{t_S} = \mathbf{A} \boldsymbol{\epsilon}_{t_S}$.

### E.14 SGS-EDM

The Sampler is a k-step DDIM as (72). The Corrector is

$$\mathbf{h}_{t_i, \text{SGS-EDM}}(\mathbf{x}_{0, t_i}, \mathbf{A}, \mathbf{y}) = \mathbf{\Lambda}^{-1} \left( \frac{1}{\sigma_{\mathbf{y}}^2} \mathbf{A}^T \mathbf{y} + \frac{\overline{\alpha}_{t_{i-1}}}{1 - \overline{\alpha}_{t_{i-1}}} \mathbf{x}_{0, t_i} \right), \tag{82}$$

where

$$\mathbf{\Lambda} = \frac{1}{\sigma_{\mathbf{y}}^2} \mathbf{A}^T \mathbf{A} + \frac{\overline{\alpha}_{t_{i-1}}}{1 - \overline{\alpha}_{t_{i-1}}} \mathbf{I}. \tag{83}$$

The Noiser is

$$\mathbf{\Psi}_{t_i, \text{SGS-EDM}} = \sqrt{\overline{\alpha}_{t_{i-1}}} \hat{\mathbf{x}}_{0, t_i} + \sqrt{\overline{\alpha}_{t_{i-1}}} \mathbf{\Lambda}^{-\frac{1}{2}} \boldsymbol{\epsilon}, \quad \boldsymbol{\epsilon} \sim \mathcal{N}(\mathbf{0}, \mathbf{I}). \tag{84}$$

### E.15 Latent-DPS

For latent-space methods, we denote $\mathcal{E}(\cdot), \mathcal{D}(\cdot)$, and $\mathbf{z}_{t_i}$ as the Encoder, Decoder, and latent variable, respectively. The decomposition of Latent-DPS is similar to that of DPS, with the Corrector adjusted as

$$\mathbf{h}_{t_i, \text{Latent-DPS}}(\mathbf{x}_{0, t_i}, \mathcal{A}, \mathbf{y}) = \mathbf{x}_{0, t_i} - \frac{1}{\sqrt{\overline{\alpha}_{t_{i-1}}}} \zeta_{t_i} \nabla_{\mathbf{x}_{t_i}} \|\mathbf{y} - \mathcal{A}(\mathcal{D}(\mathbf{x}_{0, t_i}))\|^2. \tag{85}$$

### E.16 PSLD

The Sampler is a single-step DDIM as (50). The Corrector is

$$\begin{aligned}
\mathbf{h}_{t_i, \text{PSLD}}(\mathbf{z}_{0, t_i}, \mathbf{A}, \mathbf{y}) = & \mathbf{z}_{0, t_i} - \frac{1}{\sqrt{\overline{\alpha}_{t_i}}} \eta_{t_i} \nabla_{\mathbf{z}_{t_i}} \|\mathbf{y} - \mathbf{A}(\mathcal{D}(\mathbf{z}_{0, t_i}))\|^2 \\
& - \frac{1}{\sqrt{\overline{\alpha}_{t_i}}} \gamma_{t_i} \nabla_{\mathbf{z}_{t_i}} \|\mathbf{z}_{0, t_i} - \mathcal{E}(\mathbf{A}^T \mathbf{y} + (\mathbf{I} - \mathbf{A}^T \mathbf{A})) \mathcal{D}(\mathbf{z}_{0, t_i})\|^2.
\end{aligned} \tag{86}$$

The Noiser is as (58).

### E.17 STSL

The Sampler is a single-step DDIM as (50). The Corrector is

$$\begin{aligned}
\mathbf{h}_{t_i, \text{STSL}}(\mathbf{z}_{0, t_i}, \mathcal{A}, \mathbf{y}) = & \mathbf{z}_{0, t_i} - \frac{1}{\sqrt{\overline{\alpha}_{t_i}}} \eta_{t_i} \nabla_{\mathbf{z}_{t_i}} \|\mathbf{y} - \mathcal{A}(\mathcal{D}(\mathbf{z}_{0, t_i}))\|^2 \\
& - \frac{\gamma}{d\sqrt{1 - \overline{\alpha}_{t_i}}} \nabla_{\mathbf{z}_{t_i}} \sum_{j=1}^{n} \boldsymbol{\epsilon}_j^T (\boldsymbol{\epsilon}_\theta(\mathbf{z}_{t_i} + \boldsymbol{\epsilon}_j, t_i) - \boldsymbol{\epsilon}_\theta(\mathbf{z}_{t_i}, t_i)),
\end{aligned} \tag{87}$$

where $d$ is the dimension of the latent variable, and $\boldsymbol{\epsilon}_j \sim \mathcal{N}(\mathbf{0}, \mathbf{I})$. The Noiser is as (58).

### E.18 CMInversion

We denote the consistency model as $\mathbf{g}_\theta(\cdot, \cdot)$. The Sampler of CMInversion is a single consistency model inference, i.e.,

$$\mathbf{\Phi}_{t_i, \text{CMInversion}}(\mathbf{x}_{t_i}) = \mathbf{g}_\theta(\mathbf{x}_{t_i}, t_i). \tag{88}$$

The correction process can be recursively defined as

$$\mathbf{x}_{t_i,k} = \sqrt{\overline{\alpha}_{t_i}}\mathbf{x}_{0,t_i,k-1} + \sqrt{1-\overline{\alpha}_{t_i}}\mathbf{n}_{t_i,k}, \tag{89}$$

$$\mathbf{x}_{0,t_i,k} = \mathbf{g}_{\boldsymbol{\theta}}(\mathbf{x}_{t_i,k}, t_i), \tag{90}$$

$$\mathbf{n}_{t_i,k+1} = \mathbf{n}_{t_i,k} - \zeta\frac{\mathrm{d}}{\mathrm{d}\mathbf{n}_{t_i,k}}\|\mathcal{A}(\mathbf{x}_{0,t_i,k} + \sigma_{\mathbf{y}}\boldsymbol{\epsilon}) - \mathbf{y}\|^2, \quad \boldsymbol{\epsilon} \sim \mathcal{N}(\mathbf{0}, \mathbf{I}), \tag{91}$$

where $\mathbf{x}_{0,t_i,0} = \mathbf{x}_{0,t_i}$, and $\mathbf{n}_{t_i,1} \sim \mathcal{N}(\mathbf{0},\mathbf{I})$. Then the Corrector is

$$\mathbf{h}_{t_i,\text{CMInversion}}(\mathbf{x}_{0,t_i}, \mathcal{A}, \mathbf{y}) = \mathbf{x}_{0,t_i,K}. \tag{92}$$

The Noiser is direct noise addition as

$$\boldsymbol{\Psi}_{t_i,\text{CMInversion}}(\hat{\mathbf{x}}_{0,t_i}) = \sqrt{\overline{\alpha}_{t_{i-1}}}\hat{\mathbf{x}}_{0,t_i} + \sqrt{1-\overline{\alpha}_{t_{i-1}}}\boldsymbol{\epsilon}, \quad \boldsymbol{\epsilon} \sim \mathcal{N}(\mathbf{0}, \mathbf{I}). \tag{93}$$

## F  Specific design for DDNM and DDRM on noisy linear tasks

DDRM and DDNM are specifically designed to take aware of the observation noise in noisy scenarios. In particular, the variance of the additional noise is reduced such that the variance of the observation noise adds the additional noise equal to the desired variance of the next step in the diffusion model. This implies that in noisy linear tasks, the expected ground truth of the output of the Correctors in DDRM and DDNM is not a noiseless image, but rather a noisy image perturbed by the observation noise. Therefore, we adjust the ground truth for the LLE learning in noisy scenarios for DDRM and DDNM to better align with the characteristics of these algorithms.

The implementation is straightforward by simply passing the noiseless image through the Correctors of DDRM and DDNM, i.e.,

$$\mathbf{x}_{t_i,\text{gt,DDRM}}^{(n)} = \mathbf{h}_{t_i,\text{DDRM}}\left(\mathbf{x}_0^{(n)}, \mathbf{A}, \mathbf{y}^{(n)}\right), \tag{94}$$

$$\mathbf{x}_{t_i,\text{gt,DDNM}}^{(n)} = \mathbf{h}_{t_i,\text{DDNM}}\left(\mathbf{x}_0^{(n)}, \mathbf{A}, \mathbf{y}^{(n)}\right). \tag{95}$$

The objective for the LLE training is then

$$\min_{\gamma_{t_i,j}^{\parallel}, \gamma_{t_i,j}^{\perp}, j=0,\ldots,S-i} \mathbb{E}_{n\sim\mathcal{U}\{1,\ldots,N\}}\mathcal{L}\left(\tilde{\mathbf{x}}_{0,t_i}^{(n)}, \mathbf{x}_{t_i,\text{gt,DDRM}}^{(n)}\right), \tag{96}$$

$$\min_{\gamma_{t_i,j}^{\parallel}, \gamma_{t_i,j}^{\perp}, j=0,\ldots,S-i} \mathbb{E}_{n\sim\mathcal{U}\{1,\ldots,N\}}\mathcal{L}\left(\tilde{\mathbf{x}}_{0,t_i}^{(n)}, \mathbf{x}_{t_i,\text{gt,DDNM}}^{(n)}\right), \tag{97}$$

where the ground truth in (19) is replaced by (94) and (95). The remaining training details remain unchanged.

## G  Experimental details

### G.1  Inverse problems settings

All the linear inverse problems follow [5], while the nonlinear deblurring follows [8]. We describe the details of all these problems to maintain completeness. For $4\times$ Super-Resolution, we use a $4 \times 4$ average pooling operation for downsampling. For Inpainting, we apply a random $50\%$ mask to all pixels, where the mask is generated using the same random seed to ensure fairness. For anisotropic deblurring, we use Gaussian blur kernels with standard deviations of 20 and 1 in two directions, respectively. For compressed sensing, we use Walsh-Hadamard transform with a downsampling rate of $50\%$. For nonlinear deblurring, we adopt the neural network from [70] as the observation function. Unlike [8], we follow [14] by using a deterministic observation equation generated with a fixed random seed to ensure fairness.

For the noisy tasks, the noise standard deviation is multiplied by 2 to account for the data range of $[-1, 1]$, i.e., in the implementation, we actually add Gaussian noise with a standard deviation of $0.1$ to the observations.

## G.2 Hyperparameters for base inverse algorithms

We carefully tune the hyperparameters of the inverse algorithms to ensure they achieve satisfactory performance under few steps. The detailed hyperparameter settings are as follows.

**DDRM.** We use the recommended values from [5], i.e., $\eta = 0.85$ and $\eta_b = 1.0$.

**DDNM.** We use the recommended value from [6], i.e., $\eta = 0.85$.

**DPS.** We find that the learning rate form recommended by [8] is too small for few steps setting. Therefore, we follow the learning rate from [12], i.e., $\zeta_{t_i} = \zeta\sqrt{\bar{\alpha}_{t_i}}$, which demonstrated superior performance with fewer steps. The specific value of $\zeta$ is adjusted for different tasks, as shown in Table 17. The DDIM hyperparameter $\eta$ is set to 1.0.

Table 17: Tuned learning rate $\zeta$ for DPS.

|  | Deblur (aniso) | Inpainting | $4\times$ SR | CS $50\%$ | Deblur (nonlinear) |
|---|---|---|---|---|---|
| CelebA-HQ | 0.5 | 1.0 | 6.0 | 0.1 | 0.1 |
| FFHQ | 0.5 | 1.0 | 5.0 | 0.5 | 0.1 |

**ΠGDM.** We set the DDIM hyperparameter $\eta = 1.0$.

**RED-diff.** We set the learning rate $\xi = 1.0$. The weight $\lambda$ is adjusted for different tasks, as shown in Table 18.

Table 18: Tuned weight $\lambda$ for RED-diff.

|  | Deblur (aniso) | Inpainting | $4\times$ SR | CS $50\%$ | Deblur (nonlinear) |
|---|---|---|---|---|---|
| CelebA-HQ | 0.5 | 0.5 | 7.0 | 0.5 | 0.2 |
| FFHQ | 0.7 | 0.4 | 5.0 | 0.5 | 0.2 |

**DiffPIR.** We set the DDIM hyperparameter $\eta = 1.0$ and $\lambda = 7.0$, which performs well across all tasks. For the optimizer of the proximal point problem (64), we use the schedule-free AdamW [45], which offers better stability than SGDM in some tasks under few steps. The learning rate is set to 0.1 and the number of optimization steps is 50.

**DMPS.** We specify the DDIM parameters as $c_1 = \eta\sqrt{1 - \bar{\alpha}_{t_{i-1}}}$, $c_2 = \sqrt{1 - \eta^2}\sqrt{1 - \bar{\alpha}_{t_{i-1}}}$, and set $\eta = 0.85$. The weight $\lambda$ is carefully tuned task-by-task and step-by-step.

**ReSample.** We set the DDIM hyperparameter $\eta = 1.0$ and $\gamma = 100$. The optimizer for hard consistency (68) is the SGDM algorithm, with a learning rate of 0.01, momentum of 0.9, and 50 optimization steps.

**DAPS.** We use $k = 5$ steps for the DDIM sampler and perform the Langevin dynamics for 100 iterations per timestep with the step size as $\eta_{t_i} = \eta_0 \left(\delta + t_i/T(1 - \delta)\right)$, where $\eta_0 = 0.0001$, $\delta = 0.01$, and $T = 1000$. For noiseless linear inverse problems, we find that replacing the gradient term in (73) with

$$\nabla_{\mathbf{x}_{0,t_i}^j} \frac{1}{2\eta_{t_i}} \left\|\mathbf{A}\mathbf{x}_{0,t_i}^j - \mathbf{y}_0\right\|^2 \tag{98}$$

leads to better results under few steps. For nonlinear problems and noisy scenarios, we follow the original setting and set $\sigma_{\mathbf{y}}$ in (73) to 0.02 instead of 0.05 for better results, which is consistent with the observations in [14].

**DCDP.** We use Tweedie's formula approximation in the purification stage. The optimizer is SGDM, with the learning rate of 0.01 and momentum of 0.9. The number of optimization steps is set to 50 for Super-Resolution and 25 for all other tasks.

**LGD.** The particle number $n$ is fixed as 50 in all experiments. The step-size $\zeta$ is set to 0.5

for deblurring, 1.0 for inpainting, 5.0 for super-resolution, and 0.1 for compressive sensing.

**MPGD.** The step-size $\zeta$ is set to 0.5 for deblurring, 1.0 for inpainting, 5.0 for super-resolution, and 0.5 for compressive sensing.

**FPS.** The DDIM hyperparameter $c$ is fixed as $\sqrt{c} = 0.85$ for all tasks.

**SGS-EDM.** We use 1-step DDIM (i.e., Tweedie's formula) in the prior step.

**Latent-DPS.** The step-size is set as $\zeta_{t_i} = \overline{\alpha}_{t_i} \cdot \zeta$, with $\zeta = 0.01$ for deblurring, $\zeta = 0.5$ for inpainting and super-resolution, and $\zeta = 0.005$ for compressive sensing.

**PSLD.** We fix $\gamma_{t_i} = 0.1\eta_{t_i}$, and set $\eta_{t_i}$ as 8.0 for deblurring, 10.0 for inpainting, 25.0 for super-resolution, and 9.0 for compressive sensing.

**STSL.** We fix $\gamma = 0.02$ and set the number of Monte Carlo samples to $n = 1$. $\eta_{t_i}$ is chosen as 0.01 for deblurring, 0.02 for inpainting, 0.1 for super-resolution, and 0.015 for compressive sensing.

**CMInversion.** We fix the correction steps $K = 5$. The step-size $\zeta$ is set to 0.3 for deblurring, 1.0 for inpainting, 3.0 for super-resolution, and 0.095 for compressive sensing.

In this paper, we default to sampling evenly spaced timesteps in $[0, T]$. For all algorithms, we initialize from standard Gaussian noise. Specifically for RED-diff, this is equivalent to using the random initialization method proposed in [13], which provides better stability. Starting from intermediate steps like the original setting of [7] does not affect the applicability of our method, as discussed in Appendix A.

### G.3 Details of LLE training

For the training of LLE, we generate $N = 50$ reference samples using 999-step DDIM sampler with the corresponding diffusion model. We fix the weight of PSNR and LPIPS as $\omega = 0.1$. The Schedule-free AdamW [45] is employed with the epochs set to 100 at each timestep $t_i$ and warmup steps set to 50. Gradients are calculated directly using the full batch. In most cases, the learning rate is set to $0.04/S$, where $S$ denotes the total number of steps used by the inverse algorithm. For ReSample and DAPS on the nonlinear deblurring task on the FFHQ dataset, we adopt a dynamic learning rate that at timestep $t_i$, the learning rate is set to $0.2\overline{\alpha}_{t_{i+1}}/S$. It is worth noting that we have conducted minimal learning rate tuning for different algorithms, tasks, and steps. Such general learning rates have already yielded satisfactory performance. In practical applications, further tuning of learning rates and other optimization hyperparameters could potentially further enhance the performance of LLE.

We design an adaptive initialization method for the learnable coefficients at each timestep. Before the learning stage at timestep $t_i$, we compute the loss for $\hat{\mathbf{x}}_{0,t_i}^{(n)}$ and $\tilde{\mathbf{x}}_{0,t_{i+1}}^{(n)}, n = 1, \ldots, N$, and determine the initialization strategy accordingly. For linear inverse problems, if

$$\mathbb{E}_{n \sim \mathcal{U}\{1,\ldots,N\}} \mathcal{L}\left(\tilde{\mathbf{x}}_{0,t_{i+1}}^{(n)}, \mathbf{x}_0^{(n)}\right) \geq \mathbb{E}_{n \sim \mathcal{U}\{1,\ldots,N\}} \mathcal{L}\left(\hat{\mathbf{x}}_{0,t_i}^{(n)}, \mathbf{x}_0^{(n)}\right), \tag{99}$$

then we initialize as

$$\gamma_{t_i, S-i}^{\|} = \gamma_{t_i, S-i}^{\perp} = 1. \tag{100}$$

Otherwise, we initialize as:

$$\gamma_{t_i, S-i-1}^{\|} = \gamma_{t_i, S-i-1}^{\perp} = 1. \tag{101}$$

All other parameters are randomly sampled from $\mathcal{N}(0, 10^{-6})$. Intuitively, this initialization allows the coefficients to initial with a smaller loss, thereby improving optimization efficiency.

For nonlinear inverse problems, we further introduce a soft initialization to prevent the algorithm from prematurely converging to local optima. Similarly, if

$$\mathbb{E}_{n \sim \mathcal{U}\{1,\ldots,N\}} \mathcal{L}\left(\tilde{\mathbf{x}}_{0,t_{i+1}}^{(n)}, \mathbf{x}_0^{(n)}\right) \geq \mathbb{E}_{n \sim \mathcal{U}\{1,\ldots,N\}} \mathcal{L}\left(\hat{\mathbf{x}}_{0,t_i}^{(n)}, \mathbf{x}_0^{(n)}\right), \tag{102}$$

we initialize as

$$\gamma_{t_i, S-i} = 1. \tag{103}$$

Otherwise, we initialize as:

$$\gamma_{t_i, S-i-1} = \overline{\alpha}_{t_i}, \quad \gamma_{t_i, S-i} = 1 - \overline{\alpha}_{t_i}. \tag{104}$$

All other coefficients are randomly sampled from $\mathcal{N}(0, 10^{-6})$. This approach allows LLE to follow the original trajectory in earlier timesteps, while searching for better linear combination coefficients more aggressively in later timesteps.

## H   Complete quantitive results

Here, we present the complete results of all algorithms on the CelebA-HQ and FFHQ datasets using a total of 3, 4, 5, 7, 10, and 15 steps. PSNR, SSIM, LPIPS, and FID are reported on both noiseless and noisy tasks. The table index is illustrated in Table 19 for convenience.

Table 19: Index of tables.

|  | DDNM | DDRM | ΠGDM | DMPS | RED-diff | DiffPIR | DPS | ReSample | DAPS |
|---|---|---|---|---|---|---|---|---|---|
| CelebA-HQ | Table 20 | Table 22 | Table 24 | Table 26 | Table 30 | Table 32 | Table 28 | Table 34 | Table 36 |
| FFHQ | Table 21 | Table 23 | Table 25 | Table 27 | Table 31 | Table 33 | Table 29 | Table 35 | Table 37 |

Table 20: Results of DDNM on CelebA-HQ Dataset.

| Condition | Steps | Strategy | Deblur (aniso) | Inpainting | 4× SR | CS 50% |
|---|---|---|---|---|---|---|
| | | | PSNR↑/SSIM↑/LPIPS↓/FID↓ | | | |
| $\sigma_{\mathbf{y}} = 0.05$ | 3 | - | 27.80 / 0.7577 / 0.3192 / 53.22 | 16.64 / 0.4424 / 0.4923 / 73.29 | 27.09 / **0.7732** / **0.2961** / 60.81 | 16.55 / 0.4415 / 0.5394 / **104.7** |
| | | LLE | **28.08 / 0.7842 / 0.2909 / 43.53** | **24.38 / 0.5523 / 0.4329 / 64.67** | **27.84** / 0.7702 / 0.2989 / **46.29** | **17.29 / 0.4731 / 0.5197** / 155.4 |
| | 4 | - | 28.98 / 0.7948 / 0.2845 / 47.36 | 20.27 / 0.5102 / 0.4568 / 75.48 | 28.35 / **0.8034** / 0.2758 / 59.34 | 17.39 / 0.4464 / 0.5179 / **87.57** |
| | | LLE | **29.27 / 0.8172 / 0.2558 / 42.83** | **25.29 / 0.5915 / 0.4073 / 61.19** | **28.65** / 0.7921 / **0.2700 / 39.87** | **18.43 / 0.4994 / 0.4742** / 96.63 |
| | 5 | - | 29.63 / 0.8192 / 0.2590 / 45.43 | 22.76 / 0.5501 / 0.4310 / 73.25 | 28.97 / **0.8182** / 0.2619 / 58.20 | 18.20 / 0.4742 / 0.4914 / 77.15 |
| | | LLE | **29.82 / 0.8307 / 0.2394 / 45.10** | **26.35 / 0.6586 / 0.3661 / 54.84** | **29.02** / 0.8056 / **0.2522 / 37.16** | **19.41 / 0.5358 / 0.4410 / 74.41** |
| | 7 | - | 30.19 / 0.8377 / 0.2377 / 50.72 | 25.93 / 0.6269 / 0.3880 / 68.29 | 29.52 / **0.8325** / 0.2418 / 54.07 | 20.28 / 0.5536 / 0.4344 / 65.31 |
| | | LLE | **30.29 / 0.8409 / 0.2247 / 44.66** | **27.69 / 0.7188 / 0.3279 / 50.63** | **29.33** / 0.8177 / **0.2333 / 35.57** | **21.32 / 0.6084 / 0.3852 / 54.22** |
| | 10 | - | 30.45 / 0.8446 / 0.2274 / 51.12 | 28.19 / 0.7163 / 0.3325 / 60.74 | 29.07 / **0.8210** / 0.2410 / 51.61 | 21.30 / 0.6356 / 0.3792 / 55.50 |
| | | LLE | **30.50 / 0.8454 / 0.2137 / 42.71** | **29.13 / 0.7886 / 0.2789 / 45.50** | **29.08** / 0.8075 / **0.2353 / 33.47** | **22.42 / 0.6738 / 0.3408 / 44.97** |
| | 15 | - | 30.50 / 0.8461 / 0.2165 / 47.03 | 30.57 / 0.8206 / 0.2589 / 46.30 | **29.39 / 0.8289** / 0.2274 / 38.77 | 23.39 / 0.7475 / 0.2961 / 38.77 |
| | | LLE | **30.52 / 0.8463 / 0.2058 / 40.68** | **30.74 / 0.8540 / 0.2281 / 38.23** | 29.17 / 0.8120 / **0.2266 / 33.42** | **24.04 / 0.7687 / 0.2706 / 31.03** |
| $\sigma_{\mathbf{y}} = 0$ | 3 | - | 39.44 / 0.9577 / 0.1004 / **11.60** | 19.39 / 0.4237 / 0.5946 / 137.9 | 30.20 / 0.8603 / 0.2413 / 60.50 | 17.27 / **0.5188** / 0.5521 / 252.3 |
| | | LLE | **39.56 / 0.9588 / 0.0986** / 11.72 | **26.31 / 0.7180 / 0.3790 / 78.34** | **30.29 / 0.8628 / 0.2309 / 58.37** | **17.69** / 0.5185 / **0.5385 / 229.1** |
| | 4 | - | 39.80 / 0.9608 / 0.0931 / **9.790** | 24.11 / 0.6077 / 0.4427 / 88.72 | **30.76 / 0.8728** / 0.2043 / 51.84 | 18.16 / 0.5751 / 0.4969 / 187.6 |
| | | LLE | **39.97 / 0.9623 / 0.0914** / 10.01 | **29.03 / 0.8419 / 0.2840 / 67.03** | 30.71 / 0.8710 / **0.1979 / 48.91** | **18.96 / 0.5840 / 0.4695 / 154.7** |
| | 5 | - | 39.99 / 0.9623 / 0.0897 / 8.810 | 27.83 / 0.7580 / 0.3322 / 66.63 | **30.97 / 0.8766** / 0.1876 / 46.19 | 18.98 / 0.6191 / 0.4496 / 149.1 |
| | | LLE | **40.20 / 0.9640 / 0.0865 / 8.770** | **30.93 / 0.8868 / 0.2206 / 52.78** | 30.90 / 0.8743 / **0.1823 / 43.52** | **20.15 / 0.6395 / 0.4147 / 118.1** |
| | 7 | - | 40.24 / 0.9643 / 0.0846 / 7.480 | 26.20 / 0.6481 / 0.3722 / 55.35 | **31.17 / 0.8802** / 0.1700 / 39.58 | 21.64 / 0.7143 / 0.3512 / 91.41 |
| | | LLE | **40.47 / 0.9662 / 0.0815 / 7.390** | **32.56 / 0.9159 / 0.1688 / 38.79** | 31.04 / 0.8772 / **0.1663 / 38.15** | **22.43 / 0.7270 / 0.3275 / 77.24** |
| | 10 | - | 40.42 / 0.9656 / 0.0805 / 6.650 | 33.54 / 0.9282 / 0.1440 / 30.86 | **31.26 / 0.8818** / 0.1583 / 34.82 | 22.68 / 0.7591 / 0.2992 / 68.94 |
| | | LLE | **40.70 / 0.9677 / 0.0773 / 6.500** | **33.63 / 0.9294 / 0.1386 / 29.31** | 31.16 / 0.8792 / **0.1564 / 33.90** | **23.81 / 0.7744 / 0.2773 / 57.84** |
| | 15 | - | 40.55 / 0.9667 / 0.0776 / 5.880 | 34.57 / 0.9402 / 0.1122 / 22.14 | **31.32 / 0.8828** / 0.1506 / 30.93 | 25.25 / 0.8213 / 0.2347 / 43.85 |
| | | LLE | **40.78 / 0.9684 / 0.0748 / 5.830** | 34.55 / 0.9398 / 0.1125 / **21.95** | 31.28 / 0.8812 / 0.1507 / **30.79** | **26.27 / 0.8316 / 0.2232 / 38.99** |

Table 21: Results of DDNM on FFHQ Dataset.

| Condition | Steps | Strategy | Deblur (aniso) | Inpainting | 4× SR | CS 50% |
|---|---|---|---|---|---|---|
| | | | PSNR↑/SSIM↑/LPIPS↓/FID↓ | | | |
| $\sigma_{\mathbf{y}} = 0.05$ | 3 | - | 25.89 / 0.6341 / 0.4177 / 139.1 | 16.80 / 0.3580 / 0.5557 / 167.9 | 25.78 / 0.7327 / 0.3314 / 122.7 | 16.27 / 0.3361 / 0.6475 / **227.6** |
| | | LLE | **26.34 / 0.6859 / 0.3836 / 127.7** | **22.56 / 0.4243 / 0.5012 / 145.0** | **26.29 / 0.7548 / 0.3012 / 107.6** | **16.84 / 0.4100 / 0.5900** / 280.0 |
| | 4 | - | 26.98 / 0.6671 / 0.3935 / 134.8 | 19.59 / 0.3689 / 0.5569 / 156.1 | 27.11 / 0.7704 / 0.3086 / 120.7 | 16.72 / 0.3048 / 0.6335 / **189.2** |
| | | LLE | **27.88 / 0.7610 / 0.3226 / 113.8** | **22.51 / 0.4773 / 0.5051 / 155.1** | **27.35 / 0.7814 / 0.2810 / 100.1** | **17.61 / 0.4246 / 0.5671** / 235.0 |
| | 5 | - | 28.02 / 0.7290 / 0.3526 / 126.5 | 21.30 / 0.3804 / 0.5396 / 150.5 | 27.83 / 0.7906 / 0.2924 / 116.3 | 17.27 / 0.3214 / 0.6059 / **167.0** |
| | | LLE | **28.72 / 0.8014 / 0.2802 / 98.94** | **23.65 / 0.5060 / 0.4737 / 133.8** | **27.96 / 0.7977 / 0.2650 / 96.46** | **18.24 / 0.4430 / 0.5400** / 202.4 |
| | 7 | - | 29.18 / 0.8041 / 0.2859 / 102.9 | 23.47 / 0.4379 / 0.4955 / 144.2 | **28.58 / 0.8156 / 0.2679** / 106.0 | 18.45 / 0.3801 / 0.5531 / **156.1** |
| | | LLE | **29.42 / 0.8285 / 0.2490 / 93.16** | **24.57 / 0.5386 / 0.4461 / 124.4** | 28.41 / 0.8153 / **0.2403 / 84.21** | **19.02 / 0.4635 / 0.5100** / 171.3 |
| | 10 | - | **29.79 / 0.8357** / 0.2497 / 102.9 | 25.62 / 0.5417 / 0.4336 / 134.7 | **28.08** / 0.8005 / 0.2767 / 108.8 | 19.43 / 0.4732 / 0.4859 / 145.4 |
| | | LLE | 29.75 / **0.8387 / 0.2317 / 88.46** | **26.84 / 0.6701 / 0.3602 / 114.7** | 27.89 / 0.8003 / **0.2509 / 87.98** | **20.33 / 0.5701 / 0.4212 / 127.5** |
| | 15 | - | **29.98 / 0.8432** / 0.2379 / 96.10 | 28.61 / 0.7061 / 0.3376 / 116.9 | **28.64 / 0.8168** / 0.2566 / 99.74 | 20.96 / 0.6274 / 0.3868 / 123.8 |
| | | LLE | 29.87 / 0.8417 / **0.2207 / 81.90** | **28.45 / 0.7891 / 0.2809 / 105.1** | 28.27 / 0.8114 / **0.2346 / 81.65** | **21.25 / 0.6919 / 0.3325 / 101.9** |
| $\sigma_{\mathbf{y}} = 0$ | 3 | - | 39.02 / 0.9572 / 0.1035 / **28.09** | 19.31 / 0.4262 / 0.6029 / 190.9 | 28.78 / 0.8094 / 0.3083 / **110.8** | 17.12 / **0.5089** / 0.5750 / 342.3 |
| | | LLE | **39.18 / 0.9590 / 0.1013** / 28.74 | **24.74 / 0.6379 / 0.4292 / 141.7** | **29.14 / 0.8382 / 0.2832** / 111.0 | **17.40** / 0.5075 / **0.5666 / 326.1** |
| | 4 | - | 39.54 / 0.9618 / 0.0917 / **22.70** | 23.21 / 0.5873 / 0.4732 / 152.9 | **29.69 / 0.8549 / 0.2407** / 104.2 | 17.55 / 0.5514 / 0.5308 / 288.4 |
| | | LLE | **39.75 / 0.9637 / 0.0899** / 23.41 | **27.35 / 0.7873 / 0.3474 / 129.1** | 29.66 / 0.8551 / 0.2309 / **97.61** | **18.19 / 0.5557 / 0.5136 / 253.2** |
| | 5 | - | 39.82 / 0.9638 / 0.0871 / **20.42** | 26.34 / 0.7190 / 0.3798 / 130.3 | **29.99 / 0.8646** / 0.2107 / 92.35 | 18.29 / 0.5912 / 0.4875 / 246.0 |
| | | LLE | **40.08 / 0.9662 / 0.0843** / 20.56 | **28.71 / 0.8384 / 0.2929 / 113.8** | 29.94 / 0.8627 / **0.2060 / 90.07** | **19.21 / 0.6046 / 0.4626 / 208.0** |
| | 7 | - | 40.16 / 0.9661 / 0.0803 / **16.69** | 29.96 / 0.8636 / 0.2605 / 100.2 | **30.30 / 0.8723** / 0.1825 / 76.87 | 19.99 / 0.6653 / 0.4097 / 179.3 |
| | | LLE | **40.57 / 0.9697 / 0.0763** / 17.04 | **30.49 / 0.8827 / 0.2331 / 91.22** | 30.23 / 0.8700 / **0.1802 / 75.35** | **20.42 / 0.6671 / 0.3959 / 166.6** |
| | 10 | - | 40.56 / 0.9691 / 0.074 / **13.78** | 31.75 / 0.9049 / 0.1956 / 74.17 | **30.50 / 0.8774** / 0.1657 / 64.98 | 20.61 / 0.7103 / 0.3554 / 152.5 |
| | | LLE | **40.85 / 0.9713 / 0.0708** / 14.16 | **31.80 / 0.9072 / 0.1910 / 73.11** | 30.44 / 0.8754 / **0.1645 / 63.59** | **21.45 / 0.7174 / 0.3424 / 137.4** |
| | 15 | - | 40.89 / 0.9709 / 0.0691 / **11.69** | 33.08 / 0.9266 / 0.1487 / 55.23 | 30.69 / **0.8813** / 0.1527 / 54.18 | 22.89 / 0.7816 / 0.2812 / 106.6 |
| | | LLE | **41.14 / 0.9727 / 0.0657** / 11.95 | 33.05 / 0.9264 / **0.1481 / 53.90** | 30.69 / 0.8806 / **0.1518 / 52.02** | **23.30 / 0.7820 / 0.2744 / 97.63** |

Table 22: Results of DDRM on CelebA-HQ Dataset.

| Condition | Steps | Strategy | PSNR↑/SSIM↑/LPIPS↓/FID↓ | | | |
|---|---|---|---|---|---|---|
| | | | Deblur (aniso) | Inpainting | 4× SR | CS 50% |
| $\sigma_{\mathbf{y}} = 0.05$ | 3 | - | 27.68 / 0.7950 / 0.2773 / 56.48 | 16.68 / 0.4894 / 0.4403 / **54.28** | 26.71 / **0.7639** / **0.2772** / 45.88 | 16.58 / 0.4953 / 0.4776 / **90.78** |
| | | LLE | **27.69** / **0.7951** / **0.2713** / **52.88** | **24.53** / **0.6251** / **0.4061** / 64.05 | **27.49** / 0.7614 / 0.2949 / **41.97** | **17.07** / **0.5269** / **0.4698** / 111.6 |
| | 4 | - | 28.75 / 0.8159 / 0.2561 / 54.39 | 20.48 / 0.6418 / 0.3332 / 42.93 | 27.95 / **0.7966** / **0.2603** / 52.16 | 17.46 / 0.5547 / 0.4254 / 80.66 |
| | | LLE | **28.80** / **0.8168** / **0.2469** / **48.18** | **25.01** / **0.7515** / **0.2935** / **40.65** | **28.25** / 0.7830 / 0.2645 / **36.06** | **18.32** / **0.5857** / **0.4048** / **70.41** |
| | 5 | - | 29.29 / 0.8258 / 0.2444 / 52.28 | 23.21 / 0.7165 / 0.2860 / **44.73** | 28.58 / **0.8114** / **0.2472** / 51.07 | 18.34 / 0.5985 / 0.3886 / 71.34 |
| | | LLE | **29.36** / **0.8265** / **0.2349** / **45.93** | **26.09** / **0.7801** / **0.2632** / 45.02 | **28.64** / 0.7955 / 0.2486 / **33.65** | **19.41** / **0.6320** / **0.3630** / **53.99** |
| | 7 | - | 29.82 / 0.8351 / 0.2315 / 49.98 | 26.85 / 0.7957 / 0.2506 / 47.03 | **29.28** / **0.8277** / **0.2302** / 48.74 | 20.47 / 0.6769 / 0.3292 / 60.04 |
| | | LLE | **29.91** / **0.8365** / **0.2222** / **44.17** | **28.18** / **0.8237** / **0.2304** / **43.39** | 29.08 / 0.8105 / 0.2317 / **32.77** | **21.21** / **0.6951** / **0.3090** / **40.95** |
| | 10 | - | 30.15 / 0.8411 / 0.2213 / 47.72 | 28.97 / 0.8351 / 0.2209 / 45.04 | 28.91 / **0.8181** / **0.2340** / 48.62 | 21.36 / 0.7135 / 0.3007 / 56.08 |
| | | LLE | **30.21** / **0.8411** / **0.2119** / **41.27** | **29.70** / **0.8507** / **0.2099** / **41.61** | **29.00** / 0.8043 / 0.2345 / **32.41** | **22.40** / **0.7369** / **0.2769** / **38.51** |
| | 15 | - | 30.31 / 0.8436 / 0.2141 / 45.11 | 30.69 / 0.8678 / 0.1986 / 41.94 | **29.27** / **0.8265** / **0.2232** / 45.37 | 23.05 / 0.7637 / 0.2632 / 48.76 |
| | | LLE | **30.35** / **0.8440** / **0.2075** / **40.39** | **30.95** / **0.8737** / **0.1901** / **39.44** | 29.11 / 0.8094 / 0.2270 / **33.30** | **24.04** / **0.7782** / **0.2472** / **34.88** |
| $\sigma_{\mathbf{y}} = 0$ | 3 | - | **38.20** / 0.9460 / 0.1351 / 19.74 | 19.40 / 0.4282 / 0.5959 / 137.4 | 30.16 / 0.8584 / 0.2429 / 62.40 | 17.25 / **0.5192** / 0.5508 / 242.5 |
| | | LLE | 38.18 / **0.9493** / **0.1090** / **14.32** | **25.99** / **0.7057** / **0.3885** / **80.58** | **30.27** / **0.8619** / **0.2327** / **60.90** | **17.64** / 0.5184 / **0.5361** / 221.6 |
| | 4 | - | **38.45** / 0.9488 / 0.1305 / 18.47 | 23.26 / 0.5808 / 0.4700 / 98.01 | **30.62** / **0.8694** / 0.2153 / 57.49 | 18.08 / 0.5696 / 0.5040 / 188.3 |
| | | LLE | 38.30 / **0.9506** / **0.1092** / **15.37** | **28.01** / **0.8199** / **0.3148** / **78.35** | 30.62 / 0.8689 / **0.2072** / **54.45** | **18.91** / **0.5809** / **0.4740** / **154.5** |
| | 5 | - | **38.57** / 0.9501 / 0.1281 / 17.89 | 26.09 / 0.6997 / 0.3819 / 81.43 | **30.83** / **0.8731** / 0.2005 / 52.88 | 18.85 / 0.6065 / 0.4658 / 156.8 |
| | | LLE | 38.51 / **0.9520** / **0.1064** / **14.63** | **29.74** / **0.8657** / **0.2536** / **66.80** | 30.78 / 0.8719 / **0.1937** / **50.24** | **19.97** / **0.6287** / **0.4275** / **124.0** |
| | 7 | - | 38.75 / 0.9520 / 0.1248 / 16.89 | 29.72 / 0.8506 / 0.2645 / 63.44 | **31.02** / **0.8766** / 0.1836 / 47.48 | 21.01 / 0.6851 / 0.3853 / 109.3 |
| | | LLE | **38.85** / **0.9555** / **0.1009** / **13.20** | **31.33** / **0.8975** / **0.2002** / **53.40** | 30.88 / 0.8738 / **0.1776** / **45.17** | **21.88** / **0.7041** / **0.3518** / **89.23** |
| | 10 | - | 38.89 / 0.9535 / 0.1221 / 16.11 | 31.70 / 0.8974 / 0.1977 / 48.81 | **31.13** / **0.8785** / 0.1723 / 43.07 | 21.77 / 0.7239 / 0.3402 / 89.77 |
| | | LLE | **38.97** / **0.9563** / **0.0993** / **12.67** | **32.45** / **0.9138** / **0.1642** / **41.39** | 30.94 / 0.8750 / **0.1676** / **41.16** | **22.83** / **0.7433** / **0.3093** / **73.65** |
| | 15 | - | 39.01 / 0.9548 / 0.1193 / 15.56 | 33.22 / 0.9233 / 0.1439 / 35.08 | **31.20** / **0.8799** / 0.1638 / 39.22 | 23.58 / 0.7785 / 0.2810 / 67.44 |
| | | LLE | **39.07** / **0.9571** / **0.0983** / **12.65** | **33.41** / **0.9256** / **0.1361** / **32.66** | 30.96 / 0.8758 / **0.1609** / **38.41** | **24.58** / **0.7923** / **0.2610** / **57.46** |

Table 23: Results of DDRM on FFHQ Dataset.

| Condition | Steps | Strategy | PSNR↑/SSIM↑/LPIPS↓/FID↓ | | | |
|---|---|---|---|---|---|---|
| | | | Deblur (aniso) | Inpainting | 4× SR | CS 50% |
| $\sigma_{\mathbf{y}} = 0.05$ | 3 | - | **26.38** / **0.7630** / 0.3048 / 113.2 | 16.89 / 0.4327 / 0.5204 / 160.3 | 25.27 / 0.6956 / 0.3481 / 105.8 | 16.44 / 0.4806 / **0.5110** / 185.5 |
| | | LLE | 26.35 / 0.7656 / **0.2972** / **105.2** | **23.63** / **0.5403** / **0.4618** / **142.1** | **25.90** / **0.7364** / **0.3107** / **101.5** | **17.01** / **0.5104** / 0.5255 / 228.4 |
| | 4 | - | **27.62** / 0.7933 / 0.2796 / 105.5 | 20.32 / 0.5984 / 0.4063 / 129.0 | 26.69 / 0.7622 / 0.2912 / 101.2 | 17.16 / 0.5367 / 0.4583 / 152.4 |
| | | LLE | 27.59 / **0.7947** / **0.2686** / **96.91** | **24.89** / **0.7194** / **0.3636** / **124.5** | **26.83** / **0.7689** / **0.2755** / **90.59** | **17.82** / **0.5662** / **0.4381** / **145.2** |
| | 5 | - | **28.28** / 0.8083 / 0.2667 / 101.3 | 22.66 / 0.6809 / 0.3447 / 110.5 | **27.43** / 0.7853 / 0.2708 / 94.13 | 17.62 / 0.5713 / 0.4244 / 142.8 |
| | | LLE | 28.27 / **0.8098** / **0.2540** / **90.18** | **25.09** / **0.7535** / **0.3078** / **103.8** | 27.41 / **0.7870** / **0.2564** / **86.64** | **18.57** / **0.6031** / **0.4001** / **125.7** |
| | 7 | - | **29.02** / **0.8243** / 0.2510 / 94.13 | 25.82 / 0.7682 / 0.2736 / 100.0 | **28.32** / **0.8100** / 0.2497 / 91.05 | 19.11 / 0.6373 / 0.3692 / 121.2 |
| | | LLE | 28.95 / 0.8237 / **0.2382** / **86.66** | **26.86** / **0.7958** / **0.2620** / **94.19** | 27.99 / 0.8051 / **0.2347** / **77.00** | **19.76** / **0.6508** / **0.3565** / **106.2** |
| | 10 | - | **29.51** / **0.8344** / 0.2406 / 88.23 | 27.83 / 0.8134 / 0.2360 / 89.04 | **27.91** / **0.7974** / 0.2636 / 99.40 | 19.78 / 0.6701 / 0.3392 / 115.5 |
| | | LLE | 29.42 / 0.8331 / **0.2276** / **79.88** | **28.41** / **0.8305** / **0.2297** / **86.32** | 27.68 / 0.7953 / **0.2448** / **84.89** | **20.46** / **0.6874** / **0.3197** / **98.19** |
| | 15 | - | **29.79** / **0.8405** / 0.2339 / 87.79 | 29.75 / 0.8553 / 0.2027 / 78.33 | **28.49** / **0.8138** / 0.2472 / 90.30 | 20.84 / 0.7226 / 0.2930 / 97.31 |
| | | LLE | 29.68 / 0.8385 / **0.2215** / **78.71** | **29.97** / **0.8634** / **0.1956** / **70.99** | 28.09 / 0.8074 / **0.2315** / **78.48** | **21.59** / **0.7343** / **0.2824** / **87.52** |
| $\sigma_{\mathbf{y}} = 0$ | 3 | - | **37.53** / **0.9377** / 0.1536 / 41.15 | 19.36 / 0.4288 / 0.6010 / 191.8 | 28.84 / 0.8137 / 0.3057 / **112.2** | 17.02 / 0.5058 / 0.5759 / 351.5 |
| | | LLE | 37.07 / 0.9369 / **0.1265** / **33.44** | **24.63** / **0.6309** / **0.4317** / **143.1** | **29.08** / **0.8367** / **0.2855** / 115.6 | **17.30** / **0.5068** / **0.5670** / **332.6** |
| | 4 | - | **37.88** / **0.9419** / 0.1478 / 38.74 | 22.67 / 0.5637 / 0.4968 / 158.9 | 29.58 / 0.8527 / 0.2542 / 110.4 | 17.70 / 0.5525 / 0.5345 / 290.5 |
| | | LLE | 37.36 / 0.9409 / **0.1216** / **34.50** | **26.89** / **0.7763** / **0.3610** / **133.7** | **29.62** / **0.8543** / **0.2537** / **109.0** | **18.17** / **0.5529** / **0.5166** / **255.6** |
| | 5 | - | **38.04** / **0.9437** / 0.1450 / 36.86 | 25.05 / 0.6661 / 0.4236 / 140.8 | 29.84 / **0.8614** / 0.2286 / 101.0 | 18.22 / 0.5826 / 0.5022 / 253.0 |
| | | LLE | 37.62 / **0.9445** / **0.1168** / **32.09** | **27.77** / **0.8126** / **0.3256** / **125.9** | **29.86** / 0.8604 / **0.2310** / **99.36** | **18.93** / **0.5906** / **0.4778** / **223.7** |
| | 7 | - | **38.30** / 0.9463 / 0.1405 / 34.06 | 27.98 / 0.7999 / 0.3295 / 121.4 | **30.13** / **0.8687** / 0.2017 / 89.48 | 19.17 / 0.6344 / 0.4473 / 207.5 |
| | | LLE | 37.99 / **0.9488** / **0.1085** / **28.14** | **29.35** / **0.8575** / **0.2753** / **107.9** | 30.08 / 0.8679 / **0.2018** / 89.76 | **19.97** / **0.6474** / **0.4191** / **176.6** |
| | 10 | - | **38.53** / 0.9485 / 0.1367 / 32.84 | 29.95 / 0.8623 / 0.2651 / 102.3 | **30.40** / **0.8742** / 0.1817 / 77.48 | 20.16 / 0.6809 / 0.3981 / 172.6 |
| | | LLE | 38.18 / **0.9510** / **0.1047** / **26.50** | **30.61** / **0.8869** / **0.2321** / **92.88** | 30.25 / 0.8723 / **0.1810** / **76.35** | **20.88** / **0.6890** / **0.3727** / **153.9** |
| | 15 | - | **38.77** / 0.9506 / 0.1331 / 30.91 | 31.65 / 0.9027 / 0.2019 / 81.93 | **30.64** / **0.8797** / 0.1673 / 66.16 | 21.36 / 0.7302 / 0.3386 / 141.0 |
| | | LLE | 38.41 / **0.9541** / **0.1006** / **25.90** | **31.82** / **0.9087** / **0.1900** / **75.74** | 30.34 / 0.8761 / **0.1648** / **64.45** | **22.20** / **0.7394** / **0.3218** / **124.8** |

Table 24: Results of ΠGDM on CelebA-HQ Dataset.

| Condition | Steps | Strategy | PSNR↑/SSIM↑/LPIPS↓/FID↓ | | | |
| --- | --- | --- | --- | --- | --- | --- |
| | | | Deblur (aniso) | Inpainting | 4× SR | CS 50% |
| $\sigma_\mathbf{y} = 0.05$ | 3 | - | 20.39 / 0.5361 / 0.5197 / **105.3** | 19.57 / 0.5333 / 0.4947 / 116.5 | 20.73 / 0.5519 / 0.5000 / **104.5** | 16.79 / 0.4721 / 0.5546 / **114.8** |
| | | LLE | 21.73 / 0.5877 / 0.4894 / 115.0 | 20.72 / 0.5712 / 0.4829 / 109.4 | 21.54 / 0.5833 / 0.4896 / 119.7 | 17.30 / 0.4871 / 0.5396 / 126.4 |
| | 4 | - | 19.80 / 0.5735 / 0.5076 / 87.53 | 21.85 / 0.6390 / 0.4126 / 84.91 | 19.94 / 0.5874 / 0.4743 / **76.88** | 16.47 / 0.5005 / 0.5650 / 91.76 |
| | | LLE | 24.14 / 0.6780 / 0.4062 / 79.73 | 23.61 / 0.6805 / 0.3808 / 81.92 | 24.09 / 0.6793 / 0.3973 / 82.33 | 18.71 / 0.5615 / 0.4722 / 88.50 |
| | 5 | - | 22.46 / 0.6530 / 0.4410 / 71.32 | 22.63 / 0.6945 / 0.3761 / 73.49 | 21.93 / 0.6620 / 0.4157 / **65.21** | 16.32 / 0.5179 / 0.5407 / 85.88 |
| | | LLE | 25.95 / 0.7339 / 0.3473 / 64.37 | 25.21 / 0.7349 / 0.3358 / 70.90 | 25.67 / 0.7308 / 0.3382 / 66.20 | 19.33 / 0.5914 / 0.4480 / 80.27 |
| | 7 | - | 25.75 / 0.7411 / 0.3284 / **50.28** | 26.47 / 0.7827 / 0.3012 / 62.67 | 24.78 / 0.7349 / 0.3304 / 56.29 | 15.05 / 0.4831 / 0.5576 / 107.2 |
| | | LLE | 28.29 / 0.7956 / 0.2724 / 52.73 | 28.02 / 0.8050 / 0.2730 / 55.70 | 27.70 / 0.7851 / 0.2715 / 48.56 | 21.07 / 0.6743 / 0.3682 / 63.42 |
| | 10 | - | 28.69 / 0.8064 / 0.2511 / 48.65 | 29.98 / 0.8458 / 0.2477 / 55.63 | 27.84 / 0.7993 / 0.2600 / 51.35 | 15.50 / 0.5215 / 0.5031 / 106.4 |
| | | LLE | 29.28 / 0.8179 / 0.2312 / 43.38 | 30.15 / 0.8488 / 0.2388 / 50.99 | 28.64 / 0.8098 / 0.2358 / 49.96 | 22.10 / 0.7275 / 0.3210 / 55.16 |
| | 15 | - | 29.57 / 0.8265 / 0.2260 / 46.79 | 31.69 / 0.8761 / 0.2211 / 51.18 | **28.96** / **0.8199** / 0.2356 / 48.03 | 16.12 / 0.5698 / 0.4458 / 99.84 |
| | | LLE | 29.80 / 0.8293 / 0.2134 / 42.07 | 31.77 / 0.8783 / 0.2136 / 47.69 | 28.84 / 0.8145 / 0.2230 / 44.47 | 23.89 / 0.7831 / 0.2718 / 49.05 |
| $\sigma_\mathbf{y} = 0$ | 3 | - | 30.11 / 0.8743 / 0.2294 / 37.23 | 24.41 / 0.7327 / 0.3773 / 96.26 | 26.35 / 0.7537 / 0.4303 / 83.16 | 18.24 / 0.5220 / 0.5807 / 213.2 |
| | | LLE | 33.83 / 0.9179 / 0.1733 / 31.16 | 25.57 / 0.7843 / 0.3228 / 79.60 | 27.71 / 0.7988 / 0.3578 / 73.76 | 18.54 / 0.5507 / 0.5414 / 181.7 |
| | 4 | - | 29.49 / 0.8869 / 0.2219 / 36.99 | 26.36 / 0.8115 / 0.3054 / 81.42 | 26.11 / 0.7698 / 0.3910 / 76.90 | 18.00 / 0.5318 / 0.5774 / 195.4 |
| | | LLE | 34.78 / 0.9313 / 0.1534 / 27.68 | 28.03 / 0.8450 / 0.2539 / 67.84 | 28.61 / 0.8254 / 0.2944 / 68.51 | 19.72 / 0.6157 / 0.4733 / 128.5 |
| | 5 | - | 32.93 / 0.9219 / 0.1697 / 26.36 | 27.12 / 0.8365 / 0.2734 / 72.99 | 27.78 / 0.8077 / 0.3293 / 65.38 | 17.16 / 0.5269 / 0.5723 / 196.6 |
| | | LLE | 35.38 / 0.9377 / 0.1397 / 26.30 | 29.47 / 0.8712 / 0.2196 / 60.17 | 29.47 / 0.8441 / 0.2602 / 62.04 | 20.18 / 0.6389 / 0.4475 / 118.2 |
| | 7 | - | 35.98 / 0.9437 / 0.1293 / **20.86** | 30.49 / 0.8899 / 0.1983 / 51.95 | 29.44 / 0.8446 / 0.2499 / 53.66 | 15.53 / 0.4954 / 0.5831 / 218.7 |
| | | LLE | 36.80 / 0.9505 / 0.1142 / 24.89 | 31.76 / 0.9038 / 0.1773 / 47.25 | 30.75 / 0.8709 / 0.2095 / 51.95 | 21.70 / 0.7147 / 0.3571 / 82.28 |
| | 10 | - | 37.87 / 0.9573 / 0.1015 / **21.19** | 33.32 / 0.9209 / 0.1546 / 39.95 | 30.65 / 0.8734 / 0.1947 / 45.85 | 15.83 / 0.5335 / 0.5260 / 181.9 |
| | | LLE | 37.83 / 0.9578 / 0.0992 / 23.67 | 33.44 / 0.9221 / 0.1531 / 39.85 | 31.27 / 0.8810 / 0.1838 / 45.55 | 22.58 / 0.7606 / 0.3053 / 66.55 |
| | 15 | - | **38.44** / **0.9619** / 0.0913 / **23.20** | 34.69 / **0.9343** / 0.1378 / 35.86 | 31.11 / 0.8826 / 0.1758 / 42.07 | 16.39 / 0.5868 / 0.4567 / 141.2 |
| | | LLE | 38.39 / 0.9618 / **0.0905** / 23.48 | 34.69 / 0.9341 / **0.1355** / 34.85 | **31.29** / **0.8834** / 0.1684 / 40.51 | 24.46 / 0.8165 / 0.2444 / 51.69 |

Table 25: Results of ΠGDM on FFHQ Dataset.

| Condition | Steps | Strategy | PSNR↑/SSIM↑/LPIPS↓/FID↓ | | | |
| --- | --- | --- | --- | --- | --- | --- |
| | | | Deblur (aniso) | Inpainting | 4× SR | CS 50% |
| $\sigma_\mathbf{y} = 0.05$ | 3 | - | 20.72 / 0.5570 / 0.5370 / **188.9** | 19.23 / 0.5259 / 0.5169 / 178.7 | 20.85 / 0.5671 / 0.5154 / 182.9 | 16.81 / 0.4767 / 0.5761 / **186.0** |
| | | LLE | 21.51 / 0.5835 / 0.5110 / 190.7 | 20.51 / 0.5686 / 0.5008 / 174.1 | 21.20 / 0.5746 / 0.5112 / 178.8 | 17.11 / 0.4906 / 0.5566 / 187.3 |
| | 4 | - | 19.59 / 0.5583 / 0.5305 / 181.1 | 21.44 / 0.6209 / 0.4475 / 151.9 | 19.84 / 0.5784 / 0.4999 / 159.8 | 16.25 / 0.4837 / 0.5922 / 179.9 |
| | | LLE | 23.91 / 0.6670 / 0.4366 / 149.6 | 22.48 / 0.6483 / 0.4300 / 144.7 | 23.67 / 0.6614 / 0.4324 / 153.7 | 18.11 / 0.5407 / 0.5097 / 164.0 |
| | 5 | - | 20.88 / 0.6145 / 0.5010 / 174.2 | 21.98 / 0.6700 / 0.4142 / 141.5 | 20.63 / 0.6284 / 0.4687 / 145.3 | 15.20 / 0.4723 / 0.6143 / 199.1 |
| | | LLE | 25.38 / 0.7159 / 0.3787 / 126.1 | 24.35 / 0.7080 / 0.3772 / 138.7 | 24.96 / 0.7093 / 0.3757 / 129.4 | 18.53 / 0.5754 / 0.4835 / 147.6 |
| | 7 | - | 23.21 / 0.6856 / 0.4234 / 156.8 | 25.07 / 0.7515 / 0.3491 / 133.2 | 22.43 / 0.6891 / 0.4114 / 136.3 | 13.42 / 0.4161 / 0.6818 / 238.3 |
| | | LLE | 27.27 / 0.7752 / 0.3095 / 109.9 | 26.32 / 0.7620 / 0.3289 / 124.4 | 26.53 / 0.7585 / 0.3112 / 112.2 | 19.60 / 0.6378 / 0.4331 / 130.9 |
| | 10 | - | 26.13 / 0.7565 / 0.3158 / 132.1 | 28.37 / **0.8130** / 0.2902 / 120.7 | 25.46 / 0.7564 / 0.3229 / 131.2 | 13.89 / 0.4474 / 0.6421 / 239.7 |
| | | LLE | 28.51 / 0.8056 / 0.2648 / 100.3 | 28.43 / 0.8104 / 0.2876 / 116.8 | 27.45 / 0.7855 / 0.2765 / 102.9 | 20.24 / 0.6834 / 0.3833 / 117.0 |
| | 15 | - | 28.02 / 0.7871 / 0.2658 / 104.3 | **30.44** / **0.8527** / 0.2552 / 108.3 | 27.64 / 0.7913 / 0.2679 / **97.82** | 13.99 / 0.4936 / 0.5747 / 240.8 |
| | | LLE | 29.14 / 0.8216 / 0.2472 / **97.73** | 30.28 / 0.8503 / **0.2511** / 105.2 | 27.89 / 0.7965 / 0.2598 / 97.97 | 24.59 / 0.7952 / 0.2756 / 99.09 |
| $\sigma_\mathbf{y} = 0$ | 3 | - | 31.75 / 0.9038 / 0.1889 / 67.86 | 24.06 / 0.7379 / 0.3826 / 169.9 | 26.78 / 0.7761 / 0.4028 / 150.0 | 18.25 / 0.5509 / 0.5430 / 323.8 |
| | | LLE | 35.39 / 0.9393 / 0.1358 / 44.26 | 25.56 / 0.7971 / 0.3204 / 148.8 | 27.39 / 0.8020 / 0.3478 / 142.6 | 18.47 / 0.5673 / 0.5226 / 290.6 |
| | 4 | - | 26.67 / 0.8661 / 0.2494 / 108.7 | 26.02 / 0.8131 / 0.3082 / 140.4 | 24.15 / 0.7344 / 0.4280 / 167.7 | 17.49 / 0.5334 / 0.5652 / 325.6 |
| | | LLE | 36.21 / 0.9478 / 0.1197 / 37.71 | 27.28 / 0.8406 / 0.2629 / 129.4 | 28.26 / 0.8254 / 0.3002 / 128.6 | 19.16 / 0.6139 / 0.4703 / 237.2 |
| | 5 | - | 27.90 / 0.8962 / 0.2210 / 105.4 | 26.48 / 0.8310 / 0.2826 / 131.4 | 25.31 / 0.7630 / 0.3935 / 155.6 | 15.94 / 0.5053 / 0.5858 / 345.1 |
| | | LLE | 36.96 / 0.9531 / 0.1085 / 33.22 | 28.77 / 0.8707 / 0.2228 / 111.4 | 28.81 / 0.8383 / 0.2710 / 118.9 | 19.41 / 0.6391 / 0.4487 / 212.0 |
| | 7 | - | 30.26 / 0.9213 / 0.1882 / 105.7 | 29.37 / 0.8829 / 0.2083 / 94.56 | 26.42 / 0.7816 / 0.3489 / 149.1 | 13.87 / 0.4492 / 0.6354 / 373.8 |
| | | LLE | 37.71 / 0.9610 / 0.0893 / 27.10 | 30.42 / 0.8938 / 0.1899 / 88.12 | 29.97 / 0.8646 / 0.2211 / 98.01 | 20.25 / 0.6984 / 0.3756 / 165.7 |
| | 10 | - | 36.16 / 0.9572 / 0.1118 / 50.10 | **32.37** / **0.9161** / 0.1553 / 63.87 | 28.07 / 0.8285 / 0.2533 / 111.5 | 14.22 / 0.4801 / 0.5940 / 317.6 |
| | | LLE | 39.26 / 0.9684 / 0.0712 / 19.95 | 32.28 / 0.9135 / 0.1579 / 65.59 | 30.61 / 0.8785 / 0.1858 / 76.62 | 20.67 / 0.7351 / 0.3322 / 141.4 |
| | 15 | - | **40.37** / **0.9732** / 0.0701 / 19.31 | 34.18 / **0.9332** / 0.1333 / 51.88 | 29.02 / 0.8546 / 0.1984 / 80.09 | 14.17 / 0.5222 / 0.5378 / 272.4 |
| | | LLE | 39.92 / 0.9724 / **0.0642** / 19.22 | 33.96 / 0.9317 / **0.1324** / 51.07 | 30.93 / 0.8861 / 0.1714 / 65.55 | 25.61 / 0.8493 / 0.1984 / 72.41 |

Table 26: Results of DMPS on CelebA-HQ Dataset.

| Condition | Steps | Strategy | PSNR↑/SSIM↑/LPIPS↓/FID↓ | | | |
| --- | --- | --- | --- | --- | --- | --- |
| | | | Deblur (aniso) | Inpainting | 4× SR | CS 50% |
| $\sigma_\mathbf{y} = 0.05$ | 3 | - | 14.02 / 0.1746 / 0.7466 / 203.9 | 11.91 / 0.1008 / 0.8018 / **296.5** | 15.53 / 0.2346 / **0.7632** / 203.1 | 12.45 / 0.1023 / 0.7908 / 325.1 |
| | | LLE | **14.10 / 0.1771 / 0.7440 / 197.7** | **11.97 / 0.1042 / 0.8000** / 297.8 | **18.76 / 0.2530 / 0.7761 / 186.0** | **12.52 / 0.1051 / 0.7887 / 320.5** |
| | 4 | - | 17.95 / 0.3221 / 0.6191 / 101.0 | 13.41 / 0.1441 / 0.7624 / 187.2 | 19.33 / 0.3653 / 0.6453 / 106.5 | 14.09 / 0.1508 / 0.7412 / **209.0** |
| | | LLE | **18.23 / 0.3245 / 0.6172 / 100.2** | **13.55 / 0.1490 / 0.7591 / 183.1** | **23.07 / 0.5105 / 0.5443 / 77.59** | **14.19 / 0.1554 / 0.7384** / 212.3 |
| | 5 | - | 21.20 / 0.4046 / 0.5619 / 98.57 | 14.71 / 0.1639 / 0.7375 / 151.1 | 22.95 / 0.4978 / 0.5412 / 71.13 | 14.85 / 0.1651 / 0.7194 / 186.6 |
| | | LLE | **21.63 / 0.4064 / 0.5585 / 97.43** | **16.45 / 0.1856 / 0.6902 / 140.8** | **24.66 / 0.6248 / 0.4368 / 63.00** | **15.57 / 0.1800 / 0.6989 / 169.1** |
| | 7 | - | 25.12 / 0.5594 / 0.4462 / 67.15 | 15.87 / 0.1866 / 0.7156 / 134.7 | **25.96 / 0.6690** / 0.3960 / 55.71 | 16.04 / 0.2131 / 0.6792 / 170.4 |
| | | LLE | **25.84 / 0.5759 / 0.4355 / 65.91** | **20.12 / 0.3374 / 0.5615 / 86.63** | 25.82 / **0.7065 / 0.3552 / 55.04** | **17.36 / 0.3030 / 0.5940 / 116.2** |
| | 10 | - | 25.27 / 0.5064 / 0.4439 / 56.07 | 16.74 / 0.1759 / 0.7006 / 116.5 | **27.86 / 0.7248** / 0.3425 / 45.10 | 14.91 / 0.1538 / 0.7179 / 192.9 |
| | | LLE | **26.54 / 0.5773 / 0.4051 / 53.03** | **23.01 / 0.4236 / 0.4912 / 63.78** | 27.72 / **0.7528 / 0.2976 / 41.67** | **18.49 / 0.3584 / 0.5461 / 90.12** |
| | 15 | - | 28.65 / 0.7234 / 0.3005 / 35.61 | 21.82 / 0.3564 / 0.5350 / 71.44 | **28.63 / 0.8005** / 0.2491 / **36.43** | 18.47 / 0.3153 / 0.5728 / 102.4 |
| | | LLE | **28.93 / 0.7459 / 0.2873 / 33.97** | **25.36 / 0.5604 / 0.4113 / 53.76** | 28.22 / 0.7946 / **0.2466** / 37.68 | **20.52 / 0.4838 / 0.4580 / 66.70** |
| $\sigma_\mathbf{y} = 0$ | 3 | - | 15.35 / 0.2104 / 0.7004 / 117.9 | 12.74 / 0.1432 / 0.7834 / 226.1 | 18.32 / 0.3406 / 0.7047 / 132.2 | 13.48 / 0.1607 / 0.7596 / **357.4** |
| | | LLE | **15.52 / 0.2196 / 0.6920 / 112.9** | **12.81 / 0.1478 / 0.7799 / 217.4** | **22.05 / 0.4104 / 0.7075 / 128.7** | **13.55 / 0.1656 / 0.7568** / 361.6 |
| | 4 | - | 19.18 / 0.2986 / 0.6060 / 86.10 | 14.58 / 0.2143 / 0.7242 / 155.9 | 22.69 / 0.4864 / 0.5660 / 86.40 | 15.19 / 0.2531 / 0.6863 / 264.5 |
| | | LLE | **19.43 / 0.3044 / 0.6022 / 86.55** | **14.67 / 0.2222 / 0.7203 / 155.5** | **26.73 / 0.6988 / 0.4053 / 62.17** | **15.28 / 0.2598 / 0.6826** / 272.1 |
| | 5 | - | 22.08 / 0.3838 / 0.5259 / 68.94 | 16.25 / 0.2658 / 0.6772 / 133.3 | 26.18 / 0.6359 / 0.4429 / 60.76 | 16.08 / 0.3057 / 0.6454 / **241.4** |
| | | LLE | **22.49 / 0.4037 / 0.5117 / 66.85** | **16.43 / 0.2762 / 0.6700 / 132.1** | **28.34 / 0.7947 / 0.3071 / 59.01** | **16.15 / 0.3094 / 0.6432** / 250.4 |
| | 7 | - | 26.09 / 0.5551 / 0.3991 / 50.89 | 18.06 / 0.3398 / 0.6313 / 126.8 | 28.90 / 0.7911 / 0.3168 / 52.36 | 17.29 / 0.4052 / 0.5890 / 228.6 |
| | | LLE | **26.55 / 0.5731 / 0.3877 / 49.45** | **23.47 / 0.5990 / 0.4406 / 79.94** | **29.09 / 0.8329 / 0.2516 / 58.33** | **18.64 / 0.5121 / 0.5187 / 165.2** |
| | 10 | - | 25.76 / 0.6363 / 0.3280 / 45.89 | 20.67 / 0.4443 / 0.5319 / 91.88 | 29.80 / 0.8124 / 0.2560 / 44.62 | 17.40 / 0.4532 / 0.5574 / 228.1 |
| | | LLE | **32.23 / 0.8288 / 0.1889 / 23.23** | **26.72 / 0.6933 / 0.3579 / 59.77** | **29.89 / 0.8461 / 0.2129 / 43.73** | **19.71 / 0.5756 / 0.4638 / 128.8** |
| | 15 | - | 31.22 / 0.7975 / 0.2026 / 24.48 | 26.04 / 0.6688 / 0.3702 / 61.75 | **30.53 / 0.8494 /** 0.2115 / **37.04** | 20.24 / 0.5970 / 0.4531 / 132.2 |
| | | LLE | **34.47 / 0.8791 / 0.1482 / 16.35** | **28.52 / 0.7743 / 0.2997 / 53.96** | 29.95 / **0.8537 / 0.1939** / 37.38 | **21.54 / 0.6484 / 0.3996 / 94.74** |

Table 27: Results of DMPS on FFHQ Dataset.

| Condition | Steps | Strategy | PSNR↑/SSIM↑/LPIPS↓/FID↓ | | | |
| --- | --- | --- | --- | --- | --- | --- |
| | | | Deblur (aniso) | Inpainting | 4× SR | CS 50% |
| $\sigma_\mathbf{y} = 0.05$ | 3 | - | 14.42 / 0.1843 / 0.7403 / 234.4 | 12.43 / 0.1153 / 0.7997 / 320.9 | 15.56 / 0.2302 / **0.7534** / 355.1 | 12.80 / 0.1142 / 0.7917 / **363.4** |
| | | LLE | **14.46 / 0.1854 / 0.7358 / 232.8** | **12.48 / 0.1206 / 0.7952 / 312.7** | **18.63 / 0.2575 / 0.7697 / 253.2** | **12.86 / 0.1175 / 0.7878** / 366.4 |
| | 4 | - | 17.60 / 0.2681 / 0.6507 / 171.8 | 13.75 / 0.1599 / 0.7601 / 252.2 | 18.09 / 0.2962 / 0.7003 / 329.5 | 14.22 / 0.1648 / 0.7448 / **283.6** |
| | | LLE | **17.82 / 0.2697 / 0.6484 / 168.9** | **13.81 / 0.1600 / 0.7567 / 250.0** | **22.62 / 0.4885 / 0.5586 / 149.2** | **14.33 / 0.1670 / 0.7435** / 288.7 |
| | 5 | - | 20.50 / 0.3772 / 0.5876 / **164.0** | 14.73 / 0.1674 / 0.7474 / 235.0 | 22.25 / 0.4547 / 0.5601 / 148.7 | 14.90 / 0.1650 / 0.7410 / 295.1 |
| | | LLE | **20.86 / 0.3800 / 0.5843** / 164.3 | **14.89 / 0.1688 / 0.7436 / 225.5** | **24.13 / 0.5953 / 0.4643 / 135.1** | **14.98 / 0.1672 / 0.7374 / 291.5** |
| | 7 | - | 24.24 / **0.5254** / 0.4848 / 140.9 | 15.85 / 0.1964 / 0.7243 / 210.1 | 24.90 / 0.6291 / 0.434 / 127.4 | 15.76 / 0.1984 / 0.7096 / 252.6 |
| | | LLE | **24.71** / 0.5244 / **0.4820 / 140.2** | **19.55 / 0.3066 / 0.5993 / 170.0** | **25.22 / 0.6791 / 0.3879 / 123.6** | **16.71 / 0.2682 / 0.6476 / 200.0** |
| | 10 | - | 24.48 / 0.4699 / 0.4740 / 121.4 | 16.59 / 0.1788 / 0.7164 / 191.2 | **26.91 / 0.6953** / 0.3812 / 111.0 | 14.84 / 0.1572 / 0.7356 / 299.8 |
| | | LLE | **25.65 / 0.5404 / 0.4420 / 120.0** | **22.42 / 0.4007 / 0.5249 / 138.4** | 26.76 / **0.7269 / 0.3353 / 103.6** | **17.82 / 0.3411 / 0.5819 / 168.2** |
| | 15 | - | 27.85 / 0.6932 / 0.3497 / 103.0 | 21.26 / 0.3387 / 0.5683 / 145.5 | **27.64 / 0.7774** / 0.2848 / 93.93 | 17.87 / 0.2998 / 0.6097 / 176.6 |
| | | LLE | **28.06 / 0.7155 / 0.3350 / 97.64** | **24.35 / 0.5227 / 0.4523 / 127.4** | 27.23 / 0.7750 / **0.2776 / 91.17** | **19.07 / 0.4465 / 0.5091 / 143.7** |
| $\sigma_\mathbf{y} = 0$ | 3 | - | 15.90 / 0.2429 / 0.6847 / 171.1 | 13.23 / 0.1577 / 0.7777 / 266.7 | 18.07 / 0.3122 / 0.6982 / 249.2 | 13.75 / 0.1725 / 0.7603 / **396.3** |
| | | LLE | **16.06 / 0.2551 / 0.6728 / 163.2** | **13.28 / 0.1632 / 0.7730 / 265.3** | **21.77 / 0.4038 / 0.7009 / 200.1** | **13.82 / 0.1778 / 0.7557** / 407.6 |
| | 4 | - | 18.97 / 0.2916 / 0.6285 / 152.4 | 14.85 / 0.2277 / 0.7218 / 230.0 | 20.95 / 0.3849 / 0.6343 / 247.3 | 15.24 / 0.2645 / 0.6944 / **354.1** |
| | | LLE | **19.78 / 0.3274 / 0.5919 / 140.9** | **14.92 / 0.2299 / 0.7200 / 225.8** | **25.88 / 0.6588 / 0.4420 / 130.7** | **15.35 / 0.2677 / 0.6916** / 363.0 |
| | 5 | - | 21.49 / 0.3679 / 0.5530 / 133.1 | 16.21 / 0.2660 / 0.6899 / 204.8 | 25.14 / 0.5672 / 0.4677 / 129.0 | 16.11 / 0.3034 / 0.6664 / 342.5 |
| | | LLE | **24.53 / 0.5214 / 0.4315 / 107.3** | **18.54 / 0.3406 / 0.6123 / 210.6** | **27.34 / 0.7527 / 0.3560 / 117.7** | **16.79 / 0.3442 / 0.6318 / 295.4** |
| | 7 | - | 25.21 / 0.5247 / 0.4269 / 105.0 | 17.72 / 0.3332 / 0.6465 / 202.9 | 27.89 / 0.7558 / 0.3597 / 115.1 | 17.03 / 0.3812 / 0.6157 / 309.3 |
| | | LLE | **29.57 / 0.7431 / 0.2770 / 76.39** | **22.69 / 0.5557 / 0.4724 / 160.5** | **28.43 / 0.8195 / 0.2859 / 109.0** | **17.85 / 0.4639 / 0.5580 / 251.0** |
| | 10 | - | 26.87 / 0.6778 / 0.3125 / 82.26 | 20.04 / 0.4261 / 0.5654 / 167.7 | 29.15 / 0.8085 / 0.2861 / 98.24 | 17.20 / 0.4539 / 0.5817 / 326.9 |
| | | LLE | **32.51 / 0.8513 / 0.1805 / 48.60** | **26.10 / 0.6880 / 0.3809 / 125.5** | **29.21 / 0.8438 / 0.2383 / 92.21** | **18.95 / 0.5689 / 0.4871 / 212.6** |
| | 15 | - | 32.11 / 0.8338 / 0.1916 / 49.60 | 25.48 / 0.6675 / 0.3957 / 127.1 | **29.96 / 0.8549** / 0.2320 / 91.92 | 19.52 / 0.5940 / 0.4816 / 214.9 |
| | | LLE | **34.62 / 0.8978 / 0.1398 / 35.35** | **27.45 / 0.7519 / 0.3360 / 113.8** | 29.38 / 0.8557 / **0.2116 / 81.28** | **19.91 / 0.6231 / 0.4397 / 177.4** |

Table 28: Results of DPS on CelebA-HQ Dataset.

| Condition | Steps | Strategy | Deblur (aniso) | Inpainting | 4× SR | CS 50% | Deblur (nonlinear) |
|---|---|---|---|---|---|---|---|
| | | | PSNR↑/SSIM↑/LPIPS↓/FID↓ | | | | |
| $\sigma_{\mathbf{y}} = 0.05$ | 3 | - | 23.59 / 0.6503 / 0.4152 / 79.42 | 23.57 / 0.5577 / 0.4973 / 85.49 | 25.49 / 0.6469 / 0.5276 / 82.53 | 14.30 / 0.3585 / 0.6364 / 191.8 | 16.09 / 0.4075 / 0.5754 / 120.6 |
| | | LLE | 24.59 / 0.6747 / 0.4051 / 73.57 | 27.51 / 0.7476 / 0.3664 / 60.87 | 24.57 / 0.6661 / 0.4650 / 81.79 | 15.83 / 0.4679 / 0.5913 / 183.8 | 19.06 / 0.4752 / 0.5795 / 109.9 |
| | 4 | - | 23.35 / 0.6620 / 0.3995 / 63.53 | 23.19 / 0.5653 / 0.4850 / 81.82 | 25.21 / 0.6558 / 0.4944 / 73.41 | 16.01 / 0.4543 / 0.5383 / 135.6 | 19.27 / 0.5266 / 0.4665 / 88.48 |
| | | LLE | 25.18 / 0.7063 / 0.3539 / 61.87 | 28.85 / 0.7830 / 0.3277 / 50.18 | 25.34 / 0.7196 / 0.3609 / 67.81 | 16.84 / 0.5018 / 0.5102 / 122.7 | 21.25 / 0.5796 / 0.4464 / 73.48 |
| | 5 | - | 23.94 / 0.6801 / 0.3695 / 56.70 | 25.14 / 0.6173 / 0.4342 / 64.64 | 26.07 / 0.6754 / 0.4698 / 69.38 | 17.12 / 0.5149 / 0.4811 / 114.4 | 21.19 / 0.6052 / 0.4060 / 75.86 |
| | | LLE | 25.56 / 0.7218 / 0.3264 / 54.62 | 27.42 / 0.7964 / 0.3035 / 44.76 | 26.63 / 0.7583 / 0.3148 / 56.01 | 17.73 / 0.5360 / 0.4681 / 102.0 | 22.33 / 0.6267 / 0.3930 / 64.26 |
| | 7 | - | 24.75 / 0.6989 / 0.3289 / 49.49 | 27.71 / 0.6985 / 0.3612 / 50.27 | 27.01 / 0.7020 / 0.4275 / 63.38 | 17.58 / 0.5470 / 0.4574 / 103.3 | 22.07 / 0.6282 / 0.3889 / 65.26 |
| | | LLE | 26.32 / 0.7458 / 0.2845 / 48.14 | 30.49 / 0.8599 / 0.2414 / 42.30 | 28.48 / 0.8055 / 0.2568 / 45.09 | 18.87 / 0.5763 / 0.4203 / 83.25 | 23.24 / 0.6537 / 0.3629 / 55.18 |
| | 10 | - | 25.69 / 0.7334 / 0.2784 / 46.75 | 29.90 / 0.7792 / 0.2983 / 42.25 | 27.70 / 0.7287 / 0.3905 / 61.83 | 19.09 / 0.6048 / 0.4018 / 84.22 | 23.74 / 0.6732 / 0.3504 / 57.00 |
| | | LLE | 27.00 / 0.7653 / 0.2521 / 43.98 | 32.73 / 0.8985 / 0.1968 / 41.88 | 29.26 / 0.8253 / 0.2208 / 38.15 | 20.41 / 0.6322 / 0.3708 / 70.17 | 24.47 / 0.7024 / 0.3016 / 50.39 |
| | 15 | - | 26.27 / 0.7508 / 0.2536 / 44.48 | 30.70 / 0.8109 / 0.2732 / 40.03 | 27.70 / 0.7285 / 0.3757 / 60.56 | 19.86 / 0.6506 / 0.3600 / 69.78 | 24.93 / 0.7026 / 0.3256 / 50.65 |
| | | LLE | 27.25 / 0.7705 / 0.2393 / 42.10 | 33.73 / 0.9146 / 0.1715 / 38.92 | 29.32 / 0.8285 / 0.2121 / 38.07 | 22.46 / 0.7023 / 0.3130 / 56.22 | 25.75 / 0.7376 / 0.2739 / 46.55 |
| $\sigma_{\mathbf{y}} = 0$ | 3 | - | 23.61 / 0.6539 / 0.4111 / 83.22 | 24.60 / 0.6767 / 0.4447 / 90.50 | 26.91 / 0.7824 / 0.3649 / 77.84 | 14.30 / 0.3591 / 0.6359 / 191.0 | 16.09 / 0.4080 / 0.5747 / 120.7 |
| | | LLE | 24.71 / 0.6835 / 0.3961 / 76.66 | 28.74 / 0.8309 / 0.2991 / 65.60 | 27.56 / 0.7979 / 0.3575 / 75.64 | 15.83 / 0.4730 / 0.5889 / 179.1 | 19.06 / 0.4790 / 0.5791 / 109.9 |
| | 4 | - | 23.38 / 0.6657 / 0.3961 / 66.12 | 24.21 / 0.6925 / 0.4242 / 86.20 | 26.64 / 0.7946 / 0.3265 / 71.57 | 16.02 / 0.4548 / 0.5378 / 136.4 | 19.27 / 0.5271 / 0.4661 / 86.07 |
| | | LLE | 25.30 / 0.7140 / 0.3444 / 64.70 | 30.52 / 0.8773 / 0.2445 / 55.86 | 28.49 / 0.8251 / 0.2880 / 71.06 | 16.84 / 0.5035 / 0.5092 / 123.4 | 21.26 / 0.5812 / 0.4454 / 74.49 |
| | 5 | - | 23.97 / 0.6839 / 0.3658 / 59.78 | 26.95 / 0.7772 / 0.3479 / 67.05 | 27.95 / 0.8211 / 0.2862 / 65.19 | 17.12 / 0.5156 / 0.4809 / 115.4 | 21.19 / 0.6058 / 0.4060 / 76.80 |
| | | LLE | 25.69 / 0.7288 / 0.3189 / 57.10 | 31.67 / 0.8971 / 0.2128 / 51.23 | 29.35 / 0.8436 / 0.2545 / 63.48 | 17.74 / 0.5377 / 0.4671 / 102.9 | 22.35 / 0.6284 / 0.3918 / 64.21 |
| | 7 | - | 24.79 / 0.7029 / 0.3260 / 52.98 | 31.05 / 0.8769 / 0.2296 / 48.50 | 29.49 / 0.8485 / 0.2339 / 56.48 | 17.58 / 0.5478 / 0.4574 / 104.4 | 22.08 / 0.6291 / 0.3886 / 66.33 |
| | | LLE | 26.40 / 0.7509 / 0.2800 / 50.35 | 33.69 / 0.9278 / 0.1539 / 39.17 | 30.49 / 0.8660 / 0.2114 / 53.60 | 18.88 / 0.5763 / 0.4201 / 84.60 | 23.27 / 0.6542 / 0.3637 / 56.40 |
| | 10 | - | 25.74 / 0.7373 / 0.2778 / 50.49 | 34.83 / 0.9389 / 0.1311 / 31.60 | 30.75 / 0.8714 / 0.1982 / 49.39 | 19.09 / 0.6058 / 0.4020 / 85.62 | 23.76 / 0.6746 / 0.3509 / 58.50 |
| | | LLE | 27.06 / 0.7699 / 0.2480 / 46.43 | 35.37 / 0.9458 / 0.1142 / 30.42 | 31.07 / 0.8764 / 0.1867 / 46.50 | 20.42 / 0.6340 / 0.3702 / 71.12 | 24.49 / 0.7024 / 0.3018 / 50.26 |
| | 15 | - | 26.34 / 0.7553 / 0.2533 / 48.18 | 36.29 / 0.9535 / 0.0993 / 26.88 | 31.26 / 0.8802 / 0.1825 / 44.55 | 19.87 / 0.6519 / 0.3606 / 71.49 | 24.91 / 0.7018 / 0.3285 / 52.48 |
| | | LLE | 27.36 / 0.7750 / 0.2355 / 43.98 | 36.26 / 0.9531 / 0.0964 / 26.53 | 31.10 / 0.8778 / 0.1757 / 42.11 | 22.48 / 0.7043 / 0.3133 / 57.61 | 25.77 / 0.7378 / 0.2736 / 46.41 |

Table 29: Results of DPS on FFHQ Dataset.

| Condition | Steps | Strategy | Deblur (aniso) | Inpainting | 4× SR | CS 50% | Deblur (nonlinear) |
|---|---|---|---|---|---|---|---|
| | | | PSNR↑/SSIM↑/LPIPS↓/FID↓ | | | | |
| $\sigma_{\mathbf{y}} = 0.05$ | 3 | - | 23.41 / 0.6567 / 0.4281 / 151.3 | 23.93 / 0.6065 / 0.4790 / 157.9 | 24.73 / 0.6631 / 0.4799 / 148.1 | 18.20 / 0.5301 / 0.5565 / 337.8 | 15.75 / 0.4025 / 0.6446 / 233.7 |
| | | LLE | 24.26 / 0.6759 / 0.4171 / 150.0 | 27.01 / 0.7421 / 0.3598 / 128.7 | 24.31 / 0.6600 / 0.4634 / 153.7 | 18.39 / 0.5496 / 0.5349 / 294.0 | 16.93 / 0.4403 / 0.5900 / 201.0 |
| | 4 | - | 22.90 / 0.6481 / 0.4335 / 146.9 | 21.89 / 0.5513 / 0.5110 / 171.2 | 25.00 / 0.6832 / 0.4396 / 140.5 | 17.43 / 0.5086 / 0.5788 / 335.7 | 18.05 / 0.4822 / 0.5381 / 192.5 |
| | | LLE | 24.58 / 0.6950 / 0.3785 / 128.8 | 28.25 / 0.7736 / 0.3329 / 115.8 | 25.44 / 0.7046 / 0.4014 / 137.1 | 19.09 / 0.5936 / 0.4856 / 230.1 | 18.91 / 0.4653 / 0.5512 / 182.6 |
| | 5 | - | 23.02 / 0.6588 / 0.4184 / 143.2 | 22.48 / 0.5646 / 0.4943 / 154.3 | 25.19 / 0.6938 / 0.4260 / 134.0 | 15.90 / 0.4833 / 0.5971 / 359.3 | 19.31 / 0.5286 / 0.4871 / 164.8 |
| | | LLE | 25.01 / 0.7131 / 0.3481 / 111.5 | 28.47 / 0.7663 / 0.3352 / 112.3 | 26.31 / 0.7342 / 0.3616 / 130.6 | 19.34 / 0.6173 / 0.4652 / 205.0 | 19.70 / 0.5178 / 0.4958 / 159.4 |
| | 7 | - | 23.31 / 0.6592 / 0.4037 / 146.1 | 24.04 / 0.5997 / 0.4565 / 143.3 | 26.35 / 0.7206 / 0.3846 / 124.1 | 13.85 / 0.4338 / 0.6427 / 386.1 | 19.60 / 0.5272 / 0.4873 / 157.9 |
| | | LLE | 25.56 / 0.7333 / 0.3176 / 104.3 | 29.00 / 0.8294 / 0.2799 / 102.4 | 26.97 / 0.7731 / 0.2956 / 99.94 | 20.16 / 0.6774 / 0.3938 / 161.7 | 20.22 / 0.5453 / 0.4475 / 135.3 |
| | 10 | - | 23.42 / 0.6562 / 0.3561 / 129.7 | 26.50 / 0.6595 / 0.3877 / 116.8 | 27.44 / 0.7525 / 0.3367 / 112.8 | 14.20 / 0.4676 / 0.6003 / 331.8 | 20.30 / 0.5360 / 0.4628 / 142.3 |
| | | LLE | 26.01 / 0.7457 / 0.2910 / 95.11 | 31.16 / 0.8692 / 0.2331 / 94.28 | 27.94 / 0.8006 / 0.2631 / 91.24 | 20.58 / 0.7103 / 0.3541 / 138.8 | 21.18 / 0.5800 / 0.4011 / 121.5 |
| | 15 | - | 23.72 / 0.6691 / 0.3254 / 116.3 | 28.94 / 0.7493 / 0.3038 / 90.92 | 27.72 / 0.7526 / 0.3222 / 104.7 | 14.20 / 0.5071 / 0.5451 / 280.9 | 21.00 / 0.5518 / 0.4411 / 132.8 |
| | | LLE | 26.48 / 0.7594 / 0.2743 / 93.22 | 32.97 / 0.9029 / 0.1917 / 79.57 | 28.41 / 0.8133 / 0.2412 / 83.82 | 25.22 / 0.8189 / 0.2289 / 73.91 | 22.08 / 0.6109 / 0.3724 / 118.0 |
| $\sigma_{\mathbf{y}} = 0$ | 3 | - | 23.43 / 0.6609 / 0.4228 / 150.7 | 25.00 / 0.7232 / 0.4174 / 156.1 | 25.60 / 0.7723 / 0.3362 / 142.7 | 18.25 / 0.5509 / 0.5430 / 323.8 | 15.75 / 0.4032 / 0.6465 / 234.5 |
| | | LLE | 24.40 / 0.6866 / 0.4033 / 145.0 | 28.40 / 0.8356 / 0.2753 / 122.5 | 27.12 / 0.7998 / 0.3413 / 141.2 | 18.47 / 0.5673 / 0.5226 / 290.6 | 16.94 / 0.4403 / 0.5902 / 200.8 |
| | 4 | - | 22.92 / 0.6520 / 0.4306 / 147.1 | 22.78 / 0.6793 / 0.4437 / 168.0 | 25.94 / 0.7854 / 0.3132 / 133.1 | 17.49 / 0.5334 / 0.5652 / 325.7 | 18.05 / 0.4825 / 0.5378 / 193.2 |
| | | LLE | 24.70 / 0.7024 / 0.3704 / 129.7 | 30.19 / 0.8836 / 0.2327 / 106.5 | 27.81 / 0.8158 / 0.2884 / 128.0 | 19.16 / 0.6139 / 0.4702 / 237.1 | 18.92 / 0.4663 / 0.5510 / 181.8 |
| | 5 | - | 23.05 / 0.6632 / 0.4151 / 143.6 | 23.72 / 0.7249 / 0.4120 / 158.4 | 26.25 / 0.8008 / 0.2996 / 125.6 | 15.94 / 0.5053 / 0.5858 / 345.1 | 19.32 / 0.5292 / 0.4867 / 164.1 |
| | | LLE | 25.09 / 0.7185 / 0.3434 / 117.0 | 30.48 / 0.8857 / 0.2382 / 109.0 | 28.38 / 0.8303 / 0.2551 / 117.8 | 19.41 / 0.6391 / 0.4487 / 212.1 | 19.71 / 0.5184 / 0.4952 / 158.4 |
| | 7 | - | 23.34 / 0.6642 / 0.4002 / 151.4 | 26.05 / 0.7897 / 0.3494 / 147.1 | 27.84 / 0.8310 / 0.2565 / 111.5 | 13.87 / 0.4492 / 0.6343 / 374.0 | 19.59 / 0.5276 / 0.4875 / 156.2 |
| | | LLE | 25.63 / 0.7371 / 0.3154 / 110.9 | 32.08 / 0.9191 / 0.1713 / 80.60 | 29.15 / 0.8471 / 0.2267 / 102.1 | 20.25 / 0.6983 / 0.3756 / 165.6 | 20.22 / 0.5446 / 0.4481 / 136.1 |
| | 10 | - | 23.44 / 0.6609 / 0.3543 / 134.5 | 30.32 / 0.8669 / 0.2424 / 111.4 | 29.50 / 0.8551 / 0.2220 / 95.58 | 14.22 / 0.4801 / 0.5940 / 317.4 | 20.39 / 0.5417 / 0.4574 / 140.5 |
| | | LLE | 26.10 / 0.7496 / 0.2898 / 101.0 | 34.14 / 0.9378 / 0.1242 / 51.41 | 29.90 / 0.8611 / 0.2047 / 84.51 | 20.67 / 0.7351 / 0.3322 / 141.4 | 21.18 / 0.5807 / 0.4009 / 123.1 |
| | 15 | - | 23.71 / 0.6719 / 0.3278 / 124.3 | 34.71 / 0.9275 / 0.1405 / 57.25 | 30.49 / 0.8714 / 0.2014 / 81.00 | 14.16 / 0.5218 / 0.5381 / 272.2 | 21.04 / 0.5541 / 0.4425 / 134.0 |
| | | LLE | 26.56 / 0.7634 / 0.2737 / 100.0 | 35.91 / 0.9572 / 0.0808 / 28.30 | 30.34 / 0.8706 / 0.1919 / 76.12 | 25.61 / 0.8493 / 0.1984 / 72.42 | 22.08 / 0.6110 / 0.3736 / 119.5 |

Table 30: Results of RED-diff on CelebA-HQ Dataset.

| Condition | Steps | Strategy | Deblur (aniso) | Inpainting | 4× SR | CS 50% | Deblur (nonlinear) |
|---|---|---|---|---|---|---|---|
| | | | PSNR↑/SSIM↑/LPIPS↓/FID↓ | | | | |
| $\sigma_{\mathbf{y}} = 0.05$ | 3 | - | 24.91 / 0.7020 / 0.3727 / 53.55 | 18.93 / 0.3496 / 0.6153 / 122.3 | 25.45 / 0.5728 / 0.5665 / 69.08 | 16.98 / 0.4082 / 0.5908 / 266.6 | 18.81 / 0.3511 / 0.6530 / 111.6 |
| | | LLE | 25.27 / 0.7110 / 0.3515 / 46.47 | 24.40 / 0.6222 / 0.4106 / 63.43 | 26.92 / 0.7710 / 0.2703 / 41.64 | 17.00 / 0.5211 / 0.4664 / 93.87 | 20.31 / 0.3813 / 0.6102 / 95.67 |
| | 4 | - | 25.59 / 0.7481 / 0.3505 / 45.34 | 18.73 / 0.3521 / 0.6220 / 132.2 | 24.76 / 0.5410 / 0.5874 / 79.01 | 17.45 / 0.4266 / 0.5968 / 266.3 | 19.52 / 0.3646 / 0.6271 / 103.8 |
| | | LLE | 25.80 / 0.7206 / 0.3375 / 42.15 | 23.33 / 0.6181 / 0.4289 / 77.52 | 24.81 / 0.7206 / 0.3238 / 53.77 | 17.73 / 0.5207 / 0.4832 / 124.9 | 21.19 / 0.4552 / 0.5668 / 89.76 |
| | 5 | - | 25.99 / 0.7241 / 0.3336 / 41.90 | 22.17 / 0.4567 / 0.5074 / 83.30 | 25.41 / 0.5734 / 0.5463 / 66.85 | 18.17 / 0.4464 / 0.5415 / 179.6 | 20.37 / 0.3998 / 0.5941 / 97.20 |
| | | LLE | 26.06 / 0.7259 / 0.3180 / 37.92 | 24.86 / 0.7163 / 0.3309 / 43.72 | 23.61 / 0.6895 / 0.3530 / 62.67 | 18.90 / 0.5882 / 0.3975 / 53.25 | 21.06 / 0.4977 / 0.5462 / 93.43 |
| | 7 | - | 26.50 / 0.7384 / 0.3120 / 37.90 | 23.24 / 0.5165 / 0.4824 / 79.25 | 25.18 / 0.5595 / 0.5484 / 68.63 | 19.43 / 0.4864 / 0.5262 / 154.1 | 21.30 / 0.4361 / 0.5631 / 93.23 |
| | | LLE | 26.52 / 0.7368 / 0.2982 / 35.04 | 25.77 / 0.7371 / 0.3199 / 46.87 | 24.77 / 0.7054 / 0.3480 / 56.32 | 20.28 / 0.6306 / 0.3726 / 55.38 | 21.54 / 0.5216 / 0.5248 / 95.11 |
| | 10 | - | 27.02 / 0.7519 / 0.2959 / 35.43 | 25.63 / 0.5983 / 0.4114 / 61.43 | 25.36 / 0.5674 / 0.5298 / 65.69 | 20.38 / 0.5234 / 0.4810 / 116.1 | 21.85 / 0.4618 / 0.5415 / 91.58 |
| | | LLE | 26.59 / 0.7474 / 0.2682 / 32.28 | 26.93 / 0.7547 / 0.3030 / 43.32 | 23.85 / 0.6893 / 0.3602 / 66.63 | 21.15 / 0.6660 / 0.3413 / 43.09 | 23.77 / 0.5670 / 0.4787 / 80.82 |
| | 15 | - | 27.53 / 0.7638 / 0.2849 / 34.36 | 26.80 / 0.6203 / 0.3756 / 50.50 | 25.74 / 0.5748 / 0.5128 / 63.87 | 21.66 / 0.5439 / 0.4384 / 81.80 | 22.14 / 0.4773 / 0.5237 / 86.46 |
| | | LLE | 27.16 / 0.7614 / 0.2567 / 30.89 | 28.41 / 0.7963 / 0.2646 / 36.14 | 24.61 / 0.7009 / 0.3726 / 65.55 | 21.18 / 0.6587 / 0.3453 / 47.16 | 24.24 / 0.6317 / 0.4330 / 75.50 |
| $\sigma_{\mathbf{y}} = 0$ | 3 | - | 25.05 / 0.7282 / 0.3388 / 73.97 | 19.37 / 0.4215 / 0.6027 / 138.7 | 29.88 / 0.8563 / 0.2708 / 69.79 | 17.28 / 0.5166 / 0.5589 / 252.0 | 18.93 / 0.3836 / 0.6507 / 113.9 |
| | | LLE | 25.68 / 0.7443 / 0.3223 / 63.79 | 26.23 / 0.7233 / 0.3804 / 79.06 | 29.97 / 0.8575 / 0.2262 / 60.47 | 17.37 / 0.5500 / 0.4876 / 163.3 | 20.50 / 0.4173 / 0.6101 / 99.22 |
| | 4 | - | 25.76 / 0.7426 / 0.3171 / 63.24 | 19.09 / 0.4163 / 0.6047 / 143.6 | 29.22 / 0.8428 / 0.2986 / 77.49 | 17.70 / 0.5127 / 0.5635 / 234.4 | 19.79 / 0.4089 / 0.6222 / 108.2 |
| | | LLE | 26.21 / 0.7543 / 0.3037 / 57.36 | 24.45 / 0.7270 / 0.4007 / 96.56 | 29.34 / 0.8464 / 0.2388 / 65.65 | 18.39 / 0.5508 / 0.5164 / 178.9 | 21.50 / 0.5108 / 0.5595 / 93.50 |
| | 5 | - | 26.17 / 0.7527 / 0.2997 / 57.66 | 23.16 / 0.5778 / 0.4736 / 94.36 | 29.89 / 0.8570 / 0.2477 / 65.75 | 18.58 / 0.5730 / 0.5075 / 187.9 | 20.48 / 0.4436 / 0.5920 / 102.5 |
| | | LLE | 26.49 / 0.7603 / 0.2879 / 51.68 | 26.87 / 0.8064 / 0.3138 / 76.03 | 30.03 / 0.8569 / 0.2155 / 58.20 | 19.25 / 0.6175 / 0.3877 / 76.21 | 21.02 / 0.5293 / 0.5538 / 98.71 |
| | 7 | - | 26.71 / 0.7681 / 0.2766 / 51.51 | 24.25 / 0.6456 / 0.4443 / 89.15 | 30.18 / 0.8582 / 0.2392 / 60.53 | 19.84 / 0.5933 / 0.4933 / 161.6 | 21.60 / 0.4936 / 0.5571 / 101.7 |
| | | LLE | 26.86 / 0.7700 / 0.2669 / 47.85 | 28.00 / 0.8361 / 0.2863 / 73.61 | 30.31 / 0.8626 / 0.1995 / 52.55 | 20.79 / 0.6739 / 0.3387 / 61.20 | 21.52 / 0.5343 / 0.5502 / 102.3 |
| | 10 | - | 27.26 / 0.7822 / 0.2603 / 48.26 | 27.46 / 0.7696 / 0.3469 / 73.03 | 30.70 / 0.8689 / 0.2157 / 53.54 | 20.90 / 0.6469 / 0.4384 / 128.1 | 22.13 / 0.5220 / 0.5350 / 99.85 |
| | | LLE | 27.33 / 0.7812 / 0.2508 / 44.87 | 29.78 / 0.8734 / 0.2386 / 62.33 | 30.82 / 0.8711 / 0.1860 / 47.76 | 21.69 / 0.7082 / 0.3149 / 56.38 | 23.22 / 0.6270 / 0.4790 / 91.95 |
| | 15 | - | 27.82 / 0.7947 / 0.2494 / 46.16 | 29.45 / 0.8043 / 0.2929 / 55.60 | 31.13 / 0.8756 / 0.1921 / 44.43 | 22.44 / 0.6836 / 0.3832 / 92.37 | 22.54 / 0.5430 / 0.5141 / 94.05 |
| | | LLE | 27.79 / 0.7912 / 0.2406 / 43.11 | 31.47 / 0.9002 / 0.1967 / 48.89 | 31.09 / 0.8765 / 0.1805 / 45.08 | 23.75 / 0.7666 / 0.2718 / 46.32 | 24.71 / 0.6902 / 0.4136 / 70.95 |

## Table 31: Results of RED-diff on FFHQ Dataset.

| Condition | Steps | Strategy | Deblur (aniso) | Inpainting | 4× SR | CS 50% | Deblur (nonlinear) |
|---|---|---|---|---|---|---|---|
| | | | | | PSNR↑/SSIM↑/LPIPS↓/FID↓ | | |
| $\sigma_{\mathbf{y}} = 0.05$ | 3 | - | 25.18 / **0.6954** / 0.3956 / 117.8 | 18.01 / 0.3596 / 0.6404 / 191.9 | 24.96 / 0.5636 / 0.5784 / 143.8 | 16.88 / 0.4098 / 0.6067 / 352.1 | 15.86 / 0.1890 / 0.7677 / 280.0 |
| | | LLE | **25.20** / 0.6942 / **0.3931** / **115.6** | **23.64** / **0.5499** / **0.4579** / **140.2** | **25.63** / **0.7370** / **0.3078** / **100.6** | **17.06** / **0.5046** / **0.5350** / 242.6 | **16.61** / **0.2230** / **0.7453** / 268.5 |
| | 4 | - | 25.60 / 0.7007 / 0.3871 / 112.7 | 18.43 / 0.3542 / 0.6352 / 198.2 | **24.23** / 0.5267 / 0.6176 / 162.2 | 17.02 / 0.4085 / 0.6192 / 365.3 | 16.30 / 0.1992 / 0.7538 / **252.2** |
| | | LLE | **25.65** / **0.7055** / **0.3819** / **109.8** | **22.56** / **0.5984** / **0.4585** / 164.1 | 23.99 / **0.6948** / **0.3631** / **126.4** | **17.47** / **0.4978** / **0.5433** / 229.6 | **16.55** / **0.2403** / **0.7412** / 268.0 |
| | 5 | - | **25.94** / 0.7098 / 0.3768 / 108.3 | 20.44 / 0.4367 / 0.5656 / 172.3 | **24.90** / 0.5621 / 0.5675 / 138.9 | 17.72 / 0.4386 / 0.5682 / 275.6 | 17.64 / 0.2603 / 0.7087 / 228.4 |
| | | LLE | 25.91 / **0.7111** / **0.3748** / **107.0** | **23.85** / **0.6701** / **0.4017** / **133.5** | 23.11 / **0.6701** / **0.3983** / 137.6 | **18.44** / **0.5815** / **0.4290** / **130.2** | **18.35** / **0.3275** / **0.6821** / **217.5** |
| | 7 | - | **26.26** / 0.7164 / 0.3708 / 107.0 | 22.02 / 0.4873 / 0.5188 / 158.1 | **24.67** / 0.5456 / 0.5801 / 145.1 | 18.31 / 0.4563 / 0.5634 / 254.3 | 18.60 / 0.3010 / 0.6801 / 222.0 |
| | | LLE | 25.81 / **0.7275** / **0.3450** / **100.2** | **24.87** / **0.6883** / **0.3890** / **129.2** | 24.00 / **0.6873** / **0.3770** / **128.2** | **19.23** / **0.6104** / **0.4185** / **131.5** | **19.32** / **0.4085** / **0.6317** / **210.8** |
| | 10 | - | **26.61** / 0.7262 / 0.3606 / 104.5 | 24.22 / 0.5665 / 0.4519 / 136.2 | **24.85** / 0.5540 / 0.5596 / 140.1 | 19.00 / 0.4880 / 0.5220 / 202.5 | 19.26 / 0.3303 / 0.6572 / 217.1 |
| | | LLE | 26.27 / **0.7409** / **0.3313** / **95.36** | **25.85** / **0.7361** / **0.3499** / **117.4** | 23.38 / **0.6745** / **0.3888** / 134.2 | **20.08** / **0.6384** / **0.3973** / **123.5** | **20.89** / **0.4905** / **0.5779** / **194.7** |
| | 15 | - | **26.96** / 0.7257 / 0.3612 / 105.5 | 26.21 / 0.6343 / 0.3939 / 115.5 | **25.03** / 0.5635 / 0.5367 / 131.6 | 20.26 / 0.5126 / 0.4793 / 158.5 | 20.05 / 0.3619 / 0.6336 / 208.1 |
| | | LLE | 26.77 / **0.7560** / **0.3169** / **93.77** | **27.27** / **0.7637** / **0.3221** / **106.3** | 23.75 / **0.6795** / **0.4021** / 136.7 | **21.37** / **0.6884** / **0.3546** / **107.0** | **21.75** / **0.5402** / **0.5348** / **181.8** |
| $\sigma_{\mathbf{y}} = 0$ | 3 | - | 25.49 / 0.7478 / **0.3323** / 122.4 | 18.24 / 0.4060 / 0.6313 / 200.9 | **29.00** / **0.8458** / 0.2976 / 124.6 | 17.17 / 0.5166 / 0.5716 / 334.8 | 15.96 / 0.2022 / 0.7660 / 290.4 |
| | | LLE | **25.66** / **0.7514** / 0.3324 / **121.3** | **24.57** / **0.6405** / **0.4262** / **143.1** | 28.80 / 0.8413 / **0.2687** / **113.4** | **17.38** / **0.5313** / **0.5663** / 296.6 | **16.71** / **0.2406** / **0.7440** / **274.5** |
| | 4 | - | 25.97 / 0.7571 / 0.3175 / 114.7 | 18.74 / 0.4168 / 0.6232 / 206.0 | 28.39 / 0.8306 / 0.3580 / 141.0 | 17.29 / 0.5015 / 0.5875 / 339.4 | 16.28 / 0.2148 / 0.7534 / **249.9** |
| | | LLE | **26.12** / **0.7618** / **0.3148** / **111.8** | **23.52** / **0.6709** / **0.4502** / 167.3 | **29.29** / **0.8506** / **0.2320** / **106.3** | **17.93** / **0.5311** / **0.5562** / 278.4 | **16.59** / **0.2533** / **0.7414** / 270.9 |
| | 5 | - | 26.33 / 0.7663 / 0.3075 / **107.0** | 20.86 / 0.5118 / 0.5480 / 180.4 | **29.12** / **0.8475** / 0.2810 / 121.8 | 18.08 / 0.5589 / 0.5305 / 284.6 | 17.79 / 0.2863 / 0.7059 / 227.6 |
| | | LLE | **26.38** / **0.7678** / **0.3068** / 107.5 | **25.15** / **0.7447** / **0.3889** / 147.3 | 28.26 / 0.8277 / **0.2497** / **112.8** | **18.70** / **0.6081** / **0.4232** / **154.2** | **18.29** / **0.3438** / **0.6848** / **223.4** |
| | 7 | - | 26.69 / 0.7743 / 0.2975 / **99.61** | 22.73 / 0.6013 / 0.4893 / 164.2 | **29.33** / **0.8485** / 0.2855 / 118.7 | 18.69 / 0.5699 / 0.5293 / 260.7 | 18.82 / 0.3308 / 0.6791 / 228.5 |
| | | LLE | **26.81** / **0.7798** / **0.2913** / 101.7 | **26.24** / **0.7773** / **0.3677** / **142.9** | 28.90 / 0.8407 / **0.2231** / **95.31** | **19.74** / **0.6544** / **0.3948** / **146.4** | **19.02** / **0.4136** / **0.6400** / **217.4** |
| | 10 | - | 27.08 / 0.7854 / 0.2867 / 97.51 | 25.40 / 0.7172 / 0.4063 / 144.2 | 29.84 / 0.8608 / 0.2454 / 106.7 | 19.44 / 0.6180 / 0.4795 / 221.1 | 19.41 / 0.3667 / 0.6584 / 221.7 |
| | | LLE | **27.20** / **0.7900** / **0.2802** / **96.56** | **27.67** / **0.8266** / **0.3199** / **131.8** | **30.04** / **0.8650** / **0.2072** / **92.72** | **20.57** / **0.6920** / **0.3534** / **128.5** | **20.85** / **0.4835** / **0.6021** / **199.1** |
| | 15 | - | 27.53 / 0.7885 / 0.2940 / **92.26** | 28.22 / 0.8244 / 0.3162 / 125.3 | 30.39 / 0.8710 / 0.2141 / 91.68 | 20.89 / 0.6559 / 0.4248 / 173.0 | 20.38 / 0.4195 / 0.6260 / 210.9 |
| | | LLE | **27.61** / **0.8000** / **0.2717** / 96.10 | **29.38** / **0.8666** / **0.2670** / **111.5** | **30.48** / **0.8749** / **0.1915** / **82.37** | **22.05** / **0.7443** / **0.3090** / **103.6** | **21.66** / **0.5721** / **0.5423** / **187.6** |

## Table 32: Results of DiffPIR on CelebA-HQ Dataset.

| Condition | Steps | Strategy | Deblur (aniso) | Inpainting | 4× SR | CS 50% | Deblur (nonlinear) |
|---|---|---|---|---|---|---|---|
| | | | | | PSNR↑/SSIM↑/LPIPS↓/FID↓ | | |
| $\sigma_{\mathbf{y}} = 0.05$ | 3 | - | **23.06** / **0.3875** / **0.5373** / **84.09** | 18.66 / 0.3550 / 0.6248 / 126.3 | 26.48 / 0.6443 / 0.4647 / 58.25 | 17.47 / 0.4416 / 0.5813 / 253.7 | 21.69 / 0.3931 / 0.5513 / **65.90** |
| | | LLE | 22.81 / 0.3754 / 0.5532 / 90.45 | **24.53** / **0.5645** / **0.4472** / **70.36** | **26.84** / **0.7231** / **0.3447** / **44.71** | **17.82** / **0.4474** / **0.5642** / 210.1 | **21.70** / **0.4024** / **0.5464** / 66.11 |
| | 4 | - | 25.62 / 0.5245 / 0.4300 / 57.67 | 21.93 / 0.4746 / 0.5143 / 85.17 | **27.65** / 0.7250 / 0.3563 / 44.26 | 18.14 / 0.4835 / 0.5340 / 185.7 | **23.48** / **0.4632** / **0.4970** / 59.23 |
| | | LLE | **26.71** / **0.6029** / **0.3949** / **54.77** | **25.97** / **0.6365** / **0.3924** / **57.75** | 27.59 / **0.7860** / **0.2525** / **37.77** | **18.71** / **0.4987** / **0.5061** / 141.5 | 23.28 / 0.4612 / 0.4997 / 59.95 |
| | 5 | - | 27.17 / 0.6260 / 0.3578 / 45.61 | 24.25 / 0.5692 / 0.4362 / 65.81 | **28.06** / 0.7655 / 0.2945 / 39.96 | 18.67 / 0.5151 / 0.4957 / 143.0 | **24.56** / 0.5131 / **0.4593** / **55.37** |
| | | LLE | **28.42** / **0.7437** / **0.2910** / **34.35** | **27.31** / **0.6958** / **0.3431** / **46.98** | 27.63 / **0.7879** / **0.2439** / **37.90** | **19.35** / **0.5484** / **0.4540** / 104.4 | 24.47 / **0.5157** / 0.4603 / 56.22 |
| | 7 | - | 28.50 / 0.7384 / 0.2737 / 31.81 | 27.27 / 0.6999 / 0.3385 / 46.75 | **28.17** / **0.7932** / 0.2462 / **34.03** | 20.00 / 0.5776 / 0.4262 / 93.28 | 25.83 / 0.5841 / 0.4046 / 48.87 |
| | | LLE | **28.69** / **0.7914** / **0.2309** / **27.58** | **28.55** / **0.7626** / **0.2883** / **37.34** | 27.47 / 0.7848 / **0.2389** / 39.92 | **20.69** / **0.6287** / **0.3731** / **62.78** | **26.11** / **0.6064** / **0.3947** / **48.49** |
| | 10 | - | **29.01** / 0.7937 / 0.2262 / **25.21** | 28.85 / 0.7693 / 0.2793 / 36.39 | **28.08** / 0.7978 / 0.2329 / 36.90 | 20.69 / 0.6361 / 0.3692 / 67.68 | 27.14 / 0.6632 / 0.3446 / 41.48 |
| | | LLE | 28.68 / **0.7998** / **0.2175** / 32.83 | **29.47** / **0.8174** / **0.2365** / **29.25** | 27.33 / 0.7798 / 0.2421 / 40.84 | **21.41** / **0.6874** / **0.3153** / **43.70** | **27.44** / **0.6949** / **0.3271** / **38.61** |
| | 15 | - | **29.03** / **0.8094** / **0.2098** / 32.92 | 30.19 / 0.8328 / 0.2199 / 26.73 | **27.89** / **0.7941** / 0.2354 / 42.00 | 22.02 / 0.7085 / 0.3010 / 43.44 | 28.21 / 0.7474 / 0.2743 / 30.64 |
| | | LLE | 28.47 / 0.7978 / 0.2190 / 39.10 | **30.07** / **0.8494** / **0.1969** / **25.11** | 27.30 / 0.7791 / 0.2452 / 41.51 | **22.91** / **0.7407** / **0.2673** / **35.31** | **28.34** / **0.7689** / **0.2616** / **27.66** |
| $\sigma_{\mathbf{y}} = 0$ | 3 | - | 28.29 / 0.7936 / 0.2392 / 32.79 | 18.99 / 0.4032 / 0.6115 / 144.1 | 30.05 / 0.8580 / 0.2513 / 62.85 | 17.50 / **0.4528** / 0.6015 / 276.6 | 24.05 / 0.5484 / 0.4729 / 61.26 |
| | | LLE | **33.05** / **0.8750** / **0.1967** / **24.81** | **25.85** / **0.7013** / **0.3956** / **79.35** | **30.10** / **0.8587** / **0.2395** / **59.83** | **17.67** / 0.4462 / **0.6013** / 273.9 | **24.73** / **0.6333** / **0.4279** / **51.54** |
| | 4 | - | 32.47 / 0.8373 / 0.2101 / 25.49 | 22.66 / 0.5476 / 0.4910 / 100.4 | **30.58** / **0.8701** / 0.2160 / 55.17 | 18.09 / **0.4795** / 0.5667 / 217.5 | 25.60 / 0.5998 / 0.4392 / 56.03 |
| | | LLE | **34.23** / **0.9104** / **0.1674** / **22.43** | **27.90** / **0.8067** / **0.3217** / **72.44** | 30.53 / 0.8676 / **0.2051** / **50.57** | **18.30** / 0.4714 / **0.5650** / 204.8 | **26.61** / **0.7073** / **0.3793** / **44.29** |
| | 5 | - | 32.49 / 0.8340 / 0.2068 / 24.44 | 25.49 / 0.6656 / 0.4030 / 79.67 | **30.76** / **0.8735** / 0.2002 / 50.05 | 18.55 / **0.5004** / 0.5411 / 184.4 | 26.41 / 0.6302 / 0.4182 / 53.55 |
| | | LLE | **34.33** / **0.9131** / **0.1655** / **22.88** | **29.48** / **0.8545** / **0.2678** / **63.57** | 30.67 / 0.8704 / **0.1901** / **45.68** | **18.81** / 0.4921 / **0.5352** / 168.5 | **27.77** / **0.7494** / **0.3503** / **39.87** |
| | 7 | - | 32.49 / 0.8340 / 0.2068 / 23.52 | 29.46 / 0.8375 / 0.2768 / 60.15 | **30.95** / **0.8769** / 0.1824 / 43.58 | 19.65 / **0.5385** / 0.4978 / 147.6 | 27.02 / 0.6538 / 0.4001 / 50.66 |
| | | LLE | **34.60** / **0.9158** / **0.1613** / **22.25** | **31.18** / **0.8934** / **0.2098** / **50.47** | 30.77 / 0.8726 / **0.1740** / **40.27** | **19.91** / 0.5351 / **0.4859** / 129.2 | **29.20** / **0.8059** / **0.3067** / **34.06** |
| | 10 | - | 32.44 / 0.8307 / 0.2063 / 22.71 | 31.49 / 0.8903 / 0.2131 / 47.51 | **31.06** / **0.8787** / 0.1694 / 38.51 | 20.24 / 0.5716 / 0.4613 / 125.1 | 27.53 / 0.6725 / 0.3862 / 49.79 |
| | | LLE | **34.92** / **0.9219** / **0.1541** / **20.71** | **32.34** / **0.9125** / **0.1730** / **39.62** | 30.87 / 0.8744 / **0.1641** / **36.28** | **20.53** / 0.5714 / **0.4493** / 111.2 | **29.98** / **0.8346** / **0.2824** / **30.94** |
| | 15 | - | 32.35 / 0.8255 / 0.2071 / 22.14 | 33.16 / 0.9240 / 0.1528 / 33.82 | **31.15** / **0.8802** / 0.1596 / 34.04 | 21.32 / 0.6156 / 0.4143 / 100.1 | 27.73 / 0.6794 / 0.3797 / 49.12 |
| | | LLE | **35.28** / **0.9269** / **0.1472** / **19.44** | **33.35** / **0.9261** / **0.1430** / **30.33** | 30.95 / 0.8756 / **0.1572** / **32.61** | **21.85** / **0.6252** / **0.4000** / 88.51 | **30.60** / **0.8578** / **0.2593** / **28.15** |

## Table 33: Results of DiffPIR on FFHQ Dataset.

| Condition | Steps | Strategy | Deblur (aniso) | Inpainting | 4× SR | CS 50% | Deblur (nonlinear) |
|---|---|---|---|---|---|---|---|
| | | | | | PSNR↑/SSIM↑/LPIPS↓/FID↓ | | |
| $\sigma_{\mathbf{y}} = 0.05$ | 3 | - | **22.29** / **0.3639** / **0.5628** / **150.2** | 18.62 / 0.3515 / 0.6316 / 183.4 | 25.49 / 0.5982 / 0.5035 / 131.1 | 17.01 / 0.4126 / 0.6137 / 360.1 | **19.28** / 0.3002 / 0.6489 / 166.5 |
| | | LLE | 22.16 / 0.3558 / 0.5783 / 156.3 | **23.33** / **0.4981** / **0.4854** / **140.1** | **25.75** / **0.6750** / **0.4029** / **113.9** | **17.28** / **0.4151** / **0.5980** / 305.2 | 19.13 / 0.3002 / **0.6461** / **166.3** |
| | 4 | - | 24.22 / 0.4153 / 0.4477 / 121.7 | 21.47 / 0.4555 / 0.5368 / 153.6 | **26.57** / 0.6877 / 0.3921 / 109.9 | 17.31 / 0.4469 / 0.5729 / 292.2 | **20.30** / **0.3547** / 0.5972 / **149.6** |
| | | LLE | **25.63** / **0.5569** / **0.4323** / **121.1** | **24.70** / **0.5744** / **0.4353** / **131.0** | 26.37 / **0.7470** / **0.2974** / **91.33** | **17.75** / **0.4514** / **0.5551** / 244.5 | 20.26 / 0.3502 / **0.5987** / 154.1 |
| | 5 | - | 25.64 / 0.5484 / 0.4153 / 109.4 | 23.41 / 0.5333 / 0.4716 / 136.0 | **26.92** / 0.7306 / 0.3331 / 98.21 | 17.86 / 0.4772 / 0.5371 / 243.1 | **21.27** / **0.4055** / **0.5504** / **141.3** |
| | | LLE | **26.70** / **0.6520** / **0.3705** / **106.6** | **25.74** / **0.6244** / **0.3986** / **115.5** | 26.41 / **0.7549** / **0.2850** / **89.33** | **18.37** / **0.4905** / **0.5101** / 201.2 | 21.17 / 0.3961 / 0.5585 / 142.4 |
| | 7 | - | 27.10 / 0.6608 / 0.3436 / 93.54 | 25.98 / 0.6488 / 0.3851 / 112.9 | **27.04** / **0.7624** / 0.2826 / 86.07 | 18.69 / 0.5282 / 0.4828 / 188.7 | **22.71** / 0.4837 / **0.4768** / 129.9 |
| | | LLE | **27.63** / **0.7511** / **0.2948** / **84.04** | **27.05** / **0.6972** / **0.3488** / **101.7** | 26.26 / 0.7547 / **0.2788** / 91.59 | **19.15** / **0.5522** / **0.4485** / **161.9** | 22.55 / **0.4844** / 0.4804 / **128.7** |
| | 10 | - | **27.98** / 0.7499 / 0.2845 / 81.92 | 27.48 / 0.7227 / 0.3304 / 98.12 | **26.94** / **0.7701** / 0.2707 / 83.40 | 19.13 / 0.5834 / 0.4246 / 158.4 | 23.53 / 0.5524 / 0.4214 / 119.3 |
| | | LLE | 27.83 / **0.7790** / **0.2600** / **75.12** | **28.07** / **0.7616** / **0.3040** / **95.73** | 26.10 / 0.7515 / **0.2794** / 88.93 | **19.86** / **0.6205** / **0.3850** / **131.4** | **23.58** / **0.5706** / **0.4105** / **117.2** |
| | 15 | - | **28.25** / **0.7929** / 0.2454 / **72.71** | 28.90 / 0.7957 / 0.2787 / 85.58 | **26.80** / **0.7687** / 0.2732 / 87.71 | 20.43 / 0.6613 / 0.3541 / 118.5 | 24.40 / 0.6331 / 0.3675 / 110.8 |
| | | LLE | 27.79 / 0.7883 / **0.2463** / 78.74 | **28.98** / **0.8214** / **0.2470** / **77.55** | 26.01 / 0.7502 / 0.2887 / 96.92 | **20.91** / **0.6871** / **0.3231** / **98.02** | **24.58** / **0.6506** / **0.3533** / **104.8** |
| $\sigma_{\mathbf{y}} = 0$ | 3 | - | 31.32 / 0.8131 / 0.2393 / 64.89 | 18.97 / 0.4095 / 0.6174 / 196.2 | 28.69 / 0.8081 / 0.3177 / 114.2 | 17.01 / 0.4249 / 0.6297 / **384.5** | 20.16 / 0.3585 / 0.6371 / 188.1 |
| | | LLE | **33.05** / **0.8934** / **0.1888** / **60.10** | **24.54** / **0.6356** / **0.4354** / **144.6** | **28.97** / **0.8314** / **0.2950** / **113.8** | **17.21** / **0.4355** / **0.6241** / 386.3 | **20.56** / **0.4359** / **0.5950** / **161.8** |
| | 4 | - | 31.87 / 0.8283 / 0.2253 / 59.87 | 22.17 / 0.5407 / 0.5135 / 161.6 | **29.53** / **0.8519** / 0.2565 / 110.1 | 17.25 / 0.4463 / 0.6009 / 328.5 | 20.84 / 0.3908 / 0.6138 / 176.3 |
| | | LLE | **33.79** / **0.9082** / **0.1716** / **52.60** | **26.52** / **0.7557** / **0.3723** / **135.2** | 29.49 / 0.8508 / **0.2439** / **102.2** | **18.08** / **0.5483** / **0.5245** / 261.2 | **21.63** / **0.5008** / **0.5482** / **146.9** |
| | 5 | - | 31.98 / 0.8292 / 0.2220 / 55.63 | 24.48 / 0.6396 / 0.4412 / 143.8 | **29.79** / **0.8608** / 0.2300 / 100.9 | 17.74 / 0.4680 / 0.5761 / 290.8 | 21.51 / 0.4175 / 0.5971 / 171.5 |
| | | LLE | **34.02** / **0.9108** / **0.1677** / **51.00** | **27.63** / **0.8034** / **0.3301** / **125.0** | 29.74 / 0.8584 / **0.2203** / **97.14** | **18.00** / **0.4697** / **0.5708** / 281.2 | **22.42** / **0.5518** / **0.5143** / **135.5** |
| | 7 | - | 32.07 / 0.8290 / 0.2182 / 53.50 | 27.75 / 0.7916 / 0.3334 / 121.8 | **30.09** / **0.8683** / 0.2014 / 89.10 | 18.39 / 0.4976 / 0.5452 / 255.0 | 22.58 / 0.4633 / 0.5614 / 159.1 |
| | | LLE | **34.22** / **0.9157** / **0.1638** / **50.32** | **29.24** / **0.8523** / **0.2766** / **107.2** | 29.95 / 0.8646 / **0.1939** / **84.07** | **18.61** / **0.4986** / **0.5381** / 240.5 | **23.50** / **0.6217** / **0.4667** / **125.7** |
| | 10 | - | 32.10 / 0.8280 / 0.2155 / 51.10 | 29.65 / 0.8550 / 0.2705 / 102.0 | **30.31** / **0.8737** / 0.1823 / 79.25 | 18.79 / 0.5294 / 0.5083 / 224.6 | 22.97 / 0.4806 / 0.5480 / 155.2 |
| | | LLE | **34.60** / **0.9233** / **0.1547** / **47.94** | **30.43** / **0.8815** / **0.2365** / **94.71** | 30.15 / 0.8698 / **0.1767** / **72.71** | **19.16** / **0.5337** / **0.4987** / 215.1 | **24.27** / **0.6493** / **0.4422** / **118.0** |
| | 15 | - | 32.08 / 0.8243 / 0.2150 / 49.78 | 31.39 / 0.8998 / 0.2066 / 82.40 | **30.54** / **0.8786** / 0.1651 / 65.61 | 19.88 / 0.5743 / 0.4630 / 187.3 | 23.38 / 0.5037 / 0.5306 / 150.0 |
| | | LLE | **34.95** / **0.9294** / **0.1462** / **45.32** | **31.67** / **0.9059** / **0.1932** / **75.05** | 30.36 / 0.8751 / **0.1608** / **59.14** | **20.12** / **0.5836** / **0.4489** / 173.9 | **24.98** / **0.6911** / **0.4183** / **110.8** |

Table 34: Results of ReSample on CelebA-HQ Dataset.

| Condition | Steps | Strategy | Deblur (aniso) | Inpainting | 4× SR | CS 50% | Deblur (nonlinear) |
|---|---|---|---|---|---|---|---|
| | | | PSNR↑/SSIM↑/LPIPS↓/FID↓ | | | | |
| $\sigma_{\mathbf{y}} = 0.05$ | 3 | - | 24.38 / 0.4646 / 0.5055 / 82.32 | 18.53 / 0.3401 / 0.6240 / 126.7 | 22.53 / 0.5515 / 0.6173 / 83.00 | 16.97 / 0.4086 / 0.5922 / 274.4 | 20.75 / 0.3356 / 0.5843 / 69.70 |
| | | LLE | 26.33 / 0.6132 / 0.4034 / 52.55 | 24.28 / 0.5289 / 0.4467 / 68.89 | 25.64 / 0.6811 / 0.3975 / 57.69 | 17.37 / 0.4164 / 0.5717 / 220.4 | 21.03 / 0.3533 / 0.5776 / 68.11 |
| | 4 | - | 24.58 / 0.4692 / 0.4925 / 78.89 | 22.21 / 0.4542 / 0.5113 / 82.43 | 24.44 / 0.5884 / 0.5592 / 71.05 | 17.90 / 0.4457 / 0.5445 / 186.4 | 21.78 / 0.3637 / 0.5511 / 63.35 |
| | | LLE | 26.59 / 0.6082 / 0.3913 / 51.40 | 25.68 / 0.5865 / 0.4014 / 57.20 | 25.71 / 0.7205 / 0.3172 / 44.49 | 18.77 / 0.4801 / 0.5011 / 129.6 | 22.60 / 0.4064 / 0.5232 / 61.20 |
| | 5 | - | 24.65 / 0.4723 / 0.4862 / 77.15 | 33.05 / 0.8750 / 0.1967 / 24.81 | 25.49 / 0.6101 / 0.5219 / 66.17 | 18.85 / 0.4763 / 0.5042 / 138.3 | 22.32 / 0.3813 / 0.5318 / 60.46 |
| | | LLE | 26.62 / 0.6046 / 0.3893 / 51.56 | 26.67 / 0.6266 / 0.3690 / 49.35 | 25.71 / 0.7064 / 0.3400 / 50.60 | 20.01 / 0.5173 / 0.4586 / 95.23 | 23.39 / 0.4354 / 0.4938 / 59.33 |
| | 7 | - | 24.76 / 0.4775 / 0.4778 / 74.75 | 26.80 / 0.6177 / 0.3687 / 48.70 | 26.39 / 0.6359 / 0.4795 / 60.77 | 21.27 / 0.5335 / 0.4356 / 83.15 | 22.70 / 0.3949 / 0.5184 / 59.94 |
| | | LLE | 26.57 / 0.5986 / 0.3850 / 50.77 | 27.44 / 0.6455 / 0.3467 / 43.48 | 27.00 / 0.7143 / 0.3416 / 46.86 | 22.24 / 0.5662 / 0.4017 / 63.13 | 23.82 / 0.4496 / 0.4808 / 58.36 |
| | 10 | - | 24.90 / 0.4840 / 0.4707 / 72.60 | 27.57 / 0.6412 / 0.3437 / 42.99 | 26.83 / 0.6507 / 0.4535 / 57.53 | 22.24 / 0.5618 / 0.3999 / 64.28 | 23.06 / 0.4070 / 0.5069 / 58.38 |
| | | LLE | 26.56 / 0.5946 / 0.3837 / 51.11 | 27.94 / 0.6597 / 0.3305 / 40.03 | 27.45 / 0.7247 / 0.3345 / 44.12 | 23.51 / 0.5989 / 0.3697 / 51.44 | 24.06 / 0.4571 / 0.4706 / 57.68 |
| | 15 | - | 25.12 / 0.4958 / 0.4607 / 69.47 | 28.02 / 0.6554 / 0.3266 / 39.41 | 27.18 / 0.6659 / 0.4303 / 54.99 | 24.18 / 0.5985 / 0.3618 / 48.68 | 23.29 / 0.4150 / 0.4999 / 57.65 |
| | | LLE | 26.54 / 0.5913 / 0.3841 / 51.67 | 28.28 / 0.6688 / 0.3179 / 37.92 | 27.74 / 0.7278 / 0.3331 / 43.08 | 25.07 / 0.6258 / 0.3428 / 43.29 | 24.11 / 0.4562 / 0.4708 / 57.79 |
| $\sigma_{\mathbf{y}} = 0$ | 3 | - | 29.53 / 0.8407 / 0.2367 / 33.68 | 18.97 / 0.4100 / 0.6083 / 143.9 | 23.49 / 0.7407 / 0.3827 / 83.70 | 17.28 / 0.5204 / 0.5531 / 251.9 | 23.37 / 0.5359 / 0.4908 / 66.63 |
| | | LLE | 29.84 / 0.8459 / 0.2284 / 32.06 | 26.21 / 0.7144 / 0.3837 / 78.65 | 28.40 / 0.8284 / 0.2704 / 71.5 | 17.70 / 0.5242 / 0.5336 / 221.7 | 24.40 / 0.5976 / 0.4574 / 61.64 |
| | 4 | - | 30.20 / 0.8534 / 0.2131 / 28.83 | 23.28 / 0.5739 / 0.4723 / 96.05 | 26.04 / 0.7850 / 0.3078 / 79.23 | 18.29 / 0.5763 / 0.5031 / 189.5 | 25.71 / 0.6473 / 0.4111 / 54.24 |
| | | LLE | 30.28 / 0.8537 / 0.2053 / 27.44 | 28.54 / 0.8341 / 0.2980 / 71.72 | 28.90 / 0.8362 / 0.2422 / 64.34 | 19.25 / 0.5938 / 0.4673 / 149.7 | 27.60 / 0.7450 / 0.3426 / 44.89 |
| | 5 | - | 30.46 / 0.8575 / 0.2047 / 27.67 | 26.70 / 0.7159 / 0.3720 / 75.57 | 27.66 / 0.8119 / 0.2694 / 73.56 | 19.36 / 0.6244 / 0.4564 / 150.2 | 27.23 / 0.7204 / 0.3497 / 44.37 |
| | | LLE | 30.44 / 0.8557 / 0.1970 / 26.14 | 30.17 / 0.8727 / 0.2470 / 60.69 | 29.03 / 0.8389 / 0.2322 / 62.54 | 20.65 / 0.6527 / 0.4132 / 115.4 | 29.30 / 0.8135 / 0.2761 / 33.22 |
| | 7 | - | 30.75 / 0.8625 / 0.1963 / 26.41 | 30.74 / 0.8750 / 0.2436 / 55.94 | 29.21 / 0.8433 / 0.2342 / 63.76 | 22.23 / 0.7184 / 0.3626 / 96.49 | 28.68 / 0.7859 / 0.2870 / 35.41 |
| | | LLE | 30.65 / 0.8591 / 0.1887 / 25.14 | 31.85 / 0.9051 / 0.1964 / 47.63 | 29.61 / 0.8507 / 0.2105 / 55.46 | 23.38 / 0.7425 / 0.3259 / 76.04 | 29.87 / 0.8286 / 0.2624 / 31.42 |
| | 10 | - | 31.03 / 0.8670 / 0.1906 / 25.99 | 32.76 / 0.9165 / 0.1785 / 40.94 | 30.04 / 0.8582 / 0.2141 / 56.20 | 23.50 / 0.7679 / 0.3058 / 71.95 | 29.89 / 0.8329 / 0.2374 / 27.74 |
| | | LLE | 30.86 / 0.8623 / 0.1833 / 24.38 | 33.16 / 0.9232 / 0.1616 / 36.20 | 30.16 / 0.8605 / 0.1986 / 50.66 | 25.08 / 0.7952 / 0.2735 / 56.91 | 30.73 / 0.8607 / 0.2210 / 24.42 |
| | 15 | - | 31.34 / 0.8719 / 0.1868 / 25.86 | 34.19 / 0.9352 / 0.1365 / 28.69 | 30.68 / 0.8686 / 0.2020 / 50.79 | 26.26 / 0.8305 / 0.2403 / 46.52 | 30.88 / 0.8690 / 0.1945 / 21.98 |
| | | LLE | 31.19 / 0.8683 / 0.1815 / 24.62 | 34.28 / 0.9357 / 0.1320 / 27.17 | 30.62 / 0.8675 / 0.1918 / 47.60 | 27.46 / 0.8463 / 0.2223 / 39.28 | 31.25 / 0.8778 / 0.1924 / 20.73 |

Table 35: Results of ReSample on FFHQ Dataset.

| Condition | Steps | Strategy | Deblur (aniso) | Inpainting | 4× SR | CS 50% | Deblur (nonlinear) |
|---|---|---|---|---|---|---|---|
| | | | PSNR↑/SSIM↑/LPIPS↓/FID↓ | | | | |
| $\sigma_{\mathbf{y}} = 0.05$ | 3 | - | 23.92 / 0.4564 / 0.5266 / 148.7 | 18.65 / 0.3506 / 0.6278 / 186.8 | 22.55 / 0.5577 / 0.6204 / 161.8 | 16.83 / 0.4080 / 0.6099 / 361.7 | 18.64 / 0.2604 / 0.6749 / 187.1 |
| | | LLE | 25.88 / 0.6152 / 0.4221 / 116.8 | 23.38 / 0.4981 / 0.4780 / 141.1 | 25.06 / 0.6676 / 0.4180 / 124.7 | 17.14 / 0.4138 / 0.5951 / 314.6 | 18.86 / 0.2714 / 0.6730 / 183.3 |
| | 4 | - | 24.10 / 0.4605 / 0.5159 / 145.7 | 21.88 / 0.4512 / 0.5299 / 146.1 | 24.16 / 0.5853 / 0.5760 / 142.3 | 17.57 / 0.4410 / 0.5679 / 269.2 | 19.97 / 0.2892 / 0.6388 / 164.1 |
| | | LLE | 26.20 / 0.6140 / 0.4114 / 114.4 | 25.09 / 0.5881 / 0.4243 / 126.2 | 25.19 / 0.7083 / 0.3492 / 112.1 | 18.32 / 0.4758 / 0.5307 / 210.5 | 20.13 / 0.2977 / 0.6447 / 165.2 |
| | 5 | - | 24.18 / 0.4632 / 0.5100 / 141.7 | 24.03 / 0.5267 / 0.4625 / 127.9 | 24.98 / 0.6021 / 0.5481 / 140.6 | 18.25 / 0.4668 / 0.5328 / 222.0 | 20.85 / 0.3177 / 0.6036 / 149.0 |
| | | LLE | 26.28 / 0.6117 / 0.4015 / 115.7 | 25.95 / 0.6139 / 0.3987 / 113.4 | 25.02 / 0.6957 / 0.3706 / 121.9 | 19.17 / 0.5063 / 0.4921 / 174.4 | 21.00 / 0.3309 / 0.6019 / 151.0 |
| | 7 | - | 24.30 / 0.4678 / 0.5021 / 141.6 | 26.01 / 0.5996 / 0.3998 / 111.0 | 25.79 / 0.6224 / 0.5119 / 131.0 | 19.80 / 0.5137 / 0.4722 / 159.2 | 21.30 / 0.3329 / 0.5847 / 143.9 |
| | | LLE | 26.27 / 0.6051 / 0.4054 / 114.0 | 26.85 / 0.6376 / 0.3727 / 101.4 | 26.30 / 0.7073 / 0.3716 / 113.4 | 20.60 / 0.5484 / 0.4401 / 139.1 | 21.77 / 0.3670 / 0.5727 / 143.4 |
| | 10 | - | 24.44 / 0.4749 / 0.4931 / 137.4 | 26.91 / 0.6275 / 0.3697 / 96.86 | 26.24 / 0.6376 / 0.4825 / 125.3 | 21.00 / 0.5457 / 0.4314 / 131.3 | 21.61 / 0.3478 / 0.5667 / 138.1 |
| | | LLE | 26.30 / 0.6015 / 0.4035 / 113.6 | 27.48 / 0.6551 / 0.3529 / 93.98 | 26.63 / 0.7179 / 0.3600 / 109.8 | 21.94 / 0.5831 / 0.4013 / 114.5 | 22.30 / 0.3916 / 0.5497 / 139.0 |
| | 15 | - | 24.69 / 0.4871 / 0.4827 / 135.2 | 27.55 / 0.6471 / 0.3463 / 88.74 | 26.61 / 0.6540 / 0.4532 / 119.3 | 22.45 / 0.5799 / 0.3909 / 109.2 | 22.01 / 0.3642 / 0.5486 / 133.4 |
| | | LLE | 26.33 / 0.5984 / 0.4018 / 113.7 | 27.99 / 0.6701 / 0.3329 / 86.09 | 27.11 / 0.7246 / 0.3541 / 105.1 | 23.24 / 0.6119 / 0.3683 / 102.9 | 22.66 / 0.4051 / 0.5308 / 136.3 |
| $\sigma_{\mathbf{y}} = 0$ | 3 | - | 28.90 / 0.8396 / 0.2488 / 81.28 | 19.06 / 0.4239 / 0.6122 / 197.1 | 23.59 / 0.7569 / 0.3913 / 149.4 | 17.12 / 0.5146 / 0.5731 / 345.1 | 20.11 / 0.4191 / 0.6069 / 175.1 |
| | | LLE | 29.26 / 0.8465 / 0.2402 / 77.02 | 24.62 / 0.6343 / 0.4303 / 145.8 | 27.57 / 0.8180 / 0.2945 / 129.4 | 17.44 / 0.5183 / 0.5602 / 317.3 | 20.33 / 0.4265 / 0.6041 / 179.4 |
| | 4 | - | 29.49 / 0.8509 / 0.2298 / 72.39 | 22.80 / 0.5685 / 0.4974 / 157.6 | 25.76 / 0.7901 / 0.3295 / 143.2 | 17.91 / 0.5642 / 0.5286 / 282.2 | 22.22 / 0.5097 / 0.5362 / 154.2 |
| | | LLE | 29.69 / 0.8543 / 0.2215 / 69.08 | 27.03 / 0.7907 / 0.3530 / 132.5 | 28.01 / 0.8263 / 0.2613 / 120.8 | 18.70 / 0.5792 / 0.4999 / 239.6 | 22.36 / 0.5033 / 0.5527 / 155.3 |
| | 5 | - | 29.72 / 0.8557 / 0.2233 / 69.89 | 25.65 / 0.6928 / 0.4093 / 137.2 | 27.02 / 0.8101 / 0.2951 / 131.8 | 18.67 / 0.6049 / 0.4879 / 237.6 | 23.59 / 0.5923 / 0.4740 / 139.0 |
| | | LLE | 29.81 / 0.8559 / 0.2160 / 66.27 | 28.30 / 0.8281 / 0.3133 / 122.1 | 28.19 / 0.8287 / 0.2589 / 120.1 | 19.66 / 0.6253 / 0.4550 / 199.8 | 23.78 / 0.5962 / 0.4797 / 137.0 |
| | 7 | - | 30.04 / 0.8611 / 0.2163 / 67.46 | 28.93 / 0.8339 / 0.3053 / 117.9 | 28.38 / 0.8339 / 0.2587 / 121.5 | 20.44 / 0.6807 / 0.4106 / 178.4 | 24.63 / 0.6525 / 0.4224 / 125.7 |
| | | LLE | 30.00 / 0.8601 / 0.2097 / 65.11 | 29.97 / 0.8712 / 0.2609 / 104.5 | 28.78 / 0.8420 / 0.2344 / 108.6 | 21.31 / 0.6997 / 0.3823 / 156.3 | 25.01 / 0.6743 / 0.4215 / 117.3 |
| | 10 | - | 30.34 / 0.8667 / 0.2107 / 66.18 | 30.91 / 0.8866 / 0.2384 / 94.50 | 29.18 / 0.8483 / 0.2341 / 110.8 | 21.88 / 0.7363 / 0.3491 / 141.2 | 25.35 / 0.6994 / 0.3806 / 115.5 |
| | | LLE | 30.34 / 0.8662 / 0.2050 / 64.26 | 31.36 / 0.8989 / 0.2197 / 87.44 | 29.36 / 0.8536 / 0.2167 / 98.45 | 22.94 / 0.7572 / 0.3208 / 121.8 | 25.59 / 0.7193 / 0.3775 / 110.8 |
| | 15 | - | 30.69 / 0.8721 / 0.2062 / 64.29 | 32.63 / 0.9177 / 0.1821 / 70.81 | 29.85 / 0.8606 / 0.2157 / 97.81 | 23.78 / 0.7957 / 0.2823 / 108.0 | 26.29 / 0.7525 / 0.3249 / 104.4 |
| | | LLE | 30.70 / 0.8727 / 0.2015 / 62.40 | 32.81 / 0.9217 / 0.1744 / 67.26 | 29.87 / 0.8629 / 0.2073 / 89.21 | 24.68 / 0.8110 / 0.2628 / 96.05 | 26.16 / 0.7503 / 0.3500 / 102.5 |

Table 36: Results of DAPS on CelebA-HQ Dataset.

| Condition | Steps | Strategy | Deblur (aniso) | Inpainting | 4× SR | CS 50% | Deblur (nonlinear) |
|---|---|---|---|---|---|---|---|
| | | | PSNR↑/SSIM↑/LPIPS↓/FID↓ | | | | |
| $\sigma_{\mathbf{y}} = 0.05$ | 3 | - | 24.49 / 0.4730 / 0.4519 / 57.90 | 18.54 / 0.3297 / 0.6161 / 119.6 | 21.56 / 0.4427 / 0.5080 / 66.27 | 16.70 / 0.3748 / 0.6047 / 266.9 | 22.60 / 0.4216 / 0.5087 / 58.76 |
| | | LLE | 24.92 / 0.5005 / 0.4478 / 58.94 | 21.61 / 0.3745 / 0.5213 / 74.74 | 25.41 / 0.6035 / 0.4113 / 52.38 | 16.81 / 0.3728 / 0.6011 / 248.9 | 22.55 / 0.4229 / 0.5163 / 60.80 |
| | 4 | - | 25.07 / 0.5238 / 0.4049 / 49.97 | 22.44 / 0.4492 / 0.4324 / 73.24 | 24.04 / 0.5153 / 0.4291 / 53.40 | 17.46 / 0.4152 / 0.5561 / 180.4 | 24.00 / 0.4652 / 0.4608 / 52.99 |
| | | LLE | 25.81 / 0.5525 / 0.4002 / 50.32 | 24.58 / 0.5239 / 0.4280 / 56.76 | 25.64 / 0.6475 / 0.3711 / 48.09 | 17.74 / 0.4097 / 0.5505 / 160.4 | 23.98 / 0.4674 / 0.4697 / 54.86 |
| | 5 | - | 25.93 / 0.5612 / 0.3747 / 45.24 | 24.77 / 0.5314 / 0.4225 / 55.97 | 25.10 / 0.5627 / 0.3898 / 47.60 | 18.25 / 0.4490 / 0.5167 / 135.7 | 24.56 / 0.4873 / 0.4324 / 49.44 |
| | | LLE | 26.37 / 0.5913 / 0.3709 / 45.97 | 25.64 / 0.5744 / 0.3942 / 51.58 | 25.70 / 0.6643 / 0.3581 / 47.53 | 18.71 / 0.4472 / 0.5079 / 119.1 | 24.66 / 0.4918 / 0.4347 / 50.73 |
| | 7 | - | 26.65 / 0.6134 / 0.3428 / 39.63 | 26.70 / 0.6083 / 0.3622 / 44.70 | 25.96 / 0.6211 / 0.3470 / 40.74 | 20.55 / 0.5165 / 0.4437 / 82.84 | 25.20 / 0.5167 / 0.3944 / 45.96 |
| | | LLE | 26.99 / 0.6448 / 0.3297 / 39.01 | 26.87 / 0.6182 / 0.3591 / 44.67 | 26.35 / 0.6999 / 0.3277 / 41.53 | 20.83 / 0.5190 / 0.4392 / 77.58 | 25.28 / 0.5211 / 0.3926 / 45.90 |
| | 10 | - | 27.24 / 0.6585 / 0.3124 / 35.02 | 27.61 / 0.6457 / 0.3330 / 39.74 | 26.53 / 0.6640 / 0.3166 / 35.59 | 21.76 / 0.5616 / 0.3988 / 62.31 | 25.71 / 0.5439 / 0.3638 / 42.01 |
| | | LLE | 27.43 / 0.6846 / 0.2999 / 33.93 | 27.76 / 0.6558 / 0.3310 / 40.16 | 26.79 / 0.7220 / 0.3012 / 37.97 | 22.02 / 0.5667 / 0.3951 / 59.68 | 25.71 / 0.5471 / 0.3568 / 41.59 |
| | 15 | - | 27.77 / 0.6999 / 0.2856 / 31.11 | 28.47 / 0.6837 / 0.3053 / 35.66 | 27.03 / 0.7016 / 0.2915 / 31.50 | 23.85 / 0.6207 / 0.3510 / 47.32 | 26.31 / 0.5774 / 0.3348 / 38.94 |
| | | LLE | 27.65 / 0.7191 / 0.2749 / 30.61 | 28.79 / 0.7002 / 0.2999 / 35.68 | 27.33 / 0.7414 / 0.2800 / 33.33 | 24.01 / 0.6365 / 0.3439 / 46.05 | 26.20 / 0.5806 / 0.3236 / 37.59 |
| $\sigma_{\mathbf{y}} = 0$ | 3 | - | 29.39 / 0.7143 / 0.2908 / 33.95 | 18.93 / 0.3765 / 0.5962 / 127.9 | 26.40 / 0.5880 / 0.3675 / 45.63 | 16.98 / 0.4460 / 0.5807 / 278.0 | 25.37 / 0.6509 / 0.3974 / 45.94 |
| | | LLE | 31.69 / 0.8148 / 0.2387 / 31.34 | 22.45 / 0.4363 / 0.4823 / 74.01 | 28.00 / 0.7207 / 0.3419 / 55.98 | 17.11 / 0.4491 / 0.5754 / 269.4 | 25.68 / 0.6765 / 0.3823 / 44.36 |
| | 4 | - | 30.26 / 0.7547 / 0.2626 / 30.33 | 23.42 / 0.5375 / 0.4568 / 76.84 | 27.18 / 0.6355 / 0.3321 / 40.65 | 17.79 / 0.5009 / 0.5284 / 199.6 | 27.43 / 0.7203 / 0.3413 / 37.14 |
| | | LLE | 32.43 / 0.8441 / 0.2148 / 26.91 | 26.31 / 0.6650 / 0.3844 / 49.07 | 28.71 / 0.7544 / 0.3044 / 49.07 | 18.08 / 0.4933 / 0.5224 / 182.0 | 28.13 / 0.7689 / 0.3090 / 33.98 |
| | 5 | - | 30.90 / 0.7827 / 0.2431 / 27.80 | 26.52 / 0.6652 / 0.3633 / 56.46 | 27.72 / 0.6690 / 0.3096 / 37.25 | 18.65 / 0.5482 / 0.4837 / 152.8 | 28.30 / 0.7547 / 0.3067 / 32.81 |
| | | LLE | 32.81 / 0.8591 / 0.2027 / 25.30 | 27.87 / 0.7479 / 0.3160 / 51.54 | 29.12 / 0.7747 / 0.2845 / 44.69 | 19.15 / 0.5464 / 0.4727 / 136.3 | 29.24 / 0.8064 / 0.2727 / 29.51 |
| | 7 | - | 31.80 / 0.8191 / 0.2168 / 23.93 | 29.62 / 0.7971 / 0.2698 / 41.52 | 28.43 / 0.7136 / 0.2791 / 32.58 | 21.23 / 0.6462 / 0.3875 / 92.08 | 29.07 / 0.7888 / 0.2664 / 27.35 |
| | | LLE | 33.55 / 0.8856 / 0.1834 / 23.25 | 29.82 / 0.8176 / 0.2605 / 42.71 | 29.47 / 0.7959 / 0.2693 / 41.98 | 21.56 / 0.6462 / 0.3875 / 86.80 | 30.03 / 0.8316 / 0.2387 / 24.93 |
| | 10 | - | 32.61 / 0.8486 / 0.1939 / 20.98 | 31.20 / 0.8464 / 0.2246 / 33.11 | 29.03 / 0.7508 / 0.2526 / 28.68 | 22.63 / 0.7034 / 0.3361 / 66.86 | 29.54 / 0.8100 / 0.2398 / 24.56 |
| | | LLE | 33.91 / 0.8967 / 0.1727 / 21.46 | 31.42 / 0.8637 / 0.2163 / 35.04 | 29.97 / 0.8186 / 0.2613 / 42.77 | 22.89 / 0.7067 / 0.3333 / 65.45 | 30.55 / 0.8489 / 0.2234 / 22.88 |
| | 15 | - | 33.43 / 0.8745 / 0.1725 / 17.89 | 32.47 / 0.8798 / 0.1883 / 26.74 | 29.55 / 0.7830 / 0.2284 / 24.99 | 25.14 / 0.7722 / 0.2754 / 45.95 | 29.89 / 0.8251 / 0.2180 / 22.15 |
| | | LLE | 34.24 / 0.9070 / 0.1631 / 20.27 | 32.85 / 0.8949 / 0.1781 / 26.79 | 30.41 / 0.8361 / 0.2458 / 39.49 | 25.32 / 0.7810 / 0.2708 / 46.20 | 30.97 / 0.8635 / 0.2090 / 20.72 |

Table 37: Results of DAPS on FFHQ Dataset.

| Condition | Steps | Strategy | PSNR↑/SSIM↑/LPIPS↓/FID↓ | | | | |
| --- | --- | --- | --- | --- | --- | --- | --- |
| | | | Deblur (aniso) | Inpainting | 4× SR | CS 50% | Deblur (nonlinear) |
| $\sigma_\mathbf{y} = 0.05$ | 3 | - | 23.82 / 0.4509 / 0.4872 / **130.6** | 18.11 / 0.3216 / 0.6280 / 193.8 | 21.35 / 0.4240 / 0.5346 / 135.2 | 16.43 / 0.3556 / 0.6324 / **368.5** | 18.97 / 0.2847 / 0.6416 / **168.9** |
| | | LLE | 24.18 / 0.4741 / 0.4858 / 132.1 | 19.98 / 0.3377 / 0.5652 / 160.2 | 24.42 / 0.5647 / 0.4596 / 128.3 | 16.54 / 0.3637 / 0.6309 / 377.4 | 19.10 / 0.2913 / 0.6370 / 173.0 |
| | 4 | - | 24.67 / 0.4989 / 0.4454 / 115.9 | 21.80 / 0.4299 / 0.5197 / 142.0 | 23.37 / 0.4856 / 0.4714 / **123.8** | 16.99 / **0.3922** / 0.5894 / 282.5 | **20.85** / 0.3521 / **0.5584** / **141.8** |
| | | LLE | 25.05 / 0.5243 / 0.4449 / 117.0 | 23.49 / 0.4817 / 0.4676 / 127.3 | 24.74 / 0.6058 / 0.4287 / 124.5 | 17.20 / 0.3870 / 0.5894 / 272.0 | 20.78 / 0.3525 / 0.5629 / 143.7 |
| | 5 | - | 25.22 / 0.5352 / 0.4220 / **110.9** | 23.95 / 0.5043 / 0.4546 / 120.1 | 24.24 / 0.5262 / 0.4421 / **115.7** | 17.59 / **0.4209** / 0.5552 / 228.0 | **21.85** / **0.3871** / **0.5195** / **129.5** |
| | | LLE | 25.59 / 0.5626 / 0.4181 / 112.6 | 24.48 / 0.5311 / 0.4346 / 118.1 | 24.94 / 0.6293 / 0.4113 / 119.0 | 17.92 / 0.4171 / 0.5516 / 212.8 | 21.81 / 0.3862 / 0.5236 / 131.0 |
| | 7 | - | 25.97 / 0.5866 / 0.3904 / 103.5 | 25.71 / 0.5741 / 0.3979 / **102.1** | 25.03 / 0.5806 / 0.4060 / **111.3** | 19.05 / **0.4769** / **0.4918** / 172.3 | **22.80** / **0.4227** / **0.4749** / **120.5** |
| | | LLE | 26.25 / 0.6165 / 0.3825 / 103.4 | 25.75 / 0.5803 / 0.3961 / 105.1 | 25.44 / 0.6648 / 0.3814 / 114.6 | 19.33 / 0.4740 / 0.4900 / 161.7 | 22.76 / 0.4218 / 0.4807 / 121.8 |
| | 10 | - | 26.59 / 0.6330 / 0.3612 / 98.73 | 26.71 / 0.6146 / **0.3649** / **91.47** | 25.59 / 0.6249 / 0.3776 / 106.0 | 20.12 / 0.5216 / 0.4417 / 137.8 | 23.35 / 0.4527 / 0.4472 / **116.2** |
| | | LLE | 26.79 / 0.6617 / 0.3537 / 97.59 | 26.72 / 0.6194 / 0.3664 / 92.47 | 25.92 / 0.6897 / 0.3565 / 105.7 | 20.47 / 0.5224 / 0.4385 / 133.6 | 23.37 / 0.4532 / 0.4461 / 117.8 |
| | 15 | - | **27.22** / 0.6802 / 0.3348 / **90.30** | 27.67 / 0.6557 / 0.3315 / **83.41** | 26.16 / 0.6677 / 0.3513 / 101.2 | 21.77 / 0.5804 / 0.3883 / **108.6** | **23.93** / 0.4884 / 0.4172 / 112.2 |
| | | LLE | 27.11 / **0.7017** / **0.3278** / 95.27 | 27.83 / **0.6684** / **0.3296** / 85.67 | 26.43 / **0.7120** / **0.3357** / **97.74** | **22.00** / **0.5853** / **0.3854** / 110.5 | 23.89 / **0.4914** / **0.4119** / **111.0** |
| $\sigma_\mathbf{y} = 0$ | 3 | - | 28.66 / 0.6918 / 0.3179 / 80.09 | 18.48 / 0.3716 / 0.6071 / 200.4 | 25.54 / 0.5555 / 0.4055 / **109.2** | 16.70 / 0.4218 / 0.6093 / 379.6 | 20.10 / 0.4128 / 0.5904 / 166.6 |
| | | LLE | **30.70** / **0.7863** / **0.2683** / **74.22** | 20.55 / 0.3887 / 0.5328 / 166.7 | **27.11** / **0.6894** / 0.3851 / 123.6 | **16.83** / **0.4423** / **0.6054** / 392.9 | **20.21** / **0.4244** / **0.5810** / **164.7** |
| | 4 | - | 29.56 / 0.7336 / 0.2900 / 73.34 | 22.70 / 0.5194 / 0.4807 / 147.8 | 26.34 / 0.6033 / 0.3720 / **96.68** | 17.29 / **0.4727** / 0.5619 / 297.8 | **22.37** / 0.5376 / 0.4839 / **131.5** |
| | | LLE | **31.61** / **0.8245** / **0.2394** / **66.55** | 24.98 / 0.6101 / 0.4120 / **127.2** | 27.78 / 0.7226 / **0.3480** / 111.8 | 17.52 / 0.4680 / **0.5622** / 292.1 | 22.36 / **0.5502** / 0.4787 / **129.0** |
| | 5 | - | 30.22 / 0.7626 / 0.2700 / 68.05 | 25.48 / 0.6333 / 0.3982 / 119.0 | 26.90 / 0.6380 / 0.3491 / **90.89** | 17.95 / **0.5137** / 0.5232 / 246.7 | 23.72 / 0.6058 / 0.4300 / **113.3** |
| | | LLE | **32.03** / **0.8424** / **0.2261** / **63.79** | 26.37 / **0.6934** / **0.3662** / 116.5 | 28.21 / **0.7453** / **0.3227** / 103.3 | **18.32** / 0.5100 / **0.5194** / 231.8 | 23.72 / **0.6241** / **0.4253** / 115.6 |
| | 7 | - | 31.21 / 0.8025 / 0.2417 / 62.32 | 28.18 / 0.7534 / 0.3146 / 98.21 | 27.71 / 0.6864 / 0.3184 / **83.45** | 19.56 / **0.5952** / 0.4473 / 181.1 | 24.98 / 0.6651 / 0.3759 / **102.5** |
| | | LLE | **32.89** / **0.8753** / **0.1991** / **57.87** | **28.31** / **0.7732** / **0.3085** / 97.76 | **28.61** / **0.7709** / **0.3027** / 96.55 | **19.89** / 0.5931 / **0.4445** / 173.5 | **25.25** / **0.6991** / **0.3700** / 105.2 |
| | 10 | - | 32.14 / 0.8362 / 0.2150 / 54.74 | 29.86 / 0.8130 / 0.2653 / **80.17** | 28.39 / 0.7276 / 0.2891 / **78.51** | 20.75 / 0.6570 / 0.3857 / 144.2 | 25.56 / 0.6986 / 0.3454 / **95.53** |
| | | LLE | **33.35** / **0.8907** / **0.1847** / **54.45** | **29.91** / **0.8292** / **0.2629** / 86.08 | **29.01** / **0.7923** / 0.2823 / 87.14 | **21.19** / **0.6603** / **0.3816** / 139.3 | **25.80** / **0.7309** / **0.3426** / 100.6 |
| | 15 | - | 33.15 / 0.8676 / 0.1876 / **47.85** | 31.36 / 0.8576 / 0.2203 / **66.01** | 29.08 / 0.7677 / **0.2603** / 70.31 | 22.66 / 0.7312 / 0.3172 / **107.8** | 26.04 / 0.7251 / **0.3161** / **92.32** |
| | | LLE | **33.75** / **0.9046** / **0.1715** / 51.07 | **31.51** / **0.8727** / **0.2151** / 67.73 | **29.41** / **0.8090** / 0.2626 / 77.98 | **22.91** / **0.7338** / **0.3163** / 111.4 | **26.27** / **0.7556** / 0.3213 / 96.30 |

# I  More qualitative results

Here we present more visualization results. Anisotropic Deblurring results are shown in Figure 11. Inpainting results are shown in Figure 12. Super-Resolution results are shown in Figure 13. Compressed Sensing results are shown in Figure 14. Nonlinear Deblurring results are shown in Figure 15.

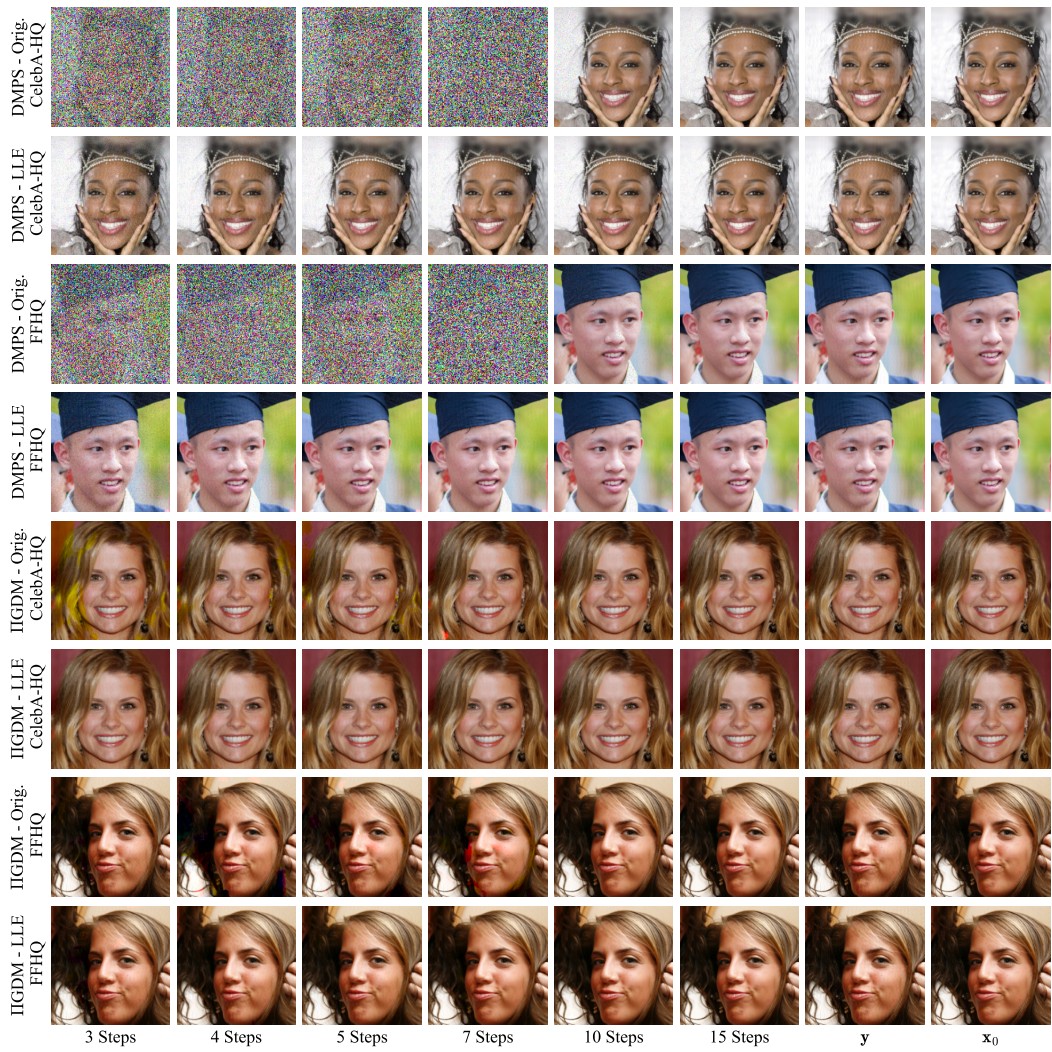

Figure 11: Visualization of anisotropic Deblurring.

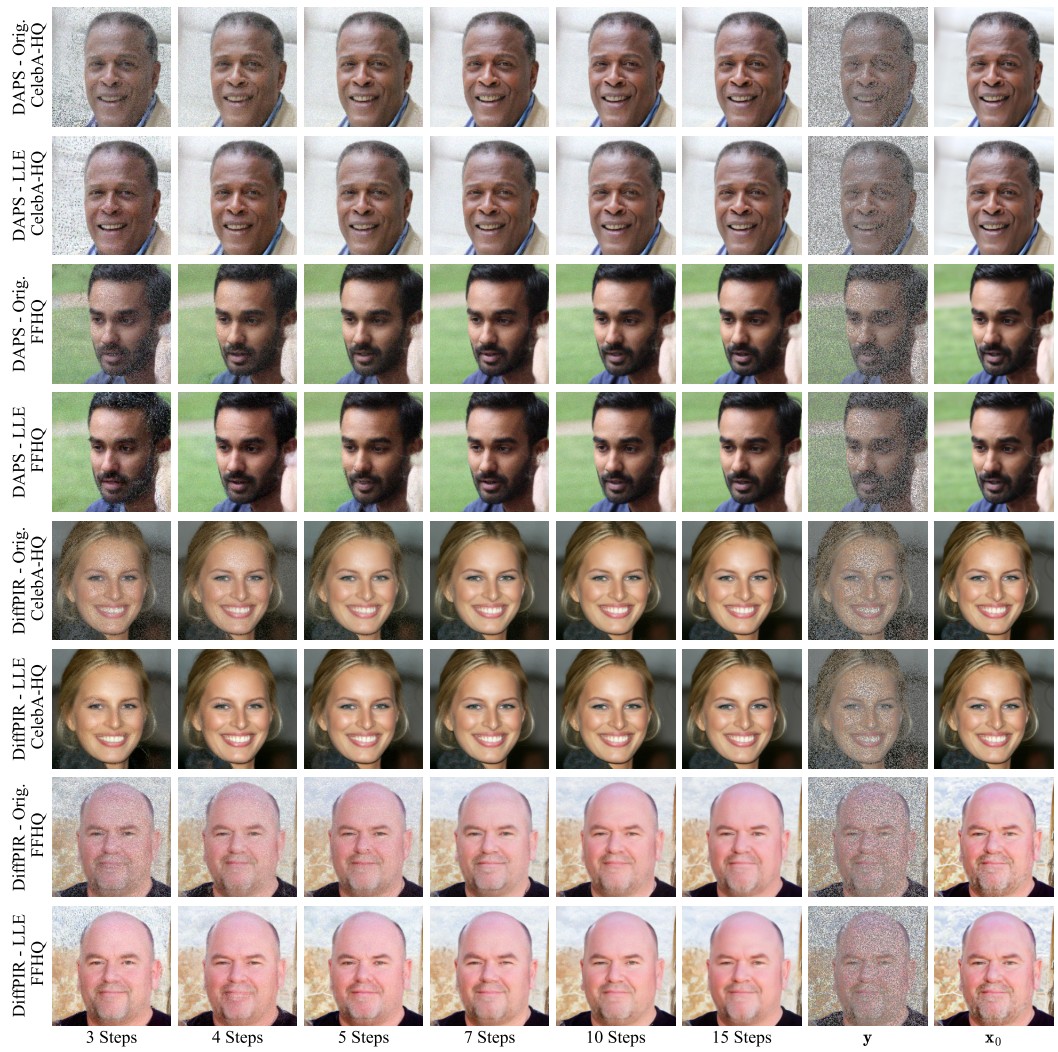

Figure 12: Visualization of Inpainting.

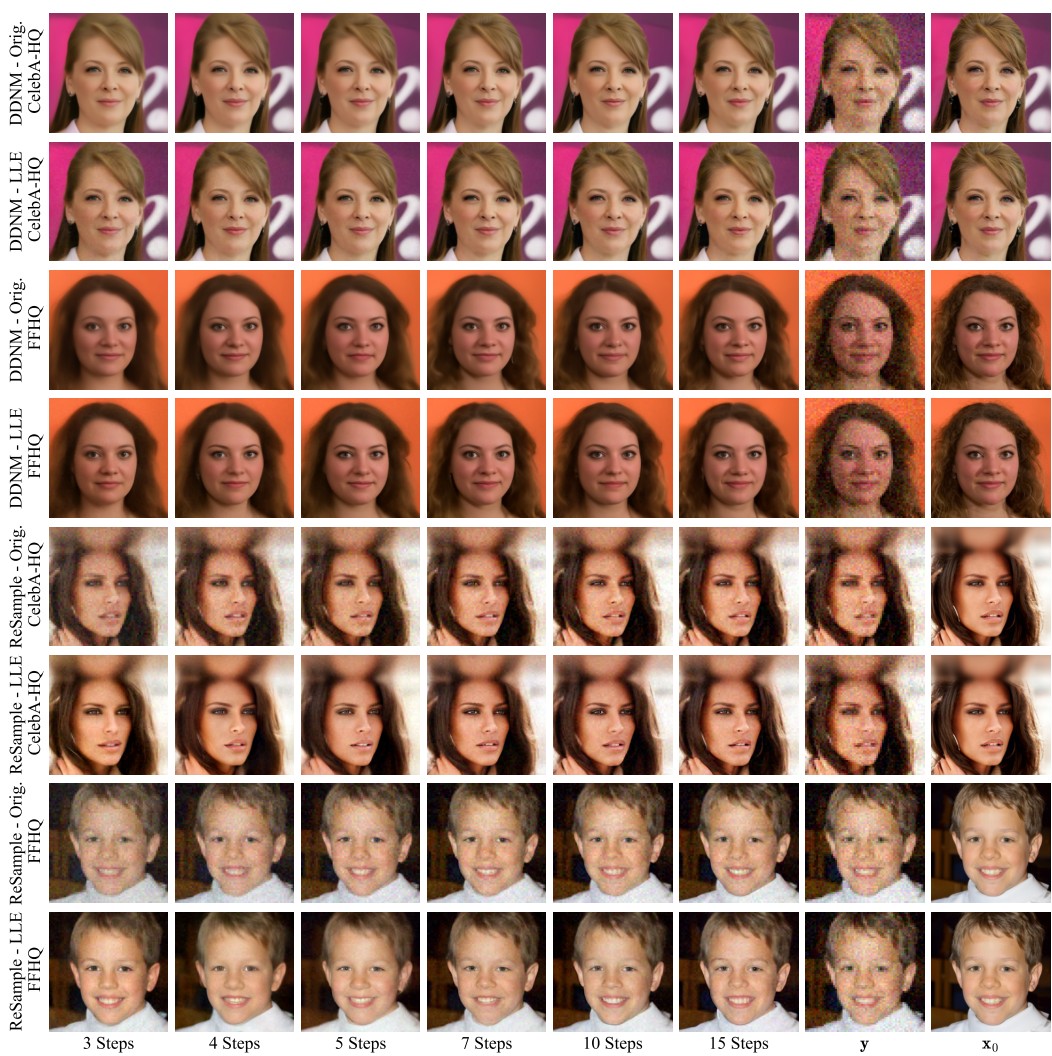

Figure 13: Visualization of Super-Resolution.

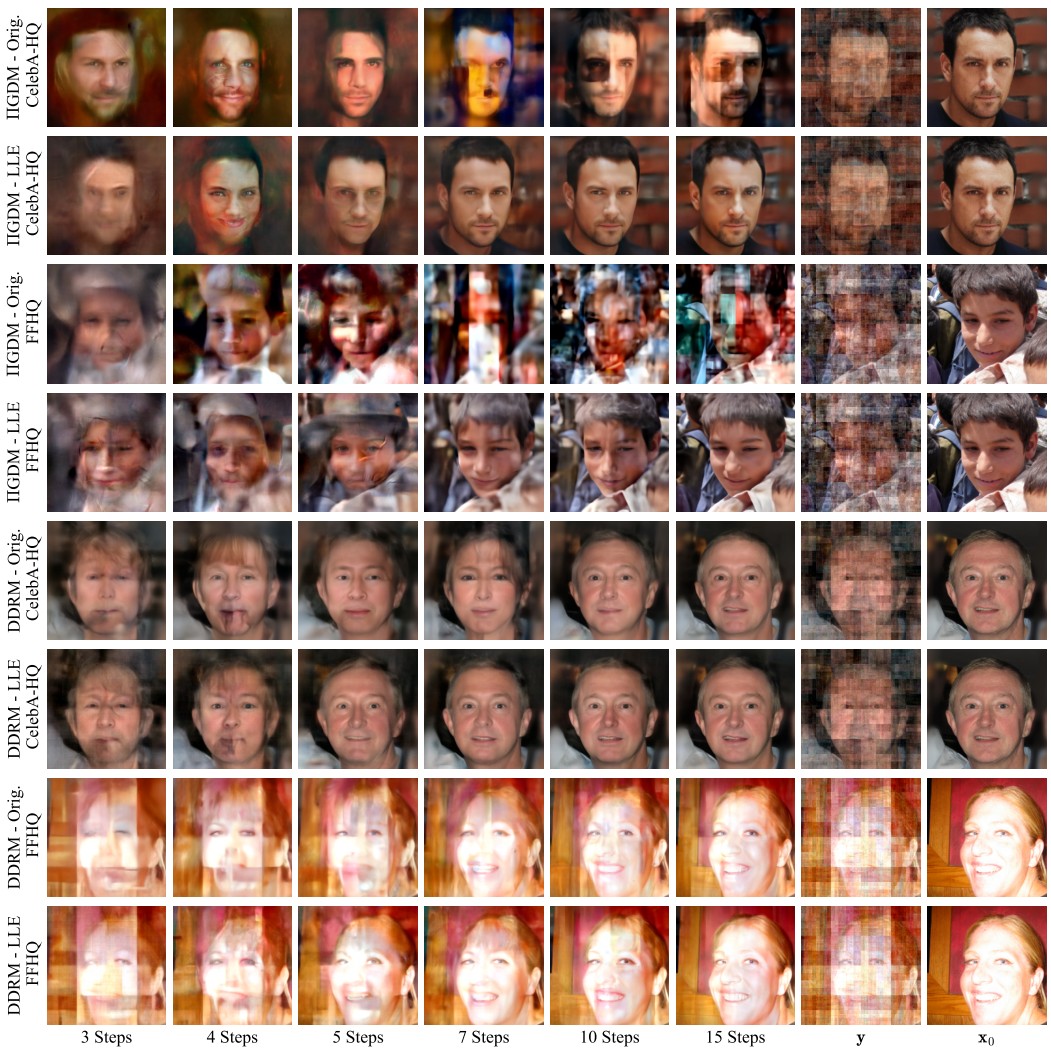

Figure 14: Visualization of Compressed Sensing.

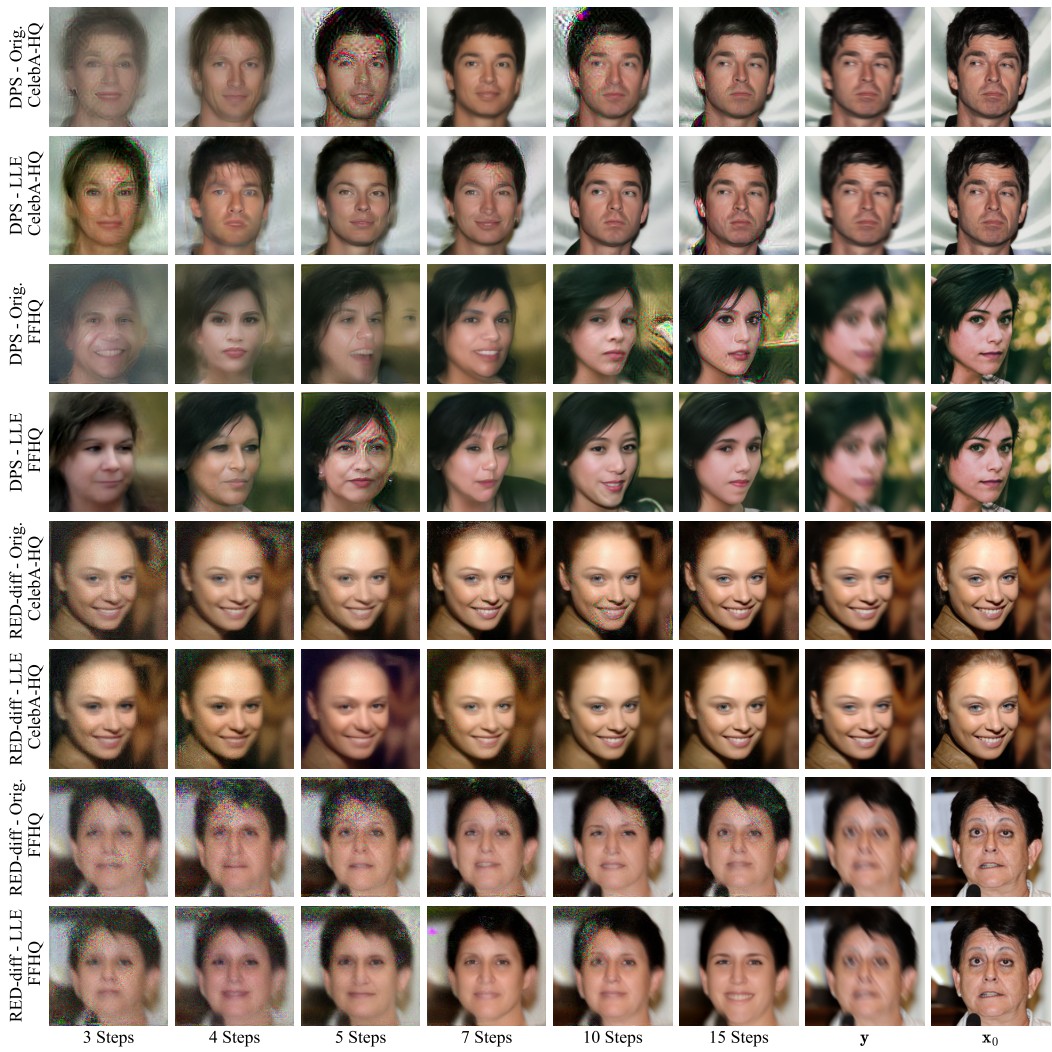

Figure 15: Visualization of nonlinear Deblurring.

