# OpenReview forum: "Improving Diffusion-based Inverse Algorithms under Few-Step Constraint via Linear Extrapolation"
_NeurIPS.cc/2025/Conference — NeurIPS 2025 poster_

### Official Review · Reviewer_7BP1 · 2025-06-25

**Clarity:** 2
**Significance:** 2
**Originality:** 3
**Rating:** 4
**Confidence:** 3

**Summary:**

This paper proposes learnable linear extrapolation (LLE), designed to improve the performance of any diffusion-based inverse algorithm that follows the steps: (i) predict the clean signal, (ii) enforce measurement consistency, and (iii) map back to the noisy manifold to continue the reverse sampling process. The main insight of this paper, as far as I can tell, comes from Corollary 2, where they show that the estimate of the clean sample conditioned on the measurements, denoted $\mathbf{x}_{0}(t, \mathbf{y})$, is a linear combination of the estimate at the current timestep along with all previous timesteps. Motivated by this, they propose solving an optimization problem to learn such linear coefficients in order to obtain an accurate estimate of the clean signal, provided that a reference dataset exists.

**Questions:**

- Why are the results for DMPS in Table 1 so poor? I suggest the authors to look into this to provide a more fair comparison.
- How precisely do Equations (19–20) need to be optimized? Did the authors find that the loss associated with this problem is easy to optimize? In cases where the optimization falls into a local minimum, did the authors find that LLE still outperforms the original method? I believe these questions are fundamental to understanding the effectiveness of LLE, if they have not already been addressed.

**Ethical Concerns:**

["NO or VERY MINOR ethics concerns only"]

**Final Justification:**

The authors have addressed my points and hence I have raised my score. I do still have some uncertainty about the actual practicality of the method, so I am on the fence.

**Limitations:**

I have stated the limitations under weaknesses above. I would be willing to raise my score if my questions can be effectively addressed.

**Paper Formatting Concerns:**

There are no formatting concerns as far as I can tell.

**Quality:**

3

**Strengths And Weaknesses:**

I would say that the main strength of this paper is that they can improve upon existing diffusion-based inverse problem solvers in a lightweight fashion. Table 7 shows that LLE does not induce a large overhead, but can improve upon the existing methods up to some extent.

For weaknesses, I am not sure how effective this method would actually be considering that the improvements are not great. I would suggest the author to put more qualitative results such that we can look at the performance differences a bit better. On top of that, this method assumes that there exists a reference dataset to perform the optimization, which I am not sure holds in practice. How were these reference points chosen for experiments? I perhaps missed this in the main text.

I also think that the presentation of this paper could be improved; I found the presentation of the methodology a bit convoluted. Perhaps there is a way to condense the main text to bring the algorithm box up to the main part.

---

> ### Author Rebuttal · Authors · 2025-07-31
>
> We thank the reviewer for recognizing the lightweight advantage and general applicability of our method. Below, we provide point-by-point responses to the concerns and questions raised. **All additional experimental results are reported in terms of PSNR ↑ / SSIM ↑ / LPIPS ↓.**
>
> **1.  "I am not sure how effective this method would actually be considering that the improvements are not great. I would suggest the author to put more qualitative results."**
>
> We sincerely thank the reviewer for the valuable comments and helpful suggestion. We would like to clarify that in many cases, LLE improves the PSNR of the original algorithm by close to or more than 1dB, which is generally regarded a considerable improvement in the context of image restoration. Moreover, we consistently observe improvements across all three metrics—PSNR, SSIM, and LPIPS—indicating that LLE not only reduces pixel-wise reconstruction error but also enhances perceptual quality.
>
> Regarding the qualitative results, we appreciate the suggestion and will include more visualizations in the camera ready version. In the meantime, we refer the reviewer to Appendix I (pages 35–39) for additional qualitative examples beyond those presented in the main paper. We apologize for the inconvenience that, per this year's policy, visual results cannot be included during the rebuttal phase. Thank you again for this thoughtful feedback.
>
> **2. "This method assumes that there exists a reference dataset to perform the optimization, which I am not sure holds in practice. How were these reference points chosen for experiments?"**
>
> In all of our experiments, we use 999-step DDIM to sample 50 synthetic images from the diffusion model as the reference dataset. This design ensures that LLE remains practical in real-world scenarios: since diffusion-based inverse algorithms assume access to a pre-trained diffusion model, it is straightforward to generate the reference dataset using the same model without requiring any real images.
>
> While this setup is mentioned at several points in the paper (e.g., line 52 on page 2, line 223 on page 7, and line 722 on page 25), we acknowledge that the current presentation may not be sufficiently clear or direct, and could lead to misunderstanding. We will revise the Experimental Setup section to explicitly describe the construction of the reference dataset used by LLE. We thank the reviewer again for this helpful question.
>
> **3. "I found the presentation of the methodology a bit convoluted. Perhaps there is a way to condense the main text to bring the algorithm box up to the main part."**
>
> We appreciate the reviewer's suggestion. We will move the algorithm box into Section 4 of the main text in the camera ready version to improve the overall flow and coherence. We will also refine the presentation of the methodology section to make the description clearer and more concise.
>
> **4. "Why are the results for DMPS in Table 1 so poor? I suggest the authors to look into this to provide a more fair comparison."**
>
> We thank the reviewer for pointing out this issue. We have double-checked our implementation of DMPS and confirmed that it is consistent with the original paper. We also verified our implementation under 100 and 1000 sampling steps, and the results closely match those reported in the original work. In fact, we observe that our implementation achieves comparable performance with the original paper using as few as 15 steps (see Table 18 and Table 19 on page 30), which supports the correctness of our implementation.
>
> Upon further investigation into the hyperparameter settings of DMPS, we find the primary issue lies in the scaling parameter $\lambda$. While the original paper suggests setting $\lambda \geq 1$, we find this choice to be suboptimal in low-step regimes, especially when the number of steps is below 10. To address this, we conduct a step-by-step and task-by-task hyperparameter search for DMPS on the CelebA-HQ dataset with 3, 4, 5, and 7 sampling steps. The table below shows the optimal values of $\lambda$ we identified.
>
> | Steps | Deblur | Inp  | SR   | CS   |
> | ----- | ------ | ---- | ---- | ---- |
> | 3     | 0.2    | 0.2  | 0.3  | 0.2  |
> | 4     | 0.3    | 0.3  | 0.4  | 0.3  |
> | 5     | 0.4    | 0.4  | 0.5  | 0.4  |
> | 7     | 0.5    | 0.6  | 0.6  | 0.5  |
>
> Using these revised hyperparameters, we re-evaluate both the original DMPS and the LLE-enhanced version on noisy tasks on CelebA-HQ. The updated results are shown in the table below. We believe these results provide a more fair and accurate comparison for DMPS. We will incorporate them into the camera ready version, along with revised results on noiseless tasks on CelebA-HQ and on FFHQ. We sincerely thank the reviewer for helping us improve the fairness and credibility of our experimental evaluation.
>
> | Steps | Algo     | Deblur (noisy)            | Inp (noisy)               | SR (noisy)                | CS (noisy)                |
> | ----- | -------- | ------------------------- | ------------------------- | ------------------------- | ------------------------- |
> | 3     | DMPS     | 14.02 / 0.174 / 0.747     | 11.91 / 0.100 / 0.802     | 15.53 / 0.235 / **0.763** | 12.45 / 0.102 / 0.791     |
> |       | DMPS-LLE | **14.10 / 0.177 / 0.744** | **11.97 / 0.104 / 0.800** | **18.76 / 0.253** / 0.776 | **12.52 / 0.105 / 0.789** |
> | 4     | DMPS     | 17.95 / 0.322 / 0.619     | 13.41 / 0.144 / 0.762     | 19.33 / 0.365 / 0.645     | 14.09 / 0.151 / 0.741     |
> |       | DMPS-LLE | **18.23 / 0.325 / 0.617** | **13.55 / 0.149 / 0.759** | **23.07 / 0.511 / 0.544** | **14.19 / 0.155 / 0.738** |
> | 5     | DMPS     | 21.20 / 0.404 / 0.562     | 14.71 / 0.164 / 0.738     | 22.95 / 0.498 / 0.541     | 14.85 / 0.165 / 0.719     |
> |       | DMPS-LLE | **21.63 / 0.406 / 0.558** | **16.45 / 0.186 / 0.690** | **24.66 / 0.625 / 0.437** | **15.57 / 0.180 / 0.699** |
> | 7     | DMPS     | 25.12 / 0.559 / 0.446     | 15.87 / 0.187 / 0.716     | **25.96** / 0.669 / 0.396 | 16.04 / 0.213 / 0.679     |
> |       | DMPS-LLE | **25.84 / 0.576 / 0.436** | **20.12 / 0.337 / 0.562** | 25.82 / **0.707 / 0.355** | **17.36 / 0.303 / 0.594** |
>
> **5. "How precisely do Equations (19–20) need to be optimized? Did the authors find that the loss associated with this problem is easy to optimize? In cases where the optimization falls into a local minimum, did the authors find that LLE still outperforms the original method?"**
>
> We thank the reviewer for the insightful question. We find that the optimization of the objective function (19–20) requires relatively few steps, is easy to optimize, and is not prone to getting stuck in local minima.
>
> To better understand the impact of optimization iterations on LLE's performance, we conduct an additional experiment evaluating LLE trained for {25, 50, 100, 150, 200} iterations. The experiment is performed using 5-step DDNM on the FFHQ noiseless tasks. The results are summarized in the table below. We observe that LLE achieves comparable and strong performance when trained for iterations between 50 and 200. This suggests that LLE converges quickly with relatively few iterations and does not show obvious signs of overfitting.
>
> Theoretically, this is reasonable since LLE learns interpolation coefficients by minimizing a weighted sum of MSE loss and LPIPS loss. When the LPIPS weight $\omega$ is zero, the problem reduces to a least squares problem, which is convex and free of local minima. Although the LPIPS loss is non-convex, our choice of a relatively small weight ($\omega = 0.1$) ensures that the overall loss landscape does not contain many undesirable local minima, making optimization easier in practice. We sincerely thank the reviewer for this insightful question and will include these additional experiments and discussions in the final version of the paper.
>
> | Train steps | Deblur (noiseless)                | Inp (noiseless)                   | SR (noiseless)               | CS (noiseless)                    |
> | ----------- | --------------------------------- | --------------------------------- | ---------------------------- | --------------------------------- |
> | -           | 39.82 / 0.964 / 0.087             | 26.34 / 0.719 / 0.380             | **29.99** / **0.865** / 0.211 | 18.29 / 0.591 / 0.487             |
> | 25          | 39.93 / 0.965 / 0.086             | 28.21 / 0.821 / 0.316             | 29.93 / 0.863 / 0.207        | 18.87 / 0.597 / 0.476             |
> | 50          | 40.12 / **0.967** / 0.085         | 28.61 / 0.834 / 0.303             | 29.95 / 0.863 / 0.208        | 19.13 / 0.602 / 0.468             |
> | **100**     | 40.08 / 0.966 / **0.084**             | 28.71 / 0.838 / 0.293             | 29.94 / 0.863 / 0.206        | 19.21 / **0.605** / 0.463             |
> | 150         | 40.10 / 0.966 / **0.084**             | 28.77 / 0.840 / 0.290             | 29.91 / 0.862 / **0.204**    | 19.23 / 0.603 / **0.461**             |
> | 200         | **40.14** / **0.967** / **0.084** | **28.82** / **0.842** / **0.288** | 29.93 / 0.863 / **0.204**    | **19.25** / 0.604 / 0.462 |
>
> We would like to express our gratitude once again for the reviewer's comments and suggestions for our work. Should there be any further questions, we welcome continued discussion.

---

> ### Author Response · Authors · 2025-08-05
> **Kind Reminder to Review Our Rebuttal**
>
> Dear Reviewer 7BP1,
>
> Thank you once again for your helpful and constructive review.
>
> As the Discussion phase is approaching its end in less than three days, we would like to kindly remind you to take a look at our rebuttal. We have added substantial experiments and clarifications in response to your valuable comments and concerns.
>
> If there are any points we may have not fully addressed, we would be grateful if you could let us know.
>
> Looking forward to your feedback.
>
> Best regards,
>
> The Authors.

---

### Official Review · Reviewer_bt2H · 2025-06-28

**Clarity:** 3
**Significance:** 2
**Originality:** 2
**Rating:** 4
**Confidence:** 3

**Summary:**

This paper tackles the high computational cost of diffusion-based inverse problem solvers, particularly in few-step scenarios. First, it proposes a "canonical form" that deconstructs diffusion-based inverse algorithms into three distinct modules: a Sampler, a Corrector, and a Noiser. This framework is designed to unify a wide range of existing methods. Second, based on this canonical form and an analysis of ODE solvers, the paper introduces Learnable Linear Extrapolation (LLE), a lightweight method that improves few-step performance by learning to linearly combine the current estimate with previous ones. The authors demonstrate that LLE consistently improves results across nine reformulated diffusion algorithms on several inverse problem tasks, such as deblurring and inpainting on facial datasets.

**Questions:**

1. How is LLE expected to perform against the baselines mentioned in the weakness section?

2. What are the specific technical challenges in applying the canonical form and LLE to Latent Diffusion Models? Does linear extrapolation hold in the latent space, and how does the VAE's compression affect the Corrector module?

3. How can the generalization of results from facial datasets to more complex domains like ImageNet be justified? The distinct priors of faces versus general scenes could affect the validity of the learned extrapolations.

4. Please elaborate on the conceptual differences and key trade-offs (e.g., performance vs. training cost) between the LLE method and consistency-based approaches like Consistency Posterior Sampling.

**Ethical Concerns:**

["Major Concern: Data quality and representativeness"]

**Final Justification:**

I thank the authors for their thorough rebuttal. In light of their response, I have raised my score to Borderline Accept.

**Limitations:**

A more thorough discussion of the challenges involved with LDMs and a justification for not addressing them is warranted.

**Quality:**

2

**Strengths And Weaknesses:**

#### **Strengths**

1.  LLE method is a lightweight and intuitive approach, well-motivated by an analysis of high-order ODE solvers.
2.  The proposed canonical form is an insightful contribution that provides a structured, unified view of diverse diffusion-based inverse algorithms.

#### **Weaknesses**

1. The claim of enhancing performance lacks validation against the true state-of-the-art. The baseline set, while broad, omits recent, competitive methods like  LGD (Song et al., 2023), DCDP (Li et al., 2024), MPGD (He et al., 2024), FPS-SMC (Dou and Song, 2024), and  SGS-EDM (Wu et al., 2024).
2. The over-reliance on facial datasets (CelebA-HQ, FFHQ) is not representative of general-purpose restoration; validation on a diverse benchmark like ImageNet is needed to prove general applicability.
3. The paper's complete focus on pixel-space models is a critical weakness. As LDMs are the de facto standard for high-resolution restoration, a method that does not demonstrate a clear path for applicability to them has severely limited practical significance.
4. The paper fails to discuss or compare its approach with consistency models (e.g., Consistency Posterior Sampling). This is a major concurrent research direction for accelerating inverse problem solvers, and its omission makes the literature review and experimental comparison incomplete.

---

> ### Author Rebuttal · Authors · 2025-07-31
>
> We sincerely thank the reviewer for acknowledging the motivation and general applicability of our work. Below, we provide detailed responses to the concerns and questions raised. For clarity, we organize our response as follows:
>
>  (1) We first present additional experiments on ImageNet to further validate the generality of LLE in more challenging settings.
>
>  (2) We then report the performance of additional baselines on the FFHQ dataset as suggested by the reviewer, including five pixel-space methods, three latent-space methods, and one method based on consistency model.
>
> **All results are reported using PSNR ↑ / SSIM ↑ / LPIPS ↓ for consistency.**
>
> **1. "Validation on a diverse benchmark like ImageNet is needed to prove general applicability."**
>
> We supplement the results of $\Pi$GDM, RED-diff, DPS, and ReSample on ImageNet with 4 steps and 10 steps in the following table. Following [12, 14] in the main paper, we sample 100 images from the ImageNet test set for evaluation. For the noisy tasks, we add Gaussian noise with $\sigma = 0.1$ with respect to data range [-1, 1] (same in the following experiments). We use the ADM pre-trained checkpoint ([4] in the main paper).
>
> The results demonstrate that LLE consistently improves performance across all methods in the few-step setting on ImageNet. This confirms that LLE is effective and broadly applicable on more complex datasets.
>
> In addition, we conduct a cross-domain experiment between FFHQ and ImageNet. Due to space limitations, we kindly refer the reviewer to our response to *Reviewer enUs (Point 4)* for detailed results. These results further supports LLE's out-of-domain robustness.
>
> | Steps|Algo|  Deblur (noiseless)|  Inp (noisy) |SR (noiseless)| CS (noisy) |
> |:-|:-| :--:|:--:|:--:|:--: |
> | 4|$\Pi$GDM |23.56/0.727/0.294|18.75/0.408/0.641|17.41/0.373/0.557|14.60/0.217/0.746|
> ||$\Pi$GDM-LLE|**30.14/0.875/0.196**|**19.41/0.435/0.619**|**20.53/0.524/0.531**|**16.73/0.359/0.650** |
> ||RED-diff |22.57/0.589/0.490|18.00/0.357/0.592|24.81/0.719/0.419|17.10/0.409/0.574|
> ||RED-diff-LLE|**22.7/0.602/0.486** |**21.69/0.524/0.492**|**25.19/0.726/0.359**|**17.77/0.468/0.531** |
> ||DPS |15.90/0.268/0.681|22.92/0.672/0.421|17.10/0.358/0.567|15.30/0.300/0.633|
> ||DPS-LLE |**18.42/0.388/0.622**|**24.41/0.690/0.410**|**20.33/0.517/0.533**|**18.03/0.474/0.556** |
> ||ReSample |25.89/0.746/0.355|20.49/0.440/0.521|23.18/0.651/0.417|17.50/0.439/0.540|
> ||ReSample-LLE|**26.18/0.757/0.340**|**23.28/0.561/0.443**|**24.98/0.712/0.369**|**17.80/0.454/0.529** |
> | 10|$\Pi$GDM |26.22/0.789/0.289|21.99/0.609/0.487|21.49/0.532/0.503|16.21/0.360/0.681|
> ||$\Pi$GDM-LLE|**29.92/0.856/0.216**|**23.69/0.611/0.471**|**22.76/0.596/0.450**|**18.61/0.509/0.540** |
> ||RED-diff|23.41/0.620/**0.458** |22.95/0.539/0.449|25.75/0.749/0.349|18.49/0.466/0.517|
> ||RED-diff-LLE|**23.42/0.623**/0.462|**25.07/0.669/0.386**|**25.92/0.754/0.309**|**19.22/0.513/0.487** |
> ||DPS |19.55/0.426/0.608|24.65/0.711/0.364|21.28/0.520/0.512|16.61/0.413/0.612|
> ||DPS-LLE |**20.33/0.463/0.555**|**27.17/0.780/0.317**|**22.42/0.585/0.460**|**19.52/0.590/0.463** |
> ||ReSample |26.49/0.765/0.331|25.26/0.625/0.367|25.27/0.721/0.353|19.60/0.519/0.449|
> ||ReSample-LLE|**26.72/0.772/0.317**|**25.78/0.651/0.349**|**25.61/0.736/0.329**|**20.02/0.541/0.434** |
>
> **2. Including more baseline methods like LGD, DCDP, MPGD, FPS-SMC, and SGS-EDM.**
>
> We supplement experiments with LGD, DCDP, MPGD, FPS-SMC, and SGS-EDM on noisy tasks on FFHQ with 4 steps. The results are reported in the **Pixel space** part of the table below. The results show that LLE consistently improves the performance of all five baselines in the few-step regime. This further validates the generality and broad applicability of LLE.
>
> **3. "What are the specific technical challenges in applying the canonical form and LLE to Latent Diffusion Models?"**
>
> There are no difficulties in extending the canonical form and LLE from pixel space to latent space. The core theoretical analysis of linear extrapolation still holds. The observation function with LDMs is $\mathbf{y} = \mathcal{A}(\mathcal{D}(\mathbf{z}_0)) + \sigma \mathbf{n}$, where $\mathcal{D}(\cdot)$ denotes the decoder of the VAE, $\mathcal{A}(\cdot)$ is the observation operator, and $\mathbf{z}_0$ is the latent. This formulation corresponds to a nonlinear inverse problem with respect to $\mathbf{z}_0$, since the composed operator $\mathcal{A} \circ \mathcal{D}$ is highly nonlinear. Our theoretical motivation remains valid, as it does not rely on any specific assumptions about the form of the observation function.
>
> To solve inverse problems with LDMs, the Corrector (or the inverse algorithm) must be capable of handling nonlinear observation functions. Moreover, as long as the inverse algorithm is applicable with LDM, LLE can still serve as an effective enhancement strategy.
>
> To validate LLE with LDMs, we include results of Latent-DPS (introduced in [12] of the main paper), PSLD [R.1], and STSL [R.2] on the FFHQ dataset. The results are reported in the **Latent space** part of the table below. LLE consistently improves performance for these LDM baselines. The only adjustment required for training LLE in the latent space is a minor change to the loss in Equation (20), which we modify to $\mathcal{L}(\mathcal{D}(\tilde{\mathbf{z}}_{0, t_i}^{(n)}), \mathcal{D}(\mathbf{z}_0^{(n)}))$, since LPIPS is computed in the pixel space. These results further confirm the generality and effectiveness of LLE when applied to LDMs.
>
> **4. The conceptual differences and key trade-offs between the LLE method and consistency-based approaches like Consistency Posterior Sampling.**
>
> We would like to clarify that LLE and Consistency Posterior Sampling (referred to CPS below) [R.3] aim to improve performance from different but complementary perspectives. Specifically, LLE enhances the efficiency by learning a better solver over the original trajectory, while CPS leverages the strong single-/few-step performance of consistency models to improve reconstruction quality.
>
> Moreover, we note that CPS also fits into our proposed canonical form, and LLE can be applied to further enhance CPS's performance in the few-step regime. More specifically, CPS uses a consistency model as the Sampler, rather than a traditional diffusion model, but it still generates a sequence of $\mathbf{x}_0$ estimates, forming a valid trajectory. This trajectory can be directly used with LLE's linear extrapolation framework.
>
> We supplement experiments with CPS on the ImageNet-64 dataset. The results are shown in the **CM** part of the table below. We use the `openai/diffusers-cd_imagenet64_l2` checkpoint available on Hugging Face, and test on 100 randomly sampled images from the ImageNet test set.
>
> LLE consistently improves CPS in the few-step setting, demonstrating that LLE and CPS are not exclusive, but rather complementary approaches. This highlights LLE's broad applicability and its ability to boost performance when applied to advanced sampling frameworks with consistency models.
>
> | Steps|Type|Algo|Deblur (noisy) |Inp (noisy)|SR (noisy)|CS (noisy)
> |:-| - |:-|:-| :-- |:-|:-|
> | 4|**Pixel space** |DCDP|26.00/0.619/0.428| 24.75/0.558/0.436|24.16/0.585/0.578| 17.70/0.461/0.535|
> ||| DCDP-LLE|**26.12/0.681/0.378**|**25.54/0.596/0.411**|**25.24/0.707/0.350**|**18.16/0.471/0.524** |
> ||| LGD |22.81/0.645/0.433| 21.77/0.548/0.513|24.95/0.680/0.439| 15.84/0.451/0.513|
> ||| LGD-LLE |**24.59/0.695/0.376**|**28.25/0.768/0.339**|**25.55/0.705/0.402**|**16.52/0.503/0.499** |
> ||| MPGD|25.29/**0.705**/0.380|27.33/0.753/0.309|24.93/0.562/0.579| 17.36/0.437/0.565|
> ||| MPGD-LLE|**25.43**/0.704/**0.366**|**27.76/0.768/0.298**|**25.42/0.703/0.360**|**18.02/0.446/0.543** |
> ||| SGS-EDM |**27.72**/0.776/0.308|19.71/0.416/0.538|26.53/0.738/0.337| 17.72/0.482/0.530|
> ||| SGS-EDM-LLE|27.67/**0.782/0.307**|**24.44/0.511/0.453**|**26.95/0.762/0.309**|**18.40/0.516/0.486** |
> ||| FPS-SMC |25.11/0.517/0.475| 17.52/0.188/0.686|24.62/0.524/0.474| 16.58/0.224/0.666|
> ||| FPS-SMC-LLE|**26.26/0.617/0.428**|**18.63/0.229/0.643**|**26.14/0.672/0.400**|**17.66/0.384/0.569** |
> | 4|**Latent space**|Latent-DPS|16.42/0.428/0.636| 15.43/0.426/**0.628**|15.30/0.369/0.682| 14.51/0.364/0.665|
> ||| Latent-DPS-LLE|**17.08/0.441/0.609**|**17.83/0.492**/0.652|**15.49/0.399/0.648**|**14.70/0.366/0.646** |
> ||| PSLD|17.55/0.468/0.630| 17.71/0.474/0.627|17.72/0.471/0.630| 15.06/**0.438**/0.647 |
> ||| PSLD-LLE|**18.10/0.477/0.607**|**18.55/0.490/0.599**|**18.57/0.489/0.597**|**15.24**/0.433/**0.644** |
> ||| STSL|16.08/0.378/0.655| 16.87/0.404/0.639|15.70/**0.368**/0.663|15.31/0.383/0.629|
> ||| STSL-LLE|**16.97/0.392/0.616**|**17.50/0.407/0.598** |**16.27**/0.360/**0.628**|**15.72/0.384/0.621** |
> | 4|**CM**|CPS |12.69/0.204/0.613| 12.71/0.269/0.649|14.65/0.317/0.559| 14.80/0.242/0.619|
> ||| CPS-LLE |**16.50/0.283/0.522**|**17.64/0.383/0.566**|**18.57/0.410/0.471**|**16.58/0.268/0.580** |
>
> **5. Ethical Concern**
>
> It was unexpected for us to receive an *Ethical Concern* flag, as all datasets and algorithms used in our work are publicly available and widely adopted. We'd appreciate it if the reviewer could elaborate on the specific ethical concerns in detail.
>
> We would like to express our gratitude once again for the reviewer's comments and suggestions for our work. All additional experiments and comparisons, including those on ImageNet and nine baseline methods on FFHQ, will be included in the camera ready version to further support the generality and validity of LLE. Should there be any further questions, we welcome continued discussion.
>
> [R.1] Rout, Litu, et al. "Solving linear inverse problems provably via posterior sampling with latent diffusion models." *NeurIPS*, 2023.
>
> [R.2] Rout, Litu, et al. "Beyond first-order tweedie: Solving inverse problems using latent diffusion." *CVPR*, 2024.
>
> [R.3] Xu, Tongda, et al. "Consistency model is an effective posterior sample approximation for diffusion inverse solvers." *ICLR*, 2025.

---

> > ### Author Response · Authors · 2025-08-01
> > **Thank you for pointing out the ethical concerns. We will make the necessary revisions.**
> >
> > We appreciate your highlighting the imbalances in ethnicity, age, and gender within the datasets we used, as well as the potential social impacts of our method. We sincerely apologize for not addressing these ethical concerns in our previous rebuttal. After carefully reviewing your comments and those from the three Ethical Reviewers, we will acknowledge the potential issues associated with the datasets in both the **Limitations** section and the checklist. The modifications are as follows:
> >
> > **In the Limitations section, We will add the following statement:**
> >
> > - We note that the CelebA-HQ and FFHQ datasets used in this paper have demographic imbalances, which may cause our results to fail to generalize to underrepresented groups. A more diverse and balanced dataset is needed in the field to ensure fairness across factors such as ethnicity, gender, and age. A more representative diffusion model trained on such data is also preferred to better support real-world applications.
> >
> > **In the checklist, we will revise our response to Question (10) ("Broader impacts") to "Yes" and add the following statement:**
> >
> > - Facial datasets such as CelebA-HQ and FFHQ are used in our experiments. While our method is general-purpose, we acknowledge that related technologies may potentially be misused, for example in DeepFake scenarios. Improved techniques such as digital watermarking and DeepFake detection may need to be further developed to mitigate the potential societal risks posed by such misuse.
> >
> > Thank you again for highlighting the ethical issues with the datasets we used. We believe that these new additions can address the ethical concerns in our paper.

---

> > ### Comment · Reviewer_bt2H · 2025-08-05
> >
> > I thank the authors for their thorough rebuttal. Most of my concerns are addressed.

---

> > > ### Author Response · Authors · 2025-08-05
> > > **Thanks for your positive feedback on our rebuttal!**
> > >
> > > Thank you again for reviewing our work and for your positive feedback on our rebuttal. We will incorporate all the additional experiments and discussions into the camera-ready version of the paper. If you have any remaining concerns, please don’t hesitate to let us know and we would be happy to address them.

---

> ### Author Response · Authors · 2025-08-05
> **Kind Reminder to Review Our Rebuttal**
>
> Dear Reviewer bt2H,
>
> Thank you once again for your helpful and constructive review.
>
> As the Discussion phase is approaching its end in less than three days, we would like to kindly remind you to take a look at our rebuttal. We have added substantial experiments and clarifications in response to your valuable comments and concerns.
>
> If there are any points we may have not fully addressed, we would be grateful if you could let us know.
>
> Looking forward to your feedback.
>
> Best regards,
>
> The Authors.

---

### Official Review · Reviewer_enUs · 2025-06-30

**Clarity:** 2
**Significance:** 3
**Originality:** 3
**Rating:** 3
**Confidence:** 5

**Summary:**

This paper introduces "Learnable Linear Extrapolation", a method to improve the performance of diffusion-based inverse algorithms with few sampling steps by leveraging previous estimation.
The authors propose a canonical form that unifies several diffusion-based inverse algorithms by decomposing them into three modules: Sampler, Corrector, and Noiser.
The method  operates by learning optimal linear combination coefficients to refine current predictions using previous estimates predictions.
The coefficient are pre-learned my minimizing a ``reconstruction + w * perceptual`` loss with respect to a reference dataset.
The authors validate the method through extensive experiments across multiple algorithms and tasks.

**Questions:**

None

**Ethical Concerns:**

["NO or VERY MINOR ethics concerns only"]

**Final Justification:**

To sum up,

- the proposed idea is original
- my concerns were addressed during the rebuttal
- the authors committed to fixing the highlighted inconsistencies and typos in the manuscript

On the other hand, the work has few limitations

- the method is bound to a specific algorithm (requires re-training for each algorithm)
- minor improvements in out-of-distribution setup (e.g. using the method trained on Imagenet for FFHQ) as shown in the authors response
- low dependencies between iterates (the matrix of weights) that may compromise the effectiveness of the extrapolation

As conclusion, I retain my rating

**Limitations:**

- Memory Footprint: The method necessitates storing previous estimate, which could lead to considerable memory overhead
- Out-of-Distribution Performance: the method might suffer from out-of-distribution samples as the extrapolation coefficient are computed beforehand on a selected dataset.
- Algorithm Dependence: the method is dependent on  the specific diffusion algorithm employed. changing the diffusion sampler requires re-training to get the extrapolation coeffcients.

**Quality:**

2

**Strengths And Weaknesses:**

## Strengths:

- A rich framework that encompasses various diffusion posterior samplers
- A novel method that improves performance with reduced required number of sampling steps


## Weaknesses:

**Theoretical Justification:**
- Ambiguity in the justification for $\hat{x}_0$ being a linear combination of previous estimate, particularly in Equation (10), it is unclear why it holds and a reference would great.
In fact, empirical observations of the extrapolation coefficient in Figures 6-8 suggests minimal dependence of the current iterate on previous estimates, in fact the matrix of weights is dominated by the diagonal or first diagonal elements, while the rest are almost zero.
- The choice and size of the ground-truth datasets are not adequately addressed. Also the robustness of the method to out-of-distrbution samples.
While the authors assess the the robustness on datasets FFHQ and CelebA-HQ, it is of little relevance as both dataset represent human faces. A more relevant discussion would be on Imagenet dataset, for instance.

**Practical Considerations:**
- **Computational Overhead:** Essential computational aspects, including the training time required to obtain the extrapolation coefficients and the memory requirements of the method during inference (as it requires storing previous estimates), are not discussed.

**Inconsistencies:**
- **Error in equation (2):** There is a missing logarithm in Equation (2)
- **Figure 4 Clarity:** Figure 4 is unclear and lacks essential information. Specifically, the values of $ w $ are missing, and while the axes are labeled (LPIPS and PSNR), the context or meaning of $ w$ in relation to these metrics is not provided.

---

> ### Author Rebuttal · Authors · 2025-07-31
>
> We sincerely thank the reviewer for recognizing the novelty and generality of our proposed method. Below, we provide point-by-point responses to the raised concerns and questions. **All additional experimental results are reported using the metrics PSNR ↑ / SSIM ↑ / LPIPS ↓.**
>
> **1. "Ambiguity in the justification for being a linear combination of previous estimate, particularly in Equation (10), it is unclear why it holds and a reference would great."**
>
> Equation (10) employs a forward difference approximation of high-order derivatives. The related derivation can be found in [R.1] and in book [R.2] (Chapter 6.5). We will add these citations in the revised version to make the presentation clearer. We sincerely thank the reviewer for the helpful suggestion.
>
> **2. "The matrix of weights in Figure 6-8 is dominated by the diagonal or first diagonal elements, while the rest are almost zero."**
>
> Yes, and this is consistent with our expectations. According to Equation (10), elements farther from the diagonal correspond to higher-order derivatives of the sampling trajectory. Since most sampling trajectories are expected to be smooth, the high-order derivatives decay rapidly to zero, which results in the weight matrix being dominated by the diagonal and first off-diagonal elements. In a few cases, we observed non-zero elements farther from the diagonal, indicating possible non-smooth regions in the trajectory with larger high-order derivatives. We will include this discussion in Appendix C.1. We thank the reviewer for this insightful comment.
>
> **3. "The choice and size of the ground-truth datasets are not adequately addressed."**
>
> **Regarding the train set for LLE**, we used 999-step DDIM to sample 50 synthetic images from the diffusion model as the reference set, as stated in the paper (line 52 on page 2, line 223 on page 7, and line 722 on page 25). Using only synthetic images makes LLE more practical in real world applications, since diffusion-based inverse algorithms assume access to a pre-trained diffusion model, thus making it straightforward to construct the reference set required by LLE.
>
> **Regarding the test set in our experiments**, we follow common practice (e.g., references [5, 6, 12, 14] in the main paper) to sample test examples, as described in the experiment section. Specifically, we use 1000 test samples for CelebA-HQ and 100 samples for FFHQ (and ImageNet in the additional experiments).
>
> We will further add a paragraph at the beginning of the Experiment section to clarify these settings. We sincerely thank the reviewer for helping us improve the clarity of our paper.
>
> **4. A more relevant discussion of cross-domain performance on Imagenet**
>
> Here we supplement cross-domain experiments of LLE between the FFHQ and ImageNet datasets. We test DDNM with 5 steps, and the results are shown in the table below ("–" denotes results without using LLE). LLE demonstrates generalization ability across FFHQ and ImageNet. The coefficients trained on one dataset can still improve performance on the other in most cases, and sometimes even outperform those trained in-domain. This further indicates the out-of-domain robustness of LLE, which is critical for ensuring its general applicability.
>
> Moreover, we have conducted broader evaluations of LLE with multiple inverse algorithms on the ImageNet test set; due to space limitations, we kindly refer the reviewer to our response to *Reviewer bt2H (Point 1)* for details.
>
> We will include these new experimental results in the camera ready version. We sincerely thank the reviewer for the valuable suggestion.
>
> | Testset     | Trainset    | Deblur (noiseless)                | Inp (noiseless)                   | SR (noiseless)                | CS (noiseless)                    |
> | -------- | -------- | --------------------------------- | --------------------------------- | ----------------------------- | --------------------------------- |
> | FFHQ     | -        | 39.82 / 0.964 / 0.087             | 26.34 / 0.719 / 0.380             | **29.99** / **0.865** / 0.210 | 18.29 / 0.591 / 0.488                 |
> |  FFHQ        | FFHQ     | **40.08** / **0.966** / **0.084** | **28.71** / **0.838** / **0.292** | 29.94 / 0.863 / **0.206**     | **19.21** / 0.605 / 0.463             |
> |   FFHQ       | ImageNet | 39.94 / 0.965 / 0.086             | 28.30 / 0.818 / 0.317             | 29.94 / 0.863 / 0.208         | 19.01 / **0.608** / 0.470         |
> | ImageNet | -        | 36.48 / **0.949** / 0.099         | 23.35 / 0.628 / 0.416             | 25.74 / **0.754** / 0.302     | 17.86 / 0.535 / 0.486             |
> |   ImageNet       | ImageNet | 36.52 / **0.949** / **0.098**     | 24.87 / 0.726 / 0.364             | **25.76**  / 0.752 / 0.297    | **18.35** / **0.541** / **0.478** |
> | ImageNet         | FFHQ     | **36.53** / **0.949** / 0.099     | **25.14** / **0.759** / **0.341** | 25.70 / 0.750 / **0.296**     | 18.20 / 0.527 / 0.485             |
>
> **5. Training time required to obtain the extrapolation coefficients and the memory requirements during inference.**
>
> **Regarding the training time**, we reported the training time of LLE with different inverse algorithms in Table 6 of Appendix C.4 (page 19). Training LLE takes only 2 to 20 minutes, thanks to its lightweight design and the use of a small reference set.
>
> **Regarding the memory footprint**, here we further report the GPU memory usage (in MB) of DDNM and DPS with and without LLE in the table below. We measure the CUDA peak allocated memory, and we account only for the memory used by model parameters, data, and additional storage required by the algorithm. The results show that LLE introduces only a few to several dozen MB of additional memory overhead, even under the 15-step setting. This is consistent with our expectations: during inference, LLE only needs to store a few 256$\times$256 RGB images from previous steps, typically around 0.7 MB per image, resulting in at most a few dozen MB of additional cost. This is negligible compared to model weights or other algorithm-specific overhead (e.g., back-propagation in DPS).
>
> In summary, both the training and inference overhead of LLE are minimal, which further supports the lightweight advantage of LLE. We will include these additional results and discussions in the camera ready version. We sincerely thank the reviewer for the helpful suggestion.
>
> | Steps    | 3       | 4       | 5       | 7       | 10      | 15      |
> | -------- | ------- | ------- | ------- | ------- | ------- | ------- |
> | DDNM     | 590.25  | 590.25  | 590.25  | 590.25  | 590.25  | 590.25  |
> | DDNM-LLE | 597.50  | 599.37  | 601.50  | 608.25  | 612.75  | 626.25  |
> | DPS      | 2388.78 | 2388.78 | 2388.78 | 2388.78 | 2388.78 | 2388.78 |
> | DPS-LLE  | 2396.41 | 2398.41 | 2400.41 | 2407.29 | 2411.79 | 2429.04 |
>
> **6. Missing logarithm in Equation (2).**
>
> We thank the reviewer for pointing out the error, and we will correct it accordingly. The correct form is: $\epsilon_{\theta}(\mathbf{x}_t, t)=-\sqrt{1-\overline{\alpha}_t} \nabla\_{\mathbf{x}\_t}\log q_t(\mathbf{x}_t)$.
>
> **7. The values and meaning of $\omega$ in Figure 4.**
>
> In Figure 4, the ablated parameter $\omega$ refers to the weight of the LPIPS loss in Equation (20). We tested $\omega = \{0.0, 0.025, 0.05, 0.075, 0.1, 0.2, 0.3, 0.4, 0.5\}$, which correspond in the exact order to the points from the top-right to the bottom-left in Figure 4. This result suggests that smaller values of $\omega$ encourage LLE to achieve higher PSNR, while larger values lead to better LPIPS. This provides a direct and practical way to control the trade-off between PSNR and LPIPS, depending on application-specific preferences.
>
> We thank the reviewer for the helpful comments. We will annotate the corresponding $\omega$ values in Figure 4 and add this explanation and discussion to the figure caption in the camera ready version. Sorry for the inconvenience as we are unable to include any updated figures during the rebuttal phase.
>
> **8. Limitations-"The method is dependent on the specific diffusion algorithm employed."**
>
> We agree that LLE requires training for each specific inverse algorithm. This is because different algorithms vary in their design principles, formulations, and posterior score estimations, which lead to different sampling trajectories. Consequently, the high-order approximation coefficients of these trajectories differ, necessitating separate training for each algorithm. Designing a more general approach, including aligning sampling trajectories across algorithms or developing unified high-order approximation parameters, is a promising direction for future work. We thank the reviewer for this comment.
>
> We would like to express our gratitude once again for the reviewer's comments and suggestions for our work. Should there be any further questions, we welcome continued discussion.
>
> [R.1] Fornberg, Bengt. "Generation of finite difference formulas on arbitrarily spaced grids." *Mathematics of computation* 51.184 (1988): 699-706.
>
> [R.2] Süli, Endre, and David F. Mayers. "An introduction to numerical analysis". *Cambridge university press,* 2003.

---

> > ### Comment · Reviewer_enUs · 2025-08-05
> > **Rely**
> >
> > I thank the authors and acknowledge reading their response. Most of my initial concerns have been addressed.
> >
> > Regarding point 2, I have a remaining question: If the off-diagonal elements of the matrix are expected to be zero, wouldn't this defy the purpose of extrapolation? Most coefficients would be zero, rendering the extrapolation useless. Could you please clarify this point further?

---

> > > ### Author Response · Authors · 2025-08-05
> > > **More explanation regarding point 2.**
> > >
> > > Thanks for your response and sorry for the misunderstanding. As you observed, there exist a lot of "zero" values in the third- and higher-order components in Figure 7 and 8. However, this doesn't mean all higher-order terms are exactly zero. Actually, they just decay to zero as the order grows, and these seemingly "zero" entries are in fact small but non-zero due to display precision, which also influence the performance. Evaluating these components requires nearly no extra computation. Hence, we choose to keep them for better performance. We will include these discussions in the camera-ready version of the paper. Thank you again for your valuable feedback, and please feel free to let us know if you have any further questions.

---

> ### Author Response · Authors · 2025-08-05
> **Kind Reminder to Review Our Rebuttal**
>
> Dear Reviewer enUs,
>
> Thank you once again for your helpful and constructive review.
>
> As the Discussion phase is approaching its end in less than three days, we would like to kindly remind you to take a look at our rebuttal. We have added substantial experiments and clarifications in response to your valuable comments and concerns.
>
> If there are any points we may have not fully addressed, we would be grateful if you could let us know.
>
> Looking forward to your feedback.
>
> Best regards,
>
> The Authors.

---

### Official Review · Reviewer_hxDr · 2025-07-02

**Clarity:** 3
**Significance:** 2
**Originality:** 3
**Rating:** 5
**Confidence:** 3

**Summary:**

The paper proposes Learnable Linear Extrapolation (LLE) which learns coefficients to combine previous estimates and the current prediction
for improved inference in solving inverse problems using diffusion models with few steps. The authors rewrite several diffusion-based algorithms for inverse problems
in a canoncial form which can be combined with LLE. They show consistent improvements for the different algorithms on several standard reconstruction problems.

**Questions:**

- Using the history of estimates combined with learned coefficients has also been used in Bespoke Non-Stationary Solvers (https://arxiv.org/pdf/2403.01329).
  Can the LLE method be seen as learning a type of non-stationary solver as defined in the paper?
- How are the time steps chosen? Are they just uniformly spaced? Are there methods to choose them in a more optimal way?
- What happens if we choose a large value for the number of steps $S$ and LLE? Do we still see improvements?

**Ethical Concerns:**

["NO or VERY MINOR ethics concerns only"]

**Final Justification:**

I thank the authors for the thorough rebuttal. I will keep my positive rating.

**Limitations:**

yes

**Paper Formatting Concerns:**

no formatting concerns

**Quality:**

3

**Strengths And Weaknesses:**

Strengths:

- LLE shows consistent improvements for a wide range of different algorithms
- The canoncial form combines several existing algorithms in a unified framework, which is useful for practitioners
- The paper is well written and easy to follow

Weaknesses:

- LLE requires additional finetuning and cannot be used out of the box
- Corallary 2 that motivates LLE is based on ODEs, while the canonical form is non-deterministic due to the additional noiser step. I saw in the appendix that there is an extension to the more general SDE case; this should be highlighted more in the main paper
- Improvements are only shown in the few-step regime (up to 14 steps); how large are the gaps to methods taking significantly more steps?
- I would appreciate if the experiment section included problems where the diversity of solutions/full posterior is evaluated.

---

> ### Author Rebuttal · Authors · 2025-07-31
>
> We sincerely thank the reviewer for acknowledging the effectiveness and generality of our method. Below, we provide responses to the concerns and questions raised. **All additional experimental results are reported in terms of PSNR ↑ / SSIM ↑ / LPIPS ↓.**
>
> **1. "LLE requires additional finetuning and cannot be used out of the box."**
>
> We appreciate the reviewer's comment. We'd like to highlight that the training process of LLE is highly efficient, and LLE demonstrates cross-domain generalization ability. First of all, as shown in Table 6 on page 19, the training phase of LLE takes only a few minutes, making it practical and feasible in real-world scenarios. Moreover, as presented in Table 4 on page 18, LLE exhibits a certain degree of cross-domain generalization. We further validate this in a more challenging ImageNet-to-FFHQ cross-domain setting, where LLE still achieves good performance (see the table below, where we test DDNM with 5 steps). This indicates that domain shift does not significantly undermine the effectiveness of LLE. Lastly, we acknowledge that LLE currently requires retraining for different inverse problems. Given the minimal training cost, we consider this an acceptable trade-off. Designing more efficient ways to enhance LLE's performance on cross-task generalization is a promising direction for future work.
>
> | Testset  | Trainset | Deblur (noiseless)                | Inp (noiseless)                   | SR (noiseless)                | CS (noiseless)                    |
> | -------- | -------- | --------------------------------- | --------------------------------- | ----------------------------- | --------------------------------- |
> | FFHQ     | -        | 39.82 / 0.964 / 0.087             | 26.34 / 0.719 / 0.380             | **29.99** / **0.865** / 0.210 | 18.29 / 0.591 / 0.488                 |
> |  FFHQ        | FFHQ     | **40.08** / **0.966** / **0.084** | **28.71** / **0.838** / **0.292** | 29.94 / 0.863 / **0.206**     | **19.21** / 0.605 / 0.463             |
> |  FFHQ        | ImageNet | 39.94 / 0.965 / 0.086             | 28.30 / 0.818 / 0.317             | 29.94 / 0.863 / 0.208         | 19.01 / **0.608** / 0.470         |
> | ImageNet | -        | 36.48 / 0.949 / 0.099             | 23.35 / 0.628 / 0.416             | 25.74 / **0.754** / 0.302     | 17.86 / 0.535 / 0.486             |
> |  ImageNet        | ImageNet | 36.52 / 0.949 / **0.098**         | 24.87 / 0.726 / 0.364             | **25.76**  / 0.752 / 0.297    | **18.35** / **0.541** / **0.478** |
> |  ImageNet        | FFHQ     | **36.53** / 0.949 / 0.099         | **25.14** / **0.759** / **0.341** | 25.70 / 0.750 / **0.296**     | 18.20 / 0.527 / 0.485             |
>
> **2. "The SDE version of the derivation should be highlighted more in the main paper."**
>
> We appreciate the reviewer's valuable suggestion. As the reviewer pointed out, most diffusion-based inverse algorithms are formulated in the SDE framework, and the injected random noise often plays a significant role in improving solution quality. Thus, the SDE-based derivation provided in Appendix B.3 offers a more complete explanation of the motivation and formulation of LLE. We will highlight the SDE formulation in the main text and explicitly clarify its connection to the motivation behind LLE in the camera ready version.
>
> **3. "How large are the gaps to methods taking significantly more steps?"**
>
> Regarding the comparison between the few-step LLE-enhanced method and the original algorithm with significantly more steps, we kindly refer the reviewer to Table 8 in Appendix C.5 (page 20), where we compare the performance of DDNM with LLE within 15 steps and the default DDNM with 100 steps. The results show that in many cases, LLE-enhanced DDNM with  only 15 steps achieves performance comparable to the 100-step baseline.
>
> Regarding the comparison between the LLE-enhanced method and the original algorithm both under significantly more steps, we supplement an additional experiment comparing DDNM with and without LLE under a larger number of sampling steps (25, 50, and 100) on the FFHQ dataset, as presented in the table below. To avoid the need to store an excessive number of past results when the number of steps is large, we limit the lookback window to 10 steps. The results show that as the number of steps increases, the performance gap between methods with and without LLE becomes smaller. This is consistent with our expectation as LLE searches for an improved sampling trajectory within the linear subspace spanned by previous samples. When the original algorithm takes more steps, the discretization error is reduced, and its trajectory already closely approximates the optimal one. In such cases, LLE tends to follow the original trajectory more closely, resulting in smaller gaps.
>
> Therefore, LLE brings more significant improvements in few-step regimes, which aligns with our primary objective of enhancing the performance of inverse algorithms under a limited number of steps. We will include these results and the accompanying discussions in the appendix as an additional ablation study. We appreciate the reviewer's thoughtful question.
>
> | Steps |      | Deblur (noiseless)        | Inp (noiseless)           | SR (noiseless)            | CS (noiseless)            |
> | ----- | ---- | ------------------------- | ------------------------- | ------------------------- | ------------------------- |
> | 25    | -    | 41.14 / 0.972 / 0.064     | **34.33 / 0.941 / 0.109** | 30.92 / **0.885 / 0.146** | 25.01 / 0.837 / 0.220     |
> | 25    | LLE  | **41.35 / 0.974 / 0.061** | 34.23 / **0.941** / 0.110 | **30.93 / 0.885** / 0.147 | **25.61 / 0.841 / 0.215** |
> | 50    | -    | 41.57 / 0.974 / 0.056     | **35.48 / 0.953 / 0.077** | 31.14 / **0.889 / 0.142** | **28.94 / 0.897 / 0.158** |
> | 50    | LLE  | **41.71 / 0.976 / 0.055** | 35.33 / 0.952 / 0.079     | **31.15 / 0.889 / 0.142** | 28.81 / **0.897 / 0.158** |
> | 100   | -    | 41.70 / 0.974 / 0.056     | **36.29 / 0.960 / 0.058** | **31.41 / 0.883** / 0.140 | **32.15 / 0.927 / 0.122** |
> | 100   | LLE  | **41.86 / 0.976 / 0.054** | 36.18 / 0.959 / 0.060     | 31.37 / **0.883 / 0.139** | 31.56 / **0.927 / 0.122** |
>
> **4. "Evaluating the diversity of solutions/full posterior."**
>
> Regarding the diversity on the full test set, we have included results in Appendix H that report FID scores, which serve as a quantitative measure of diversity across the dataset. Regarding the diversity of individual posteriors, we would like to clarify that the image restoration tasks considered in this paper typically have highly concentrated posteriors. For instance, in the 4$\times$ super-resolution task on 256$\times$256 images, the solution is generally close to deterministic and does not deviate significantly from the ground truth. As a result, a common practice in such settings is to evaluate performance using reconstruction metrics such as PSNR, SSIM, and LPIPS. We will include visualizations of solutions corresponding to different initial noises in the camery ready version to better illustrate the diversity of solutions. Sorry for the inconvenience that we are unable to include any visualizations during the rebuttal phase due to submission constraints. We thank the reviewer for this insightful suggestion.
>
> **5. "Can the LLE method be seen as learning a type of non-stationary solver?"**
>
> Yes, LLE can be viewed as learning a broad class of non-stationary solvers as defined in the provided paper. Specifically, the non-stationary solver in the paper is formulated as a linear combination of all previous steps' estimates and velocity fields (or scores equivalently). Accordingly, LLE can be interpreted as a data-driven approach to learn the optimal coefficients of this linear combination. We will include this paper as a reference and discuss the connection between LLE and these non-stationary solvers in the camera ready version. We thank the reviewer for the valuable comment!
>
> **6. "How are the time steps chosen? Are they just uniformly spaced? Are there methods to choose them in a more optimal way?"**
>
> Yes, in this paper, the time steps are uniformly spaced, as stated in Appendix F.2 (line 716 on page 25). We will add this clarification in the main text to make this setup clearer.
>
> Adaptive selection of time steps may indeed further improve the performance of inverse algorithms. For example, by leveraging LD3 [R.1], the time steps of inverse algorithms could be parameterized and optimized in a data-driven manner to find more optimal schedules. Combining such adaptive time step schedules with LLE is a promising direction for future work. We appreciate the reviewer's suggestion and will include this discussion in the camera ready version.
>
> **7. "What happens if we choose a large value for the number of steps for LLE?"**
>
> See our response to point 3 above.
>
> We would like to express our gratitude once again for the reviewer's comments and suggestions for our work. Should there be any further questions, we welcome continued discussion.
>
>
>
> [R.1] Tong, Vinh, et al. "Learning to Discretize Denoising Diffusion ODEs." *ICLR*, 2025.

---

> > ### Comment · Reviewer_hxDr · 2025-08-07
> >
> > I thank the authors for answering my questions and appreciate the additional results from the rebuttal. In an updated version, the authors should highlight the connection between LLE and non-stationary solvers. I will keep my current rating.

---

> > > ### Author Response · Authors · 2025-08-07
> > > **Thanks for your kind consideration!**
> > >
> > > We sincerely appreciate your insightful review and kind consideration. In the camera-ready version, we will explicitly highlight the connection between LLE and non-stationary solvers. Thanks for your valuable suggestion!

---

> ### Author Response · Authors · 2025-08-05
> **Kind Reminder to Review Our Rebuttal**
>
> Dear Reviewer hxDr,
>
> Thank you once again for your helpful and constructive review.
>
> As the Discussion phase is approaching its end in less than three days, we would like to kindly remind you to take a look at our rebuttal. We have added substantial experiments and clarifications in response to your valuable comments and concerns.
>
> If there are any points we may have not fully addressed, we would be grateful if you could let us know.
>
> Looking forward to your feedback.
>
> Best regards,
>
> The Authors.

---

### Note · Authors · 2025-08-13

Dear AC and Reviewers,

We sincerely thank the AC and the reviewers for their assistance during the review and discussion phases. In the review stage, we received many positive comments, including:

- *LLE shows consistent improvements for a wide range of different algorithms* (Reviewer hxDr)
- *A rich framework that encompasses various diffusion posterior samplers* (Reviewer enUs)
- *LLE method is a lightweight and intuitive approach, well-motivated by an analysis of high-order ODE solvers* (Reviewer bt2H)
- *They can improve upon existing diffusion-based inverse problem solvers in a lightweight fashion* (Reviewer 7BP1)

During the rebuttal stage, we carefully addressed all the reviewers' concerns and questions in detail, including

1. **Additional ablations for deeper understanding of LLE**. We provided more comprehensive cross-domain ablations, evaluated LLE under larger sampling steps, measured the extra memory cost, and analyzed performance under different optimization steps.
2. **Broader benchmarks and more base algorithms**. We extended our evaluation to more realistic datasets (ImageNet) and nine additional inverse problem solvers (five pixel space methods, three latent space methods, and one consistency model-based method). LLE yields substantial improvements across all these datasets and algorithms, further confirming its effectiveness and generality.
3. **Clearer methodological and experimental details**. We elaborated on the choice of sampling steps and the reference set for LLE. We also re-evaluated DMPS under fewer than 10 steps after step-by-step hyperparameter tuning.
4. **Deeper methodological discussions**. We added further analysis of the relationship between LLE and non-stationary solvers, as well as the forward difference approximation of higher-order derivatives.

**In the discussion stage, all reviewers acknowledged that our rebuttal effectively addressed their concerns.** Reviewer enUs further raised one remaining question regarding the nearly-zero entries, to which we provided a detailed response (though, at the time of posting this final remark, we have not yet received their reply or confirmation). **As far as we can tell, based on the reviewers' feedback, there are no known remaining concerns.**

All supplementary experiments, analyses, and discussions will be incorporated into the camera-ready version. We once again thank the AC and reviewers for their valuable feedback and assistance.

Best regards,

The Authors.

---

### Decision · Program_Chairs · 2025-09-17

**Decision:**

Accept (poster)

**Comment:**

The paper examines the computational challenges of diffusion-based inverse algorithms, which achieve strong performance but typically require many denoising steps. The authors analyze ODE solvers in the inverse setting and identify a linear combination structure underlying the approximations used to trace inverse trajectories. From this perspective, they propose a canonical form that unifies various diffusion-based inverse methods. Building on this framework, they introduce Learnable Linear Extrapolation (LLE), a lightweight technique that refines predictions by optimizing linear combination coefficients of current and past estimates, thereby mitigating the instability of analytical solvers. The authors also carry out experiments for various algorithms and tasks to show that LLE enhances efficiency and performance.

Reviewers highlighted multiple strengths: the framework provides a clear and structured perspective on diverse diffusion methods, the idea of learning linear combination coefficients is interesting, and the method consistently yields performance improvements across a broad set of inverse problems. The paper is also praised for its minimal training overhead, and thorough ablations. Some weaknesses were raised, including overlap with prior work on non-stationary solvers, the reliance on facial datasets in the initial submission, and missing comparisons to certain SOTA baselines and consistency models. However, the rebuttal effectively addressed many of these issues by adding diverse experiments, clarifying theoretical connections, and discussing extensions to broader settings. Ethical concerns regarding dataset bias and potential misuse were acknowledged and will be addressed in the final version. While one reviewer was not fully satisfied with the paper and the response of the authors, overall I think the paper merits acceptance.